# MERIT: Multilingual Semantic Retrieval with Interleaved Multi-Condition Query

**Wei Chow**[1*], **Yuan Gao**[1*], **Linfeng Li**[1*], **Xian Wang**[1], **Qi Xu**[1], **Hang Song**[1],
**Lingdong Kong**[1], **Ran Zhou**[1], **Yi Zeng**[1], **Yidong Cai**[1], **Botian Jiang**[1], **Shilin Xu**[1],
**Jiajun Zhang**[1], **Minghui Qiu**[1], **Xiangtai Li**[1], **Tianshu Yang**[1], **Siliang Tang**[2], **Juncheng Li**[2,†]

[1]ByteDance Inc.    [2]Zhejiang University    [*]Equal Contributions    [†] Corresponding Author

 **Data & Code:** MERIT-2025.github.io

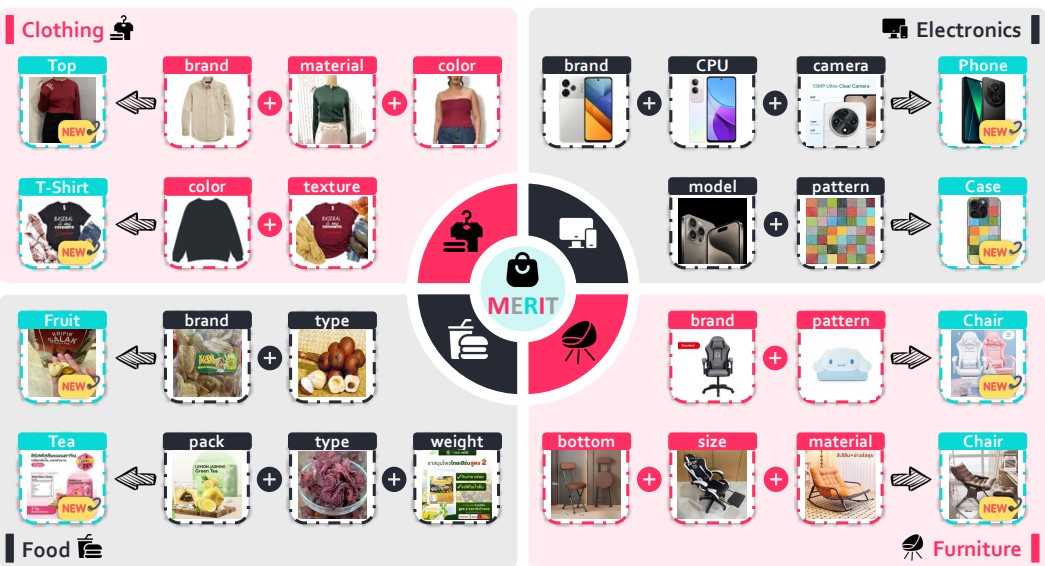

Figure 1: **Illustrative examples of interleaved multi-condition semantic retrieval.** MERIT enables the first multilingual semantic retrieval with composite multi-condition queries that interleave textual descriptions and visual references, reflecting real-world product search scenarios where users specify multiple attributes through both text and images.

## Abstract

Semantic retrieval is crucial for modern applications yet remains underexplored in current research. Existing datasets are limited to single languages, single images, or singular retrieval conditions, often failing to fully exploit the expressive capacity of visual information, as evidenced by maintained performance when images are replaced with captions. However, practical retrieval scenarios frequently involve interleaved multi-condition queries with multiple images. Hence, this paper introduces MERIT, the first multilingual dataset for interleaved multi-condition semantic retrieval, comprising 320,000 queries with 135,000 products in 5 languages, covering 7 distinct product categories. Extensive experiments on MERIT identify the existing models' critical limitation: focusing solely on global semantic information while neglecting specific conditional elements in queries. Consequently, we propose CORAL, a novel fine-tuning framework that adapts pre-trained MLLMs by integrating embedding reconstruction to preserve fine-grained conditional elements and contrastive learning to extract comprehensive global semantics.

39th Conference on Neural Information Processing Systems (NeurIPS 2025).

Experiments demonstrate that CORAL achieves a $45.9\%$ performance improvement over conventional approaches on MERIT, with strong generalization capabilities validated across 8 established retrieval benchmarks. Collectively, our contributions – a novel dataset, identification of critical limitations in existing approaches, and an innovative fine-tuning framework – establish a foundation for future research in interleaved multi-condition semantic retrieval.

# 1   Introduction

Semantic retrieval is a pivotal task that involves sourcing relevant information from vast data collections to meet specific user requirements [79, 62, 26, 45]. This task has become increasingly important with the advent of AI, as it not only enables precise user recall [100, 86, 73] but also mitigates the risk of inaccuracies in the generated content of Multimodal Large Language Models (MLLM) [2, 72].

However, semantic retrieval remains confined to narrow research scopes, which are limited to single languages [102, 84], single images [29, 11, 104], or employing only a singular retrieval condition [64, 87], as illustrated in the left part of Fig. 2. Furthermore, many existing works [86, 109, 101] fail to fully exploit the expressive capacity of images, as evidenced by their maintained performance when images are replaced with corresponding captions (Vision Unnecessary in Fig. 2). Moreover, in practical applications, product retrieval tasks frequently involve interleaved multi-condition queries (*e.g.*, specific patterns and particular texture), with many aspects requiring visual representation through images [77, 78, 8], as demonstrated in the right part of Fig. 2.

To further investigate this issue, we pose **two fundamental research questions**:

*1) How can we comprehensively measure the capability of existing models in the interleaved multi-condition semantic retrieval task?* To address this question and comprehensively assess the performance gap in interleaved multi-condition semantic retrieval tasks, we introduce MERIT, the first multilingual semantic retrieval dataset with composite multi-condition queries. Our dataset comprises 135,000 products, forming 320,000 retrieval pairs in 5 languages, covering 7 distinct product retrieval scenarios. Given the challenges in acquiring such data, we employed open-set attribute annotation to increase diversity, closed-set product annotation to improve precision and recall, and designed three sampling algorithms to enhance richness and distributional uniformity. After multiple rounds of filtering, we finalized the dataset, investing a total of 10,000 labor hours in the annotation process.

*2) What are the important factors that limit their performance, and how can we enhance the retrieval effectiveness for such a challenging task?* To address this question, we evaluate 9 existing retrieval models on MERIT and demonstrate that recall rates remain substantially below expectation, despite these methods effectively solving established semantic retrieval tasks [41, 102]. Through in-depth analysis, we identify that these methods neglect specific conditional elements in queries, failing to correctly extract targeted attributes and misinterpreting visual content. This limitation stems primarily

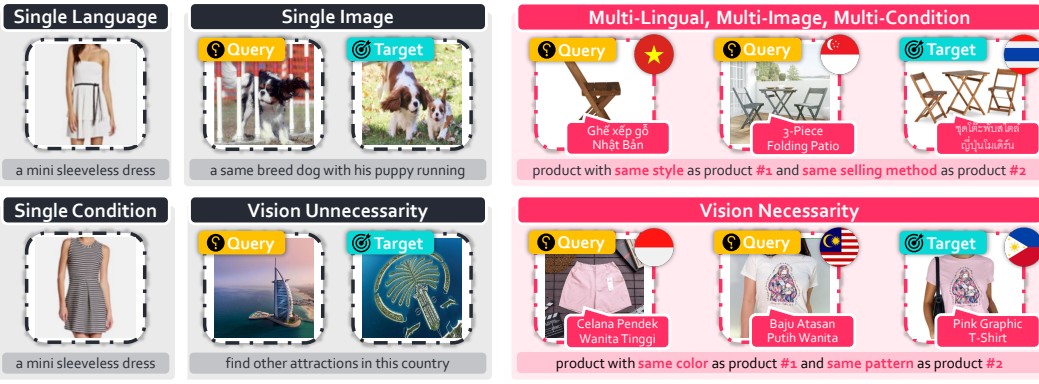

Figure 2: **Comparisons among and existing datasets** [59, 86, 101]. **Left:** Previous works are limited to single-condition, single-image, single-language scenarios. **Right:** Our benchmark enables multilingual semantic retrieval, featuring composite multi-condition queries.

from existing retrieval models [5, 38, 41] that typically fine-tune pre-trained MLLMs through contrastive learning with supervision applied exclusively at the [EOS] token [92], thereby prioritizing global semantic information while inadequately addressing specific conditional elements [46], such as material attributes in product descriptions or distinctive visual textures in images.

To address this limitation, we propose **Co**ntrastive-**r**econstruction for multimod**a**l retrieva**l** (CORAL), a novel fine-tuning framework to adapt pre-trained MLLMs into multimodal retrieval models. CORAL simultaneously preserves detailed conditional elements through multi-modal embedding reconstruction while effectively extracting global semantics via contrastive learning. Experimental results demonstrate that our method achieves a $45.9\%$ performance improvement compared to conventional approaches on MERIT, with efficacy further validated across $8$ established retrieval benchmarks.

Interestingly, we discover that existing MLLM-based retrieval models achieve performance approximately 16 times higher in R@1 when multiple images are concatenated into a single input image, compared to an interleaved input of multiple images. This occurs despite the fact that pre-trained MLLMs support interleaved image inputs, which contradicts established MLLM behavior on visual comprehension tasks [90, 47] and zero-shot performance. We hypothesize that this discrepancy may stem from existing retrieval datasets containing at most one image, potentially causing MLLMs to lose their capability to process interleaved inputs effectively. After training on MERIT, sequence input performance improved by 14.3%, further validating our hypothesis. These findings underscore the significance of our dataset as the first interleaved semantic retrieval dataset.

In summary, this paper makes three contributions to the retrieval research community:

- We introduce MERIT, the first multilingual dataset for interleaved multi-condition semantic retrieval, and provide insightful observations based on it.
- We identify critical limitations of existing methods: focusing solely on global semantic information while neglecting conditional query elements, failing to extract specific attributes, and misinterpreting visual content.
- We propose CORAL, which combines embedding reconstruction to preserve fine-grained conditional elements and contrastive learning to extract comprehensive global semantics, demonstrating strong performance across our dataset and eight standard benchmarks.

## 2 Related Work

**Multimodal Large Language Models (MLLMs)** are large-scale models that integrate visual modalities with language understanding [33, 54, 4, 66, 13, 9]. Real-world multimodal documents [108] often contain interleaved image-text pairs, and recent research [50, 47] has begun to extend MLLMs toward processing such interleaved inputs, resulting in the development of several relevant benchmarks [23, 43, 24]. Prior to the emergence of interleaved MLLMs, models supporting only single-image inputs [55, 107] typically processed multiple images by concatenating them into a single image. However, this approach results in reduced resolution and loss of sequential information, leading to inferior performance compared to sequential input of multiple images [73, 89]. Contrary to these findings, for fine-tuned retrieval models, concatenating image conditions into a single input image significantly outperforms sequential image input. Interestingly, on MERIT, models fine-tuned with sequential inputs demonstrate superior performance. This suggests that fine-tuning with previous datasets, which predominantly contain single-image examples, compromises the model's ability to maintain information across sequential inputs, further emphasizing the uniqueness of our dataset.

**Semantic Retrieval** is not only a crucial application in real-world scenarios, such as product search [86, 24, 47] and webpage retrieval [7], but also facilitates content generation (retrieval-augmented generation)[42, 88, 31, 30] and training for reasoning tasks[72]. However, existing semantic retrieval datasets are limited to single languages [102, 84], single images [29, 11, 104], or singular retrieval condition [64, 87], often failing to fully exploit [14] the expressive capacity of visual information as evidenced by maintained performance when images are replaced with captions as shown in Fig. 6. MERIT is the first multilingual dataset for interleaved multi-condition semantic retrieval. Comparison of related works can be seen in Tab. 1.

**Multimodal Retrieval Models** have primarily focused on cross-modal retrieval [58, 84, 105, 106], typically leveraging models such as CLIP [68, 29] or BLIP [84, 48] for multimodal embeddings. However, these approaches exhibit limited instruction comprehension capabilities. Subsequent

Table 1: **Summary of multi-modal query retrieval datasets.** We compare existing works from aspects including: [1]semantics, [2]multilingual data, [3]multiple types, [4]interleaved queries, [5]multi-attributes queries, and [6]whether manual annotations and filtering are applied. Note that this table does not include datasets [84, 102] that are collated and summarized but not yet newly marked.

| Benchmark | Venue | Sem. | Multi Lingual | Multi Type | Inter Leaved | Multi Attri. | Manual Anno. | #Queries |
|---|---|---|---|---|---|---|---|---|
| Fashion200K [27] | ICCV'17 | ✓ | ✗ | ✗ | ✗ | ✗ | ✓ | 200,000 |
| CIRR [59] | ICCV'21 | ✓ | ✗ | ✗ | ✗ | ✗ | ✓ | 36,554 |
| Fashion-IQ [86] | CVPR'21 | ✓ | ✗ | ✗ | ✗ | ✗ | ✓ | 20,090 |
| DTIN [70] | CVPR'23 | ✗ | ✗ | ✗ | ✗ | ✗ | ✗ | 10,000 |
| OVEN [29] | ICCV'23 | ✓ | ✗ | ✗ | ✗ | ✗ | ✗ | 139,000 |
| InfoSeek [11] | EMNLP'23 | ✓ | ✗ | ✗ | ✗ | ✗ | ✗ | 1,350,000 |
| CIRCO [6] | ICCV'23 | ✓ | ✗ | ✗ | ✗ | ✗ | ✓ | 800 |
| INSTRUCTIR [64] | arXiv'24 | ✗ | ✗ | ✗ | ✗ | ✗ | ✗ | 16,072 |
| SciMMIR [87] | ACL'24 | ✗ | ✗ | ✗ | ✗ | ✗ | ✗ | 530,975 |
| Magiclens [101] | ICML'24 | ✓ | ✗ | ✗ | ✗ | ✗ | ✗ | 36,700,000 |
| MIRACLE [63] | CVPR'24 | ✓ | ✗ | ✗ | ✓ | ✗ | ✗ | 26,221 |
| MERIT | Ours | ✓ | ✓ | ✓ | ✓ | ✓ | ✓ | 320,000 |

| Statistic | Number |
|---|---|
| ● Total Queries | 320,000 |
| - Two Conditions | 319,600 |
| - Three Conditions | 300 |
| - Four Conditions | 100 |
| ● Unique Attributes | 116 |
| - Unique Values | 2,594 |
| ● Products Number | 135,000 |
| - Maximum Product Title Length | 190 |
| - Average Product Length | 95.83 |

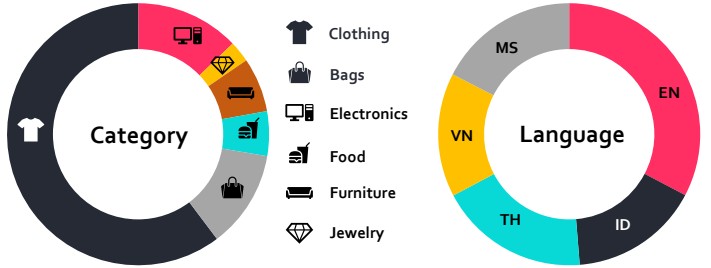

Figure 3: Dataset statistics.

Figure 4: Summary of product categories and language distributions.

research has adapted MLLMs to function as embedding models for retrieval tasks, capitalizing on their robust instruction comprehension capabilities [51, 56, 38, 102, 39, 34]. These methods [5, 38, 41] typically fine-tune existing MLLMs through contrastive learning, utilizing only the [EOS] token [92] for supervision. This approach results in an over-reliance on contrastive learning to supervise global information [92], while neglecting detailed semantic information, which often leads to semantic misunderstandings [78, 74]. To overcome these shortcomings, CORAL preserves original detailed information by integrating embedding reconstruction and contrastive learning as a fine-tuning framework to adapt pre-trained MLLMs into multimodal retrieval models. Our experiments demonstrate strong performance across MERIT and eight standard benchmarks.

## 3 MERIT: A Multi-Condition Smantic Retrieval Benchmark

To evaluate the effectiveness of existing retrieval models in addressing the interleaved multi-condition semantic retrieval task, we introduce MERIT in Sec. 3.1. Subsequently, we provide a comprehensive description of the data collection methodology in Sec. 3.2. Leveraging our data, we conduct extensive experiments on 9 state-of-the-art retrieval models and derive insights regarding visual conditioning necessity, interleaving support, and out-of-distribution scenarios in Sec. 3.3. Finally, in Sec. 3.4, we present an in-depth analysis of the factors potentially contributing to suboptimal performance.

### 3.1 Benchmark Overview

In practice, product retrieval tasks frequently encompass multiple simultaneous conditions (*e.g.*, specific patterns, precise colors, and particular styles), with many attributes necessitating visual representation through images [8]. However, existing semantic retrieval datasets are limited to single languages [102, 84], single images [29, 11, 104], or singular retrieval conditions [64, 87], often

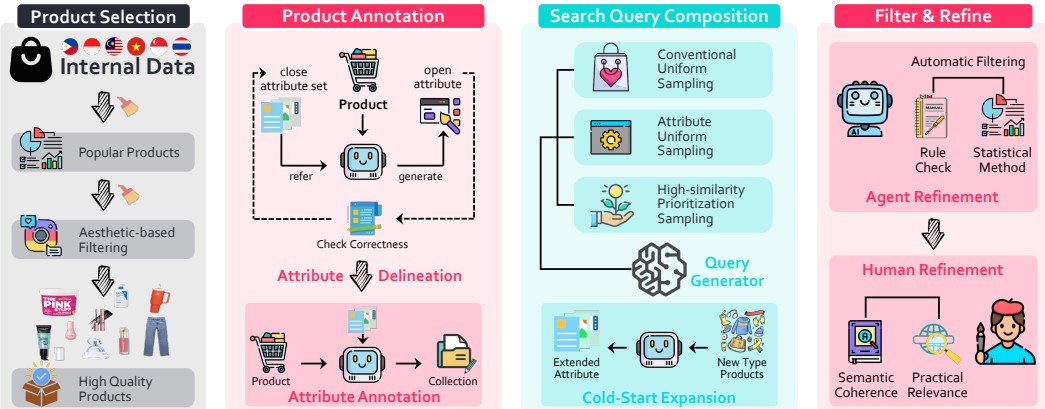

Figure 5: **The data annotation pipeline for** MERIT. We ensure data diversity and quality through open-set deduplication and multi-round filtering procedures in 4 steps. We first select high-quality products and annotate their attributes, then combine them into query pairs before performing data cleaning to produce MERIT. Details can be found in Appendix A.

failing to fully exploit [14] the expressive capacity of visual information—a limitation evidenced by maintained performance when images are replaced with textual captions.

To bridge this gap, we present MERIT, which encompasses $135,000$ products, resulting in $320,000$ retrieval pairs across 5 languages (English, Malay, Indonesian, Vietnamese, Thai), encompassing 7 distinct product retrieval scenarios. Our dataset constitutes a structured query dataset, where each fundamental unit is a product comprising an image and its corresponding title generated by GPT-4o [33], as illustrated in Fig. 2. Tab. 3 presents the key statistical characteristics of our MERIT, while Fig. 4 displays the category and language distribution of the product pool. Each search query contains at least one positive sample. For convenience, the dataset is partitioned into training and test sets, containing $310,000$ and $10,000$ entries respectively. Additional statistical results and examples are provided in Appendix C and Appendix D, respectively.

## 3.2 Data Collection Process

To ensure data quality, all data underwent manual filtering by annotators proficient in all five languages, complemented by multiple rounds of automated filtering during the collection process. Specifically, our dataset collection pipeline comprises the following four steps:

**1) High-Quality Product Selections.** While maintaining diversity, we carefully selected popular products from our internal dataset across 6 Southeast Asian countries in 5 languages with each product title is generated by GPT-4o [33]. Each product was further filtered based on popularity and aesthetic scores [71, 95] to form our product inventory used in the following steps.

**2) Product Annotations.** To accommodate diverse real-world search requirements, we needed to obtain a variety of fine-grained product attributes for combination. However, attribute information in real-world E-commerce data is often insufficient, resulting in suboptimal retrieval for specific user needs. This gap arises from the limited attribute richness constrained by operational attribute structure versus the need for fine-grained, precise product attribute information in search relevance systems. Consequently, we adopted an open annotation approach followed by statistical analysis for Attribute Delineation, and subsequently tagged products based on these derived attributes.

**3) Search Query Compositions.** To simultaneously enhance dataset quality and diversity, we implemented a composite sampling approach for constructing retrieval pairs. This approach integrates three distinct methods: Conventional Uniform Sampling (Appendix A.3.1), Attribute Uniform Sampling (Appendix A.3.2), and High-Similarity Product Prioritization Sampling (Appendix A.3.3). Furthermore, our pipeline supports cold-start expansion, enabling the extension of our dataset to previously unseen product classes, as detailed in Appendix A.3.4.

**4) Filtering & Refinement.** Finally, we introduce a two-stage filtering process, encompassing automatic filtering and manual curation, respectively. The automatic filtering stage employs rule-

Table 2: **Comparative study of retrieval performance** on MERIT. Details about the baselines can be found in Appendix E.1. "Seq", "Cat", and "Avg" denote sequential multi-image input, concatenated images as a single image input, and averaged embeddings, respectively.

| Type | Method | Size | Venue | Type | R@1↑ | R@5↑ | R@10↑ | MRR↑ |
|---|---|---|---|---|---|---|---|---|
| **Zero-Shot MLLM** | • InternVL2.5 [12] | 1B | arXiv'24 | Cat | 0.20 | 0.98 | 1.72 | 0.56 |
| | • InternVL2.5-MPO [83] | 1B | arXiv'24 | Cat | 0.24 | 1.04 | 1.81 | 0.60 |
| | • Qwen2.5-VL [5] | 3B | arXiv'25 | Cat | 0.05 | 0.27 | 0.40 | 0.14 |
| | • InternVL2.5 [12] | 1B | arXiv'24 | Seq | 0.27 | 1.24 | 1.94 | 0.69 |
| | • InternVL2.5-MPO [83] | 1B | arXiv'24 | Seq | 0.41 | 1.37 | 2.28 | 0.87 |
| | • Qwen2.5-VL [5] | 3B | arXiv'25 | Seq | 0.09 | 0.39 | 0.56 | 0.21 |
| **Embedding MLLM** | • E5-V [38] | 8B | arXiv'24 | Avg | 3.10 | 7.54 | 9.90 | 5.03 |
| | • LLaVE [41] | 0.5B | CVPR'25 | Cat | 4.89 | 33.11 | 41.98 | 16.95 |
| | • GME-Qwen2VL [102] | 2B | arXiv'24 | Cat | 8.47 | **47.13** | **56.18** | **25.02** |
| | • LLaVE [41] | 2B | CVPR'25 | Cat | 5.80 | 43.62 | 53.51 | 21.78 |
| | • LamRA-Qwen2.5VL [56] | 7B | arXiv'24 | Cat | **12.05** | 39.13 | 48.03 | 23.80 |
| | • LLaVE [41] | 7B | CVPR'25 | Cat | 8.03 | _45.34_ | _55.32_ | _24.25_ |
| | • BGE-VL [104] | 7B | arXiv'25 | Cat | _11.55_ | 38.01 | 46.26 | 23.00 |
| | • LLaVE [41] | 0.5B | CVPR'25 | Seq | 0.38 | 1.17 | 1.79 | 0.71 |
| | • GME-Qwen2VL [102] | 2B | arXiv'24 | Seq | 5.29 | 24.18 | 30.66 | 13.42 |
| | • LLaVE [41] | 2B | CVPR'25 | Seq | 0.12 | 1.03 | 1.67 | 0.51 |
| | • VLM2Vec [39] | 4B | arXiv'24 | Seq | 0.43 | 1.86 | 2.97 | 1.04 |
| | • LamRA-Qwen2.5VL [56] | 7B | arXiv'24 | Seq | 3.26 | 13.10 | 19.03 | 7.57 |
| | • LLaVE [41] | 7B | CVPR'25 | Seq | 0.39 | 1.76 | 2.77 | 1.03 |

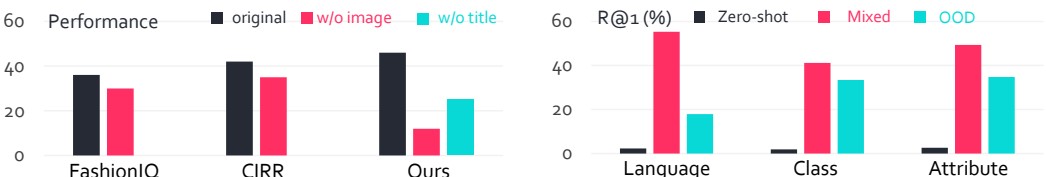

Figure 6: Comparisons on **(a)** Visual Necessity Test, and **(b)** Out-of-Distribution Scenarios.

based systems and statistical methods to eliminate obvious inconsistencies and low-quality samples, while the manual filtering stage involves expert annotators who apply nuanced judgment to ensure semantic coherence and practical relevance. This rigorous quality control process results in a high-fidelity dataset that meets stringent academic standards.

Due to space limits, we provide a more detailed description of our annotation process and the rationale behind our design choices in Appendix A.

### 3.3 How Far to MERIT

To evaluate the effectiveness of existing retrieval models in addressing the interleaved multi-condition semantic retrieval task, we conduct experiments on 9 state-of-the-art retrieval models. The principal results are presented in Tab. 2. Detailed information regarding the experimental settings and datasets can be found in Appendix E. MERIT is divided into training and test sets, consisting of 310,000 and 10,000 queries respectively as mentioned in Sec. 3.1.

**Main Results.** Existing retrieval methods struggle to address interleaved multi-condition semantic tasks, with even the best Recall@1 being only 12.05%. Additionally, we identify several key insights:

**Visual Conditioning Necessity.** To verify the necessity of visual information, we conducted experiments using BGE-VL [104] on CIRR [59], FashionIQ [86], and MERIT. We report R@1 for CIRR, R@10 for FashionIQ, and our dataset. As shown in Fig. 6(a), when replacing images with their corresponding captions for retrieval, the performance on FashionIQ and CIRR does not significantly deteriorate. In contrast, we exhibit substantial performance degradation when either replacing images with their corresponding captions (w/o image) or removing product titles (w/o title), with image removal resulting in a particularly severe decline of 73.9%. This demonstrates the effectiveness of our dataset, indicating that both images and product titles are indispensable components.

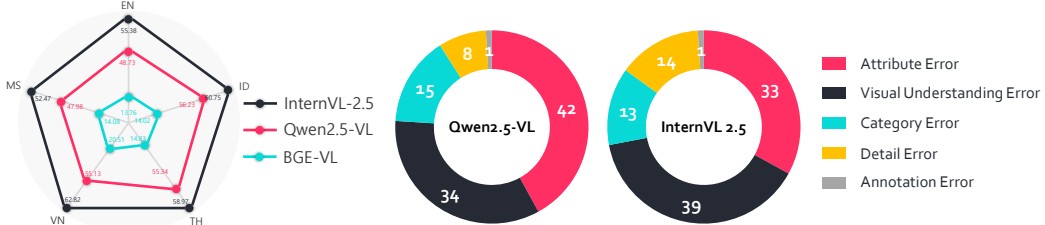

Figure 7: **(a)** Different Language's Performance on MERIT (R@1). **(b)** Distribution of Error Types.

**Interleaving Support.** As shown in Tab. 2, concatenating multiple images into a single image significantly outperforms sequential input such as GME-Qwen2VL [102], with concatenation achieving a 119.7% improvement in R@5 over its sequential version. This occurs despite the fact that pre-trained MLLMs support interleaved image inputs [5], which contradicts established MLLM behavior on visual comprehension tasks [90, 47] and zero-shot performance on MERIT, where sequential processing typically excels by preserving more image information [40, 89, 44, 1, 25]. We hypothesize that this discrepancy may stem from existing retrieval datasets containing at most one image, potentially causing MLLMs to lose their capability to process interleaved inputs. After training, sequence input performance improved by 14.3% in Tab. 3, further validating our hypothesis. This underscores the significance of MERIT as the first interleaved semantic retrieval dataset.

**Out-of-Distribution Scenarios.** We evaluated Qwen2.5-VL [5] on three types of OOD scenarios (Class OOD, Language OOD, and Attribute OOD), with results illustrated in Fig. 6(b). Detailed numeric results can be seen in Tab. 7,8,9 in the Appendix E.4. Specifically, performance in the Language OOD scenario shows a notable gap compared to full training (Mixed); however, it still demonstrates substantial improvement over zero-shot performance due to the activation of the MLLM's multilingual capabilities. In both Class and Attribute OOD scenarios, the performance gap between OOD and full training is relatively small, reflecting the diversity of our dataset.

### 3.4 Error Analysis

To investigate the poor performance of retrieval models on MERIT, we first analyzed whether success rates correlate with specific languages. As shown in Fig. 7(a), the statistical results reveal minimal variation across different languages, with no observable advantage for English despite its predominance in the initial training data of MLLMs.

We then randomly selected 500 queries and obtained explanations from Qwen2.5-VL and InternVL 2.5, both of which underwent full-parameter contrastive learning training. Expert annotators classified the root causes of mispredictions into five categories (details can be seen in Appendix E.5).

The distribution of these error types, shown in Fig. 7(b), reveals that attribute and visual understanding errors constitute the largest proportion of failures. This analysis reveals these methods neglect conditional query elements, failing to extract specific attributes and misinterpreting visual content. This likely stems from retrieval-oriented fine-tuning, where MLLMs prioritize global over specific semantic information. Furthermore, since current retrieval datasets are predominantly single-image based, existing methods fail to leverage the image sequence understanding capabilities of interleaved MLLMs as analyzed in Sec. 3.3. This limitation likely leads to failures in understanding precise semantics, resulting in attribute extraction errors (causing Attribute Errors) and incorrect interpretation of visual features such as patterns (causing Visual Understanding Errors).

## 4 CORAL: Contrastive Reconstruction for Multimodal Retrieval

Recognizing neglecting specific conditional elements in queries as a primary source of error highlighted in Sec. 3.4, we introduce CORAL in Sec. 4.1 to enhance MLLM-based retriever performance in addressing interleaved multi-condition semantic retrieval tasks through the integration of visual reconstruction during the fine-tuning process of the MLLM-to-retrieval model adaptation. Subsequently, we validate the effectiveness of our approach in Sec. 4.2.

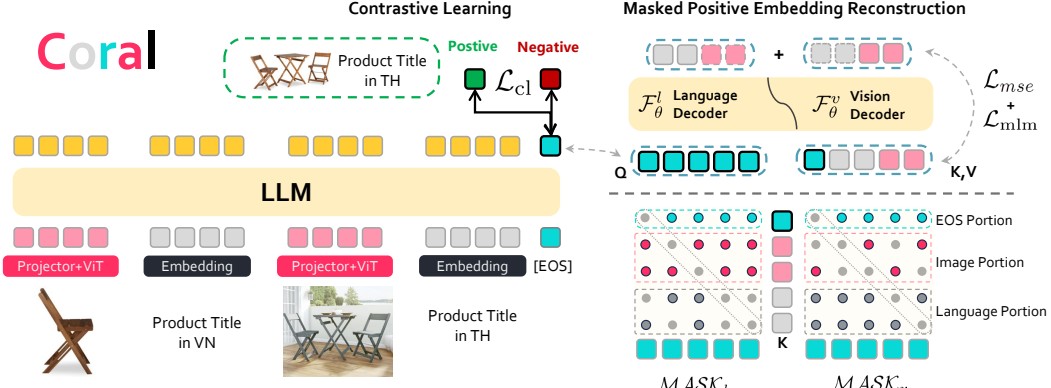

Figure 8: **Overview for CORAL.** The loss function of CORAL consists of three components: Contrastive Learning Loss $\mathcal{L}_{cl}$, Vision Reconstruction Loss $\mathcal{L}_{mse}$, and Masked Language Modeling Loss $\mathcal{L}_{mlm}$. During training, we reconstruct both the query and its corresponding positive sample.

## 4.1 Preliminaries

**Prerained MLLM.** For a common MLLM [55, 61, 40], it has image and text input $x_{img}$ and $x_{txt}$. We assume $d$ as the hidden state dimension of the language model. We first process $x_{img}$ subject to a visual representation backbone [68, 98] $V_\omega$ that outputs a sequence of features $p_{img} = V_\omega(x_{img}) \in \mathbb{R}^{N_V \times d_v}$. Next, we map $p_{img}$ to a sequence of embeddings via a learned projector $F_\psi$, where $e_{img} = F_\psi(p_{img}) \in \mathbb{R}^{N_V \times d}$. Finally, we concatenate the sequence $e_{img}$ with the text prompt embeddings $e_{txt} = \text{embed}(x_{txt}) \in \mathbb{R}^{N_L \times d}$, passing the result to the language model.

Generally, we have the interleaved image-text input $x_{input}$ by concatenating all the $e_{txt}$ and $e_{img}$. The language model generates output hidden state $h_{gen} = \text{LM}_\theta(e_{img}; e_{txt})$. In particular, we denote the hidden layer representation of the [EOS] position as $h_{eos}$. Finally, $h_{gen}$ can be transferred into text output $u_{gen}$. The composition $\text{LM}_\theta(F_\psi(p_{img}); \text{embed}(x_{txt}))$ then defines the MLLM. Given a triple $(x_{img}, x_{txt}, \hat{u}_{gen})$ during training, MLLM minimizes that loss $\mathcal{L} = -\log p(\hat{u}_{gen}|x_{img}, x_{txt})$.

**Masked Embedding Reconstruction.** During training, we apply masking to the attention maps of individual modalities. We define the functions $\mathcal{MASK}_v(E)$ and $\mathcal{MASK}_l(E)$ to mask the visual and linguistic portions [91, 80] of the input multi-modal embedding $E = [e_{img}; e_{txt}]$ at a fixed ratio $\delta$ (we set $\delta = 0.5$ in our experiments). Taking $\mathcal{MASK}_v$ as an example, for a given multi-modal input embedding, we retain all textual attention while randomly masking the visual self-attention and cross-attention from the complete text to the image. We set $\mathbf{e} = [E, h_{eos}]$ and $\mathbf{q} = [h_{eos}, h_{eos}, ...]$ with position embedding [18], which has the same length with $\mathbf{e}$. This process is illustrated in Fig. 8 (a), and the reconstructed multi-modal embedding is then obtained using:

$$\mathbf{Q} = \mathbf{q}\mathbf{W}^Q, \qquad \mathbf{K} = \mathbf{e}\mathbf{W}^K, \qquad \mathbf{V} = \mathbf{e}\mathbf{W}^V,$$

$$\mathbf{M}_{ij} = \begin{cases} 0, & \text{attended,} \\ -\infty, & \text{masked;} \end{cases} \qquad \mathbf{e}_{rec} = \text{softmax}(\frac{\mathbf{Q}^T\mathbf{K}}{\sqrt{d}} + \mathbf{M})\mathbf{V}. \tag{1}$$

**CORAL.** We introduce a fine-tuning method designed to adapt pretrained MLLMs into multimodal retrieval models. It enhances visual understanding capabilities while preserving the model's original linguistic comprehension. Specifically, for a pretrained MLLM, we perform fine-tuning as follows:
• *Contrastive Learning Loss* $\mathcal{L}_{cl}$. We employ the InfoNCE Loss [85] for supervised contrastive learning. Given a batch of $N$ samples, where $\tau$ denotes the temperature coefficient, $q_i$ represents the query sample, and $k_{i+}$ is the encoded vector of the positive sample corresponding to query $i$, the contrastive loss is computed as:

$$\mathcal{L}_{cl} = -\frac{1}{N}\sum_{i=1}^{N}\log\left(\frac{\exp\left(\frac{q_i \cdot k_{i+}}{\tau}\right)}{\sum_{j=1}^{N}\exp\left(\frac{q_i \cdot k_j}{\tau}\right)}\right). \tag{2}$$

• *Vision Reconstruction Loss* $\mathcal{L}_{mse}$. We employ a decoder $\mathcal{F}_\theta^v$, randomly initialized as a BERT layer [91, 17, 67]. Using the full input representation $h_{eos}$ as the query, we compute the MSE loss

Table 3: **Ablation results** of existing methods and CORAL on MERIT using Qwen2.5-VL [5].

| Ablation Factor | Method | LoRA | Type | R@1↑ | R@5↑ | R@10↑ | MRR↑ |
|---|---|---|---|---|---|---|---|
| ○ **Baseline** | CL | ✓ | Seq | 48.52 | 73.11 | 77.93 | 59.48 |
| | CL | ✗ | Seq | 47.76 | 73.97 | 80.47 | 59.06 |
| ○ **Input Type** | Zero-Shot | - | Seq | 0.09 | 0.39 | 0.56 | 0.21 |
| | Zero-Shot | - | Cat | 0.05 | 0.27 | 0.40 | 0.14 |
| | +CORAL (**Ours**) | ✗ | Cat | **60.94** | **85.60** | **90.40** | **71.70** |
| ○ **Partial Reconstruct** | +Vison | ✓ | Seq | 58.18 | 83.19 | 88.02 | 69.13 |
| | +Language | ✓ | Seq | 58.38 | 83.01 | 88.26 | 69.35 |
| | +Vision | ✗ | Seq | 59.46 | 85.46 | 90.81 | 70.89 |
| | +Language | ✗ | Seq | 59.98 | 86.01 | 90.72 | 71.22 |
| ● **Final Version** | +CORAL (**Ours**) | ✓ | Seq | 59.40 | 82.80 | 87.94 | 69.74 |
| | +CORAL (**Ours**) | ✗ | Seq | **69.68** | **89.26** | **93.08** | **78.33** |

Figure 9: Comparisons of our method with other methods on eight established retrieval tasks. We take zero-shot Qwen2-VL [81] as our baseline. CL denotes contrast learning.

between the original unmasked embedding and the reconstructed embedding from $\mathcal{F}_\theta^v$ as follows:

$$\mathcal{L}_{\mathrm{mse}} = -\frac{1}{N} \sum_{i=1}^{N} \left\| \hat{E} - E \right\|_2^2, \text{where } \hat{E} = \mathcal{F}_\theta^v [\, \mathcal{MASK}_v(E); h_{eos}]. \tag{3}$$

● *Masked Language Modeling Loss* $\mathcal{L}_{mlm}$. Similar to vision reconstruction, we use a decoder $\mathcal{F}_\theta^l$ for reconstruction. To reduce trainable parameters, $\mathcal{F}_\theta^l$ shares weights with the language modeling head of the MLLM. The masked language modeling loss is computed as:

$$\mathcal{L}_{\mathrm{mlm}} = -\frac{1}{N} \sum_{i=1}^{N} \log P\left(\hat{x}_i \mid X\right), \text{where } \hat{x}_i = [\mathcal{F}_\theta^l [\, \mathcal{MASK}_l(E); h_{eos}]]_{(i)}. \tag{4}$$

The overall training objective of CORAL is formulated as:

$$\max_{\theta, \theta_v, \theta_l} \mathcal{L} = \mathcal{L}_{\mathrm{cl}} + \lambda_1 \mathcal{L}_{\mathrm{reg}} + \lambda_2 \mathcal{L}_{\mathrm{rec}}. \tag{5}$$

Here, $\mathcal{L}_{\mathrm{reg}}$ and $\mathcal{L}_{\mathrm{rec}}$ represent the reconstructions of the retrieval target using the conditions' [EOS] token and the target's own [EOS] token as attention queries, respectively. For both terms, the attention keys and values referenced in Eq. 1 are derived from the embeddings of the retrieval target. Each reconstruction component encompasses both image reconstruction and language reconstruction.

## 4.2 Experiments

To validate the effectiveness of CORAL, we conducted experiments on MERIT and 8 established retrieval tasks. Due to space constraints, the implementation details are placed in the Appendix.

**Main Results on** MERIT. Results lead to the following conclusions: *(i) Embedding reconstruction contributes significantly to retrieval performance.* Both partial feature reconstruction (Tab. 3, rows 6-11) enhance model performance, with multimodal reconstruction yielding a 45.9% improvement compared to contrastive learning alone. *(ii) Multi-modal reconstruction outperforms partial reconstruction.* Comparing Tab. 3, rows 6-9 and 10-11 reveal superior performance when reconstructing

both modalities simultaneously. *(iii) Sequential input surpasses image concatenation.* Based on rows 3-5 and 11, we observe that sequential inputs achieve higher performance. We hypothesize that sequential representation preserves more information than image concatenation [37, 97, 24, 47], which aligns with our findings in Sec. 3.3. *(iv) Full parameter fine-tuning yields optimal results.* Due to the substantial divergence between retrieval tasks and pre-training objectives, full parameter fine-tuning generally produces better outcomes, consistent with conclusions from previous work [41, 39].

**Results on Eight Retrieval Tasks.** To further validate the efficacy of CORAL, we conducted evaluations on 8 retrieval benchmarks with experimental configurations following the methodology described in [39]. The results are illustrated in Fig. 9. Comparative analyses between our approach and other foundational models, such as CLIP [68] and E5-V [38], are presented in Appendix E.4. Experimental results demonstrate that our method achieves consistent improvements across these eight retrieval tasks, with particularly notable performance on VisDial [16], where our approach exhibits a 181% enhancement over the baseline.

# 5 Conclusion

This paper introduces MERIT, the first multilingual dataset for interleaved multi-condition semantic retrieval. Extensive experiments identify existing critical limitations of existing models: focusing solely on global semantic information while neglecting conditional query elements, failing to extract specific attributes, and misinterpreting visual content. To address this limitation, we propose CORAL, a novel fine-tuning framework that adapts pre-trained MLLMs by integrating embedding reconstruction for preserving detailed conditional elements with contrastive learning for extracting global semantics. Ablation study across benchmarks demonstrates the effectiveness of CORAL.

**Acknowledgment.** This work was supported by the National Natural Science Foundation of China (62436007).

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

# Contents

# A    Detailed Dataset Collection Process

The complete data generation pipeline is illustrated in Fig. 5. Below, we provide a more detailed description of our annotation process and the rationale behind our design choices than in the main text. We confirm that all data has been anonymized and complies with open-source guidelines. Specifically, in Appendix A.1, we describe the sources of our product data, and in Appendix A.2, we elaborate on our methodology for labeling products to obtain diverse attributes.

Subsequently, in Appendix A.3, we describe our approach for generating a rich set of retrieval pairs (which were later manually filtered and curated). Specifically, to enhance dataset quality and diversity, we employed a composite sampling approach integrating three methods: Conventional Uniform Sampling (Appendix A.3.1), Attribute Uniform Sampling (Appendix A.3.2), and High-similarity Product Prioritization Sampling (Appendix A.3.3). Additionally, our pipeline supports cold-start expansion as detailed in Appendix A.3.4.

Finally, we introduce our two-stage filtering process in Appendix A.4.1 and Appendix A.4.2, encompassing automatic filtering and manual curation, respectively. The automatic filtering stage employs rule-based systems and statistical methods to eliminate obvious inconsistencies and low-quality samples, while the manual filtering stage involves expert annotators who apply nuanced judgment to ensure semantic coherence and practical relevance. This rigorous quality control process results in a high-fidelity dataset that meets stringent academic standards.

## A.1    Product Source

The MERIT dataset comprises high-quality, anonymized items from our internal dataset. Specifically, we curated popular products across six Southeast Asian countries, covering five languages. The items were sampled from both category-specific listings (as the primary source) and high-quality open-category offerings to enhance data diversity. Each item underwent aesthetic scoring [71, 95] before inclusion in the final repository. The distribution of categories and their respective quantities is detailed in Tab. 4. Note that these categories are simplified for annotation purposes and do not reflect the full hierarchical taxonomy. All sensitive information has been removed to comply with open-source data standards.

However, among these well-performing products, some exhibit suboptimal visual presentations that diminish their search augmentation potential. The imagery associated with these products lacks visual efficacy and fails to fulfill the objective of facilitating retrieval through visual cues. Fig. 10 illustrates several examples of such products.

## A.2    Attribute Delineation and Annotation

### A.2.1    Attribute Delineation

To support diverse real-world retrieval needs, it is essential to obtain fine-grained and varied product attributes for effective combinatorial search. However, existing product retrieval data often suffers from insufficient attribute richness, leading to suboptimal recall for specific user queries. This limitation stems from a fundamental gap: operational attribute schemas (typically rigid and sparse) fail to meet the demands of search relevance tasks, which require precise, granular attribute information.

Directly predefining attributes and values for each product category – whether through manual annotation or MLLMs [33, 76, 21] – introduces two challenges: (1) restricted coverage due to finite predefined options, and (2) biases (human or LLM-induced) that may overlook critical product features. To address this, we propose an open-ended attribute delineation approach, where attributes and candidate values are first generated openly and then refined. As illustrated in Fig. 11, our pipeline begins with raw platform properties, employs LLM verification to ensure diversity, and finally derives structured attribute-value pairs through frequency inversion and label refinement.

Specifically, the **LLM-based attribute extraction** module takes a product's image and textual information as input and generates a set of open-ended attributes. Figs. 12, 13, 14, 15, and 16 provide

Table 4: Product Categories and Countings in MERIT.

| Primary Type | Second Type | Number |
|---|---|---|
| **Food** | • Fruits | 1,001 |
| | • Snacks | 3,077 |
| | • Beverages | 3,481 |
| **Clothing** | • Tops | 32,100 |
| | • Pants | 26,398 |
| | • Shoes | 24,267 |
| **Electronics** | • Phones | 11,671 |
| | • Headphones | 4,670 |
| | • Laptops | 842 |
| | • Other Screens | 351 |
| **Bags** | • Backpacks | 5,881 |
| | • Handbags | 4,755 |
| | • Suitcases | 5,540 |
| **Jewelry** | • Gold | 405 |
| | • Silver | 608 |
| | • Diamonds | 590 |
| **Furniture** | • Tables | 3,104 |
| | • Chairs | 6,101 |
| **Others** | | 158 |
| **Total** | | 135,000 |

examples of annotated outputs. However, such open-set annotations frequently generate excessive irrelevant attributes (*i.e.*, attributes that do not align with practical product retrieval needs). For instance, in Fig. 12, attributes such as "Isbn/Issn": ["9780226264219"], "Language": ["English"], "Version": ["First Edition"], "Cover Type": ["Paperback"], "Number Of Pages": ["464"], "Features": ["27 photographs", "6 maps", "25 illustrations/diagrams"], "Publication Date": ["May 15, 2018"], and "Publisher": ["Basic Books"] are rarely used for product retrieval queries and should therefore be excluded. Similarly, in Fig. 14, the attribute "Batteries Included": ["no"] is ambiguous and unsuitable for retrieval purposes.

The **LLM attribute verification** step evaluates whether the extracted attributes are reasonable for the given product category, retaining only those that are valid to narrow down the candidate set. During this process, we also standardize the format of attributes and their corresponding values by ensuring consistent formatting, avoiding capitalization of first letters, and removing underscores.

Next, **attribute frequency inversion and filtering** organizes the verified attributes by product category, as outlined in Tab. 4. This step accounts for the distinct characteristics of different products– for instance, clothing may include attributes such as pattern, neckline style, zipper type, and sleeve length, whereas electronics like smartphones may feature RAM, storage capacity, and battery size.

Considering these factors, we employed the following methodology to define a closed set of attributes:

- **Rule-based filtering**: We initially identified 100 attributes using inverted indexing (employing normalized strings: converting all text to lowercase and removing extraneous spaces

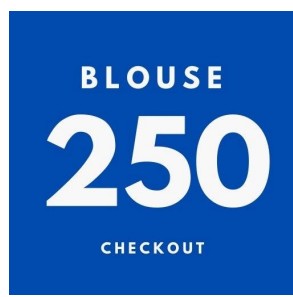 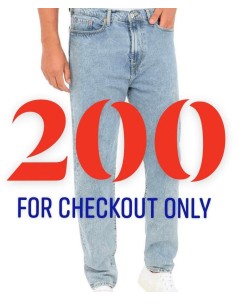 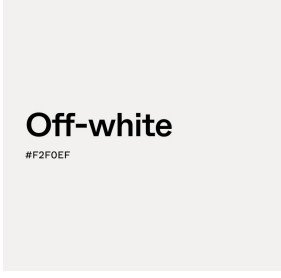

Silk Blouse for Work: Versatile, Comfortable, and Stylish Office Attire

Men's Regular Fit Cotton Blend Denim Jeans with Button Closure

Kerudung Burberry Off White dengan motif kupu-kupu

Figure 10: Examples of low-quality products. The associated images fail to accurately represent the visual characteristics of the products, instead containing irrelevant content.

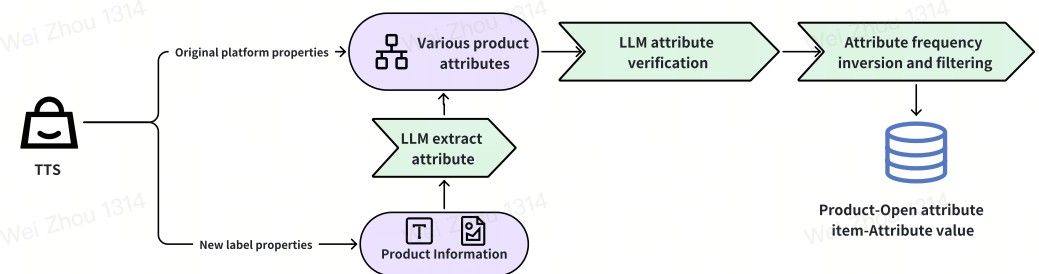

Figure 11: **An Overview of our Attribute Delineation Pipeline**. The process begins with extracting original platform properties and diverse product attributes, followed by LLM-based attribute verification. We then perform attribute frequency inversion and generate refined labels, resulting in structured product information with open-domain attribute-value pairs.

> and punctuation). Selecting frequently occurring instances helps avoid annotating niche attributes that are unlikely to be used in retrieval tasks.

- **Human-LLM collaboration**: Based on the 100 identified attributes and our domain expertise, we curated a distinctive set of 20 attributes for each product category.

The prompt used for attribute selection via LLM is as follows:

> You are an assistant who communicates only in JSON and is an expert in physics. Do not I am working with a dataset for attribute retrieval in clothing categories. Below is a list of extracted attributes. Your goal is to help me select the top 20 attributes that best meet the following criteria:
> Relevance: The attribute should be common across multiple clothing categories (*e.g.*, dresses, shirts, pants).
> Searchability: The attribute should be easily combinable with other attributes in queries (*e.g.*, "long sleeve + striped").
> User Demand: The attribute should reflect features users frequently search for or filter by.
> Visual Detectability: The attribute should be identifiable from a product image (*e.g.*, "color", not subjective traits).
> Please prioritize attributes that satisfy all four criteria.
> If needed, suggest rephrasing attributes for clarity or consistency (*e.g.*, "sleeve length" instead of "long sleeve").
> Exclusions: Exclude overly niche or rarely used attributes.
> No overlap: Remove synonymous terms
> You should return your answer in a Python list:
> list: [tape here]

The following is an example of the list of attribute values returned by this operation (products of cloth type):

```
{
"color": ["red", "green", "yellow", "orange", "purple"],
"size": ["small", "medium", "large", "extra large"],
"flavor": ["sweet", "sour", "tart", "bland", "tangy"],
"fruit type": ["apple", "berry", "citrus", "melon", "tropical"],
"organic": ["organic", "non-organic"],
"quantity per pack": ["1 piece", "3-pack", "5 lbs", "10 kg"],
"ripeness": ["unripe", "ripe", "overripe"],
"texture": ["smooth", "bumpy", "fuzzy", "rough"],
"shape": ["round", "oval", "elongated", "lobed"],
"weight": ["100g", "500g", "1kg", "2kg"],
"pattern": ["striped", "spotted", "solid", "mottled"],
"seed presence": ["seedless", "with seeds"],
"skin thickness": ["thin-skinned", "thick-skinned"],
"juiciness": ["juicy", "dry", "moderate"],
"sweetness": ["low sugar", "medium sweet", "very sweet"],
"storage type": ["refrigerate", "room temperature", "cool dry place"],
"region of origin": ["Florida", "California", "Chile", "Spain"],
"material": ["plastic clamshell", "mesh bag", "paper bag", "loose"],
"height": ["short", "medium", "tall"],
"allergen information": ["citrus-free", "latex-free", "sulfite-free"]
}
```

After obtaining the attributes, we expanded each attribute to its corresponding values. Specifically, similar to the previous step, we employed the following methodology:

- **Rule-based approach:** We retrieved the top 100 values for each of the 20 attributes through an inverted index (ensuring that attributes were clearly specified and excluding cases where attribute values contained multiple values). The returned content was consistent with the previous step.

- **Human + LLM approach:** From the 100 values for each attribute, we identified 20 distinctive values.

The prompt used for attribute selection via LLM is as follows:

I am working on attribute retrieval for clothing categories and need to select the top 10 to 20 values per attribute for labeling. The chosen values should optimize for:
Relevance – Common across multiple clothing categories (*e.g.*, dresses, shirts, pants).
Searchability – Easily combinable in queries (*e.g.*, "long sleeve + striped").
User Demand – Frequently searched or filtered by users.
Visual Detectability – Clearly identifiable from product images. For example, the specific numerical value of weight and size cannot be seen and should not appear in your answer).
Universality – Collectively covers all scenarios without redundancy.

Requirements:
No overlap: Remove synonymous terms (*e.g.*, "round neck" → "crew neck").
Prioritization: Favor attributes that meet all five criteria.
Clarity: Suggest rephrasing for consistency (*e.g.*, "button-front" vs. "buttoned").
Exclusions: Omit niche, ambiguous, or rarely used terms (*e.g.*, "peasant sleeve").

Please return the attribute (dict's key) with its choice values (dict's value) as a JSON.
[tape here]

Our approach yields attributes with two key advantages:

**Title:**

Sapiens: A Brief History of Humankind Paperback – May 15, 2018

**Description:**

From a renowned historian comes a groundbreaking narrative of humanity's creation and evolution - that explores the ways in which biology and history have defined us and enhanced our understanding of what it means to be "human." One hundred thousand years ago, at least six different species of humans inhabited Earth. Yet today there is only one - homo sapiens. What happened to the others? And what may happen to us? books about the history of humanity pursue either a historical or a biological approach, but Dr. Yuval Noah Harari breaks the mold with this highly original book that begins about 70,000 years ago with the appearance of modern cognition. From examining the role evolving humans have played in the global ecosystem to charting the rise of empires, Sapiens integrates history and science to reconsider accepted narratives, connect past developments with contemporary concerns, and examine specific events within the context of larger ideas. Dr. Harari also compels us to look ahead, because over the last few decades humans have begun to bend laws of selection that have governed life for the past four billion years. We are acquiring the ability to design not only the world around us, but also ourselves. Where is this leading us, and what do we want to become? Featuring 27 photographs, 6 maps, and 25 illustrations/diagrams, this provocative and insightful work is sure to spark debate and is essential reading for aficionados of Jared Diamond, James Gleick, Matt Ridley, Robert Wright, and Sharon Moalem.

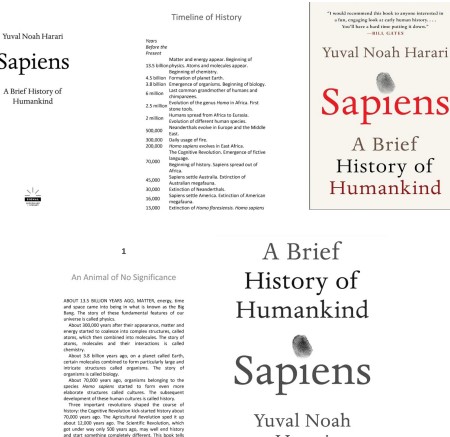

**Attribute:**

{"Special Selling Point": ["Groundbreaking narrative", "Explores human evolution", "Integrates history and science"], "Isbn/Issn": ["9780226264219"], "Language": ["English"], "Version": ["First Edition"], "Cover Type": ["Paperback"], "Number Of Pages": ["464"], "Features": ["27 photographs", "6 maps", "25 illustrations/diagrams"], "Gift For": ["History enthusiasts"], "Year": ["2018"], "Author": ["Yuval Noah Harari"], "Publication Date": ["May 15, 2018"], "Publisher": ["Basic Books"]}

Figure 12: Open-ended Attributes Annotated Product Case.

**Title:**

Even Darkness Must Pass Sticker: Samwise Gamgee Lord of the Rings Quote, Tolkien Inspired, Waterproof Vinyl Decal

**Description:**

Embrace the wisdom of Middle-earth with our Even Darkness Must Pass sticker. Featuring the iconic quote from Samwise Gamgee, this sticker is a perfect reminder of hope and perseverance. Design: Inspired by J.R.R. Tolkien's Lord of the Rings. Features the powerful quote: "Even darkness must pass." Material & Features: Made from high-quality PVC/PP material. Waterproof and stain-resistant for durability. Easy-remove adhesive for hassle-free application. Flat sticker type with smooth finish. Perfect For: Laptops, water bottles, notebooks, and more. Decorating walls, windows, doors, and wardrobes. Ideal for any fan of Lord of the Rings and inspiring quotes. Dimensions: Height: 1 inch Width: 8 inches Length: 8 inches Weight: 0.02 lbs Care Instructions: Apply to a clean, dry surface. Avoid excessive bending or scratching to maintain quality. Carry the spirit of Samwise with you, wherever you go. Get your Even Darkness Must Pass sticker today!

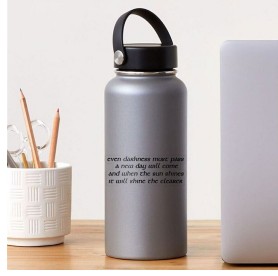

**Attribute:**

{"Special Selling Point": ["Inspired by J.R.R. Tolkien's Lord of the Rings", "any fan of Lord of the Rings"], "Length": ["8 inches"], "Size": ["Height: 1 inch", "Width: 8 inches", "Length: 8 inches"], "Batteries Included": ["No"], "Material": ["PVC/PP"], "Product Type": ["Flat sticker"], "Occasion": ["Decorating"], "Features": ["Waterproof", "Stain-resistant", "Easy-remove adhesive", "Flat sticker type"], "Gift For": ["Fans of Lord of the Rings"], "Pattern/Print": ["Quote"], "Color": ["Black"], "Benefits": ["Durability", "Hassle-free application"], "Style": ["Inspired by J.R.R. Tolkien"], "Room/Location": ["Walls", "Windows", "Doors", "Wardrobes"], "Quantity Per Pack": ["1"]}

Figure 13: Open-ended Attributes Annotated Product Case.

**Title:**

Watchful Monk iPhone Tough Case

**Description:**

Materials & Design: •\tMade from durable polycarbonate (outer shell) and TPU (inner lining) for added shock absorption. •\t2-piece design enhances impact resistance and disperses shock effectively. •\tIncludes an interior rubber liner for extra protection (may vary based on phone model). •\tGlossy finish for a sleek and polished look. •\tWireless charging compatible (MagSafe not included). Technology & Inks: •\tDesigned using dye sublimation for vibrant, long-lasting prints. •\tThe process involves printing on heat-resistant transfer paper and embedding the image into the case with a heat press. Production Time: •\tCases are produced within 1-2 business days, ensuring quick turnaround times. Shipping: •\tAfter production, orders are shipped within 3-5 business days, depending on the destination. Quality Control: •\tEach case undergoes individual inspection for print quality, design alignment, and functionality before shipping. Packaging Control: •\tPackaged in peel-and-seal mailing envelopes that ensure the case is securely protected during transit.

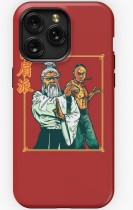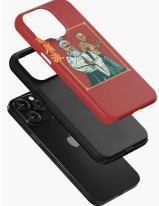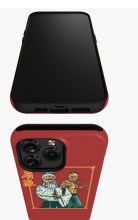

**Attribute:**

{"Batteries Included": ["no"], "Applicable Mobile Brand": ["iPhone"], "Material": ["polycarbonate", "TPU", "rubber"], "Case Type": ["tough case"], "Features": ["shock absorption", "wireless charging compatible"], "Pattern/Print": ["anime print"], "Color": ["red"]}

Figure 14: Open-ended Attributes Annotated Product Case.

**Title:**
Make Christmas Great Again, Funny Trump Dancing Ugly Sweater, Trump 2024 Christmas All-Over-Printed Sweatshirt P161110

**Description:**
Make Christmas Great Again, Funny Trump Dancing Ugly Sweater, Trump 2024 Christmas All-Over-Printed Sweatshirt P161110 BrianJoseph86s Product Details _ Weight: Crafted with a substantial weight of 300gsm, our ugly sweater ensures exceptional warmth and comfort, making it perfect for those chilly days. _ Soft & Cozy: Featuring a brushed texture, this sweater offers a luxuriously soft and plush feel against the skin, ensuring you stay comfortable all day long. _ Durable Construction: Designed to withstand wear and tear, the thick and sturdy fabric makes it an ideal choice for colder climates, providing lasting quality. _ Moisture-Absorbent: Our ugly sweater effectively wicks away moisture, keeping you dry and comfortable whether you're indoors or outdoors. Care Instructions _ To maintain the quality and longevity of your sweater, please follow these care instructions: _ Washing: Machine wash in cold or warm water on a gentle cycle to preserve the fabric and colors. _ Drying: You may tumble dry on low heat or opt for line drying to ensure the sweater maintains its shape and softness. _ Ironing: If needed, iron on low heat using a pressing cloth to avoid direct contact with the fabric. _ Stain Treatment: For stains, treat with a mild detergent and avoid bleach to prevent damage to the fabric. _Storage: Store your sweater in a cool, dry place away from direct sunlight to preserve its vibrant colors and integrity. _ Avoid: Please do not dry clean or soak the sweater, as this may lead to unwanted damage.

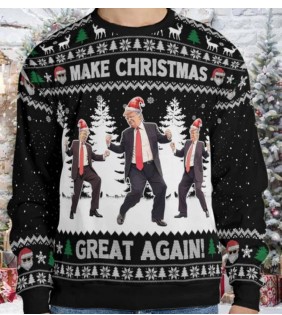

**Attribute:**
{"Design": ["Trump 2024 Christmas", "Funny Trump Dancing"], "Closure Type": ["Pullover"], "Suitable Seasons": ["Winter"], "Neckline": ["Crew Neck"], "Size": ["P161110"], "Fit Type": ["Regular Fit"], "Sleeve Length": ["Long Sleeve"], "Care Instruction": ["Machine wash cold", "Tumble dry low", "Iron on low heat", "Store in cool, dry place", "Avoid dry clean"], "Material": ["300gsm fabric"], "Occasion": ["Christmas"], "Features": ["Soft & Cozy", "Durable Construction", "Moisture-Absorbent"], "Gift For": ["Those who enjoy political satire"], "Pattern/Print": ["All-Over-Printed"], "Color": ["Black"], "Style": ["Ugly Sweater"], "Composition": ["300gsm"], "Quantity Per Pack": ["1"]}

Figure 15: Open-ended Attributes Annotated Product Case.

**Title:**
Lipstick Silk Finish, Warna Bibir yang Kaya dan Mudah Ditumpuk dengan Kandungan Pelembap, Diformulasikan dengan Vitamin A, E, & Minyak Macadamia untuk Hidrasi Maksimal

**Description:**
LIPSTICK SILK FINISH: Warna bibir kami yang sangat pigmen diformulasikan dengan Vitamin A & E, Lidah Buaya, dan minyak kacang macadamia kaya antioksidan, sehingga bibir terasa lembap dan halus dengan warna yang mencolok.
RAWAT BIBIR ANDA: Lipstick Silk Finish memberikan warna tahan lama yang bold dengan kandungan vitamin dan bahan alami, sehingga Anda bisa menciptakan tampilan bibir sempurna. Gunakan sendiri atau lapisi dengan lip liner dan lip gloss kami.
TAMPIL SEMPURNA: Mulai dari bronzer, blush on, primer, highlighter, bedak natural, hingga concealer—kami menyediakan semua yang Anda butuhkan untuk tampil flawless dan siap menghadapi momen penting. Kini, Anda telah menemukan pasangan makeup terbaik. BEBAS KEKERASAN HEWAN 100%: Produk kami bebas uji coba pada hewan (*cruelty-free*) sejak awal. Kami juga menyediakan produk vegan dan alat makeup seperti foundation, kutek, palet kontur, lipstik, kuas makeup, dan lainnya.
STAY WILD: Tujuan kecantikan terpercaya bagi pecinta makeup dari segala usia, suku, warna kulit, dan status ekonomi. Siapa pun Anda dan di mana pun Anda berada, kami punya produk yang cocok untuk Anda.

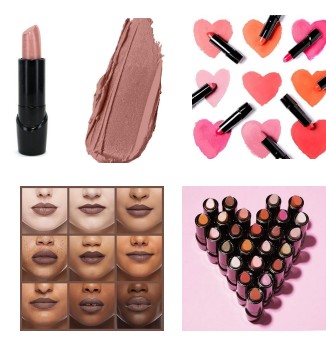

**Attribute:**
{"Special Selling Point": ["Vitamins A & E", "Aloe Vera", "Macadamia nut oil"], "Makeup Finish Type": ["Silk Finish"], "Suitable Seasons": ["All seasons"], "Application Area": ["Lips"], "Product Form": ["Stick"], "Skin Types": ["All skin types"], "Gender": ["Unisex"], "Batteries Included": ["No"], "Product Type": ["Lipstick"], "Occasion": ["Face forward"], "Features": ["Hydrating", "Long-lasting", "Bold color", "Cruelty free"], "Gift For": ["Beauty lovers"], "Color": ["Rich"], "Benefits": ["Moisturizing", "Long-lasting bold color"], "Pack Type": ["Single"], "Quantity Per Pack": ["1"]}

Figure 16: Open-ended Attributes Annotated Product Case.

1. **Diversity and Reduced Bias**: The extracted attributes are both comprehensive and varied, demonstrating greater resistance to biases from either the LLM or human annotators [3, 32], while simultaneously improving recall of long-tail attributes.

2. **Product-Centric Representation**: Since attributes are first annotated at the product level before being aggregated, they more accurately reflect real-world product characteristics.

### A.2.2 Attribute Annotation

Through the methodology described in Appendix A.2.1, we obtain a rich set of attributes and their corresponding values for each product category. This attribute set forms a closed set that is suitable for retrieval tasks. Utilizing these closed attribute tables, we can annotate each product with its corresponding attributes. Specifically, we employ the following prompt:

Objective:
Annotate clothing product attributes based on the provided inputs. Your annotations must align with the given attribute options and labeling rules.

Inputs:
List of Attributes and Options: A predefined list of attributes (*e.g.*, "color," "sleeve length") and their valid options.
Product Image: The primary reference for annotation.
Image Attribute Table (Optional): Existing attribute descriptions for reference (may be incomplete or empty).
Product Description (Optional): Textual description of the product (may be inaccurate – verify against the image).

Labeling Rules:
Select the most accurate option for each attribute based on visible/verifiable details. Skip an attribute if it cannot be determined from the inputs. Do not add new attributes. Leverage the Image Attribute Table (if available):
If an attribute (or a semantically similar one) exists in the table, summarize its content as one of the predefined options.
If your summarized value isn't in the options, ensure it adheres to:
Exclusions: Omit niche/rare attributes.
No Overlap: Remove synonymous terms.

Your Output Format:
Please output your answer in JSON format. The answer is a dict, where the key is the attribute and the value is the value of the attribute

The list of Attributes and Options is [paste here]
Product Image is [paste here]
Image Attribute Table is [paste here]
Product Description is [paste here]

Following annotation completion, automatic filtering is performed to: eliminate newly constructed attributes; remove attributes with values designated as "none" or "skip"; exclude the most frequent value within each attribute category to avoid generic descriptors such as "all ages"; and discard terms that appear only once in the dataset.

## A.3   Retrieve Data Build

To enhance dataset quality and diversity, we employed a composite sampling approach integrating three methods: Conventional Uniform Sampling (Appendix A.3.1), Attribute Uniform Sampling (Appendix A.3.2), and High-similarity Product Prioritization Sampling (Appendix A.3.3). Additionally, our pipeline supports cold-start expansion as detailed in Appendix A.3.4. It is worth emphasizing that all data generated by our automatic combination algorithm subsequently underwent manual review and refinement. Furthermore, due to the inherent limitations of closed-set attributes, many matches lacked precision (*e.g.*, products labeled as "Bohemian style" might not exhibit particularly similar stylistic elements, thus being filtered out). Consequently, a significant proportion of data was eliminated during post-processing, reflecting the high quality standards of our final dataset.

### A.3.1   Conventional Uniform Sampling

Conventional Uniform Sampling is our most frequently utilized sampling algorithm. Specifically, we randomly select two products and randomly choose product attributes, then retrieve all products from the product pool that simultaneously satisfy these two conditions. We apply certain constraints to these recalled products:

- The value of attribute $i$ for product $a$ and product $b$ cannot be identical, as this would render the use of two products for retrieval unnecessary

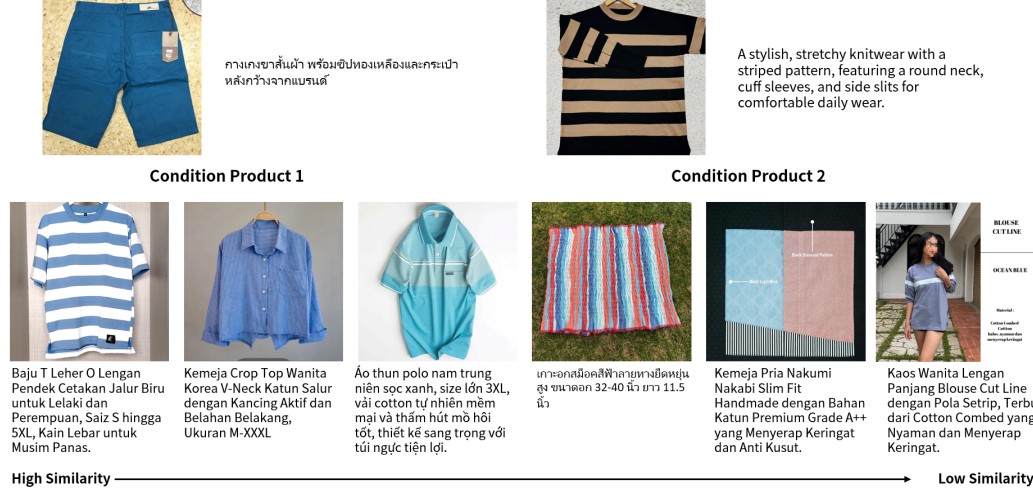

Figure 17: **The CLIP Similarity** [35, 98] textbfRanking of Recalled Products Case 1. The first row is the two condition products, and the second row is the recalled products. The CLIP similarity with the condition products decreases from left to right.

- When randomly selecting values, we employ a probability distribution that suppresses frequently occurring keys, such as color
- Our dataset contains 2-4 conditions per query. When forming combinations, we set the probability distribution as follows: 75 for 2 conditions, 10 for 3 conditions, and 15 for 4 conditions (the higher proportion for 4 conditions accounts for potential filtering during subsequent annotation)

At this stage, numerous products still remain. To enhance the efficiency of subsequent manual annotation, we use CLIP Similarity to rank all qualifying samples according to their average similarity with the two products in descending order. This arrangement prioritizes similar products, facilitating the annotation process. The CLIP Similarity Ranking of Recalled Products can be seen in Figs. 17, 18, and 19. These figures demonstrate that CLIP similarity ranking serves a meaningful purpose.

However, this sampling method suffers from a long-tail distribution of attributes (uncommon attributes rarely appear in the combined retrieval entries). Furthermore, due to the extensive size of the product repository, we do not process all qualifying products in each iteration but instead conduct selective sampling, which may result in the absence of products that genuinely match the conditions in the recalled set. To address these issues, two novel sampling algorithms are introduced in Appendix A.3.3 and Appendix A.3.4.

### A.3.2  Attribute Uniform Sampling

Relying solely on Conventional Uniform Sampling strategies can be limited by the long-tail distribution of attributes, as shown in Fig. 20. For instance, the "color" attribute appears with significantly higher frequency, while other attributes occur much less frequently [20]. To address this issue, we introduce the Attribute Uniform Sampling algorithm. Specifically, we first uniformly select a particular value of a given attribute, and then identify corresponding items to form combinations. This approach effectively increases the occurrence of less common keys and introduces long-tail scenarios, thereby enhancing data diversity. As illustrated in Fig. 20, after implementing this sampling method, the attribute distribution becomes more balanced, and a wider range of attributes emerges.

### A.3.3  High-Similarity Product Prioritization Sampling

Due to the vast size of the product catalog, we do not process all qualifying products at once, but instead employ sampling, which may result in the retrieved products not truly matching the specified conditions. To address this issue, we introduce the High-similarity Product Prioritization Sampling algorithm, which identifies a product and one of its similar products (manually pre-annotated).

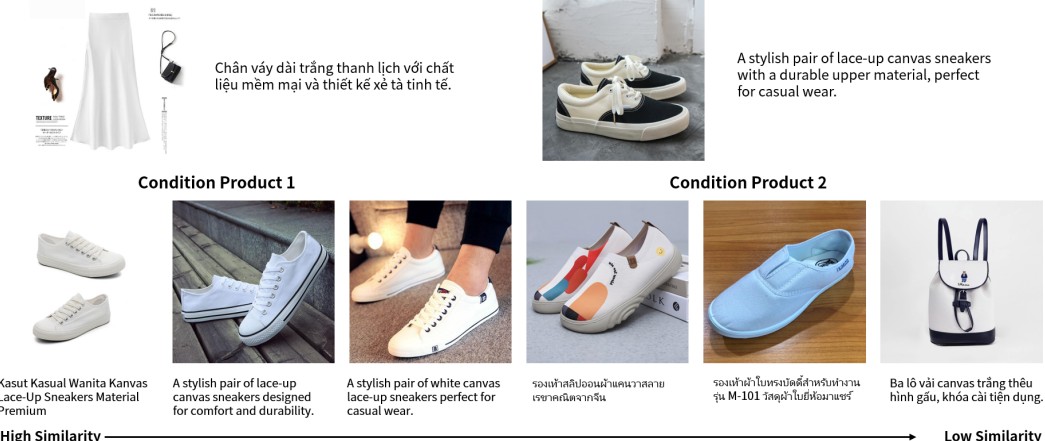

Figure 18: **The CLIP Similarity Ranking of Recalled Products Case 2.** The first row is the two-condition products, and the second row is the recalled products. The CLIP similarity with the condition products decreases from left to right.

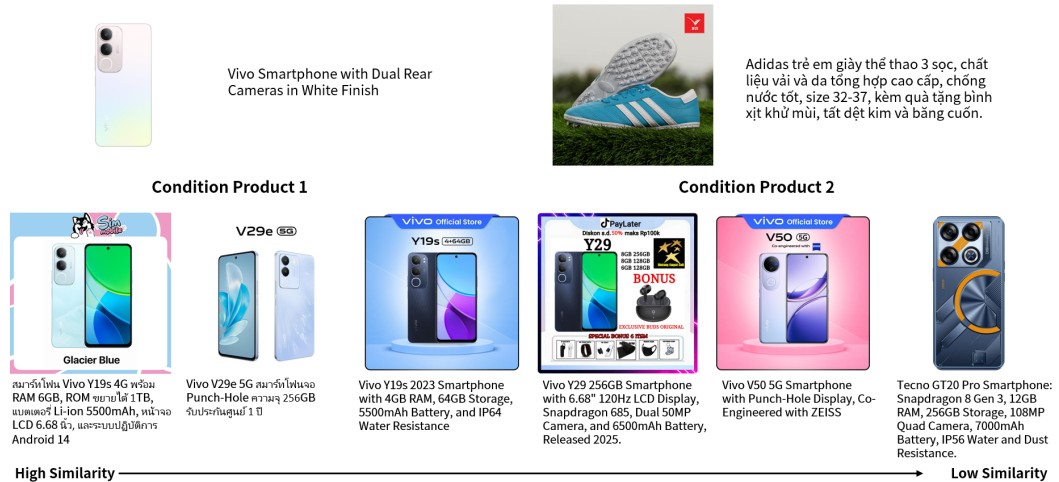

Figure 19: **The CLIP Similarity Ranking of Recalled Products Case 3.** The first row is the two-condition products, and the second row is the recalled products. The CLIP similarity with the condition products decreases from left to right.

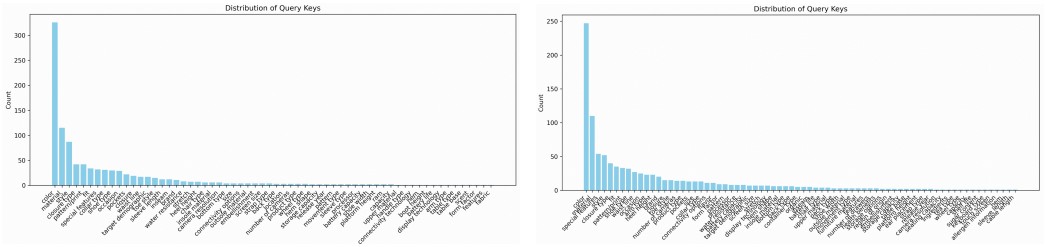

Figure 20: Frequency distribution of conditional attributes before and after applying the Attribute Uniform Sampling strategy. The left panel shows the distribution before applying the Attribute Uniform Sampling algorithm, while the right panel demonstrates the distribution after implementing 10 Attribute Uniform Sampling. Notably, the attribute distribution becomes more balanced after applying the Attribute Uniform Sampling algorithm.

The algorithm then derives several conditions that would transform product 1 into product 2, and

subsequently retrieves candidate products from the catalog based on these conditions. Finally, human annotators perform additional filtering to construct the retrieval query.

Additionally, the High-similarity Product Prioritization Sampling algorithm accounts for long-tail distributions by probabilistically reducing the frequency of the most common attributes (such as color-based differences between product pairs).

### A.3.4 Cold start expansion

Furthermore, our pipeline demonstrates extensibility; upon establishing a comprehensive attribute list at scale, we can perform cold-start initialization for previously unseen categories of data. Specifically, we leverage LLMs for automatic expansion through the following systematic methodology.

First, for any given product, we employ similarity matching approaches [49, 65, 68, 19] to retrieve analogous products as query conditions and corresponding target images. Subsequently, we utilize the following LLM prompt to characterize the distinctions between the two products. Where { all_attribute } represents the distinguishing attributes previously identified from our retrieval dataset, thereby enabling newly integrated data to seamlessly adapt to the existing retrieval paradigm.

Task: Identify the Most Significant Attribute Difference Between Two Products

Instructions:
1. Examine the two product images carefully.
2. From the provided attribute list, select ONLY ONE attribute that represents the most significant difference between these two products. By modifying this attribute, you can convert picture 1 to picture 2
3. Even though multiple differences may exist, focus on identifying the single most visually distinct and important attribute difference. Try not to answer vague attributes like style.
4. Your response should contain ONLY the attribute name - no explanations, no quotes, no additional text.
Image 1: <image> Image 2: <image>
Available attributes to choose from: { all_attribute }

Remember:
- Your response must be exactly one attribute from the provided list
- Do not add quotation marks or any additional text
- Select the most visually distinctive difference
- If multiple differences exist, choose the most significant one

Response (attribute name only):

Upon obtaining the differential product attributes, analogous statistical analysis of the original retrieval data enables the extraction of the various differential values for each attribute. Given that our preceding data generation methodology prioritizes comprehensive diversity, the differential value options are nearly certain to encompass the most salient distinguishing characteristics within product pairs. Specifically, the annotation process utilizes the following prompt:

Task: Identify the Specific Value of a Differing Attribute Between Two Products

Instructions:
1. Carefully examine the two product images.
2. Focus ONLY on the following differing attribute: {diff_attribute}
3. From the provided list of possible values, select EXACTLY ONE value that best describes the {diff_attribute} that can make the first product (Image 1) into the product (Image 2).
4. Your response must contain ONLY the selected value - no explanations, no quotes, no additional text.
5. If none of the values in the provided list accurately describe the difference, respond with ONLY the word: no
Image 1: <image> Image 2: <image>
Differing attribute: {diff_attribute}
Available values to choose from: {all_values}

Remember:
- Your response must be exactly one value from the provided list
- Do not add quotation marks or any additional text
- Focus exclusively on the {diff_attribute} difference between the products

Response (value only):

---

Task: Generate Enhanced Product Search Queries Based on Visual Comparison

Instructions:
1. Examine both the reference product image (Product 1) and the target product image carefully.
2. Create a natural, concise search query that helps retrieve the target product by:
- Maintaining the original statement structure
- Adding ONE new distinctive attribute unique to the target image
- Including the target product's category/type in your search query

Key Requirements:
- Focus only on visually detectable attributes of the specific product in both images
- Highlight meaningful differences or similarities between products
- Use natural, conversational language
- Incorporate the new attribute seamlessly without explicitly stating "this product has X attribute"
- Preserve all Product tag formatting exactly as shown: <Product 1> |image| </Product 1> and <Product 2>product description</Product 2>

Note: Product 1 will be represented by an image, while Product 2 will be represented by text attributes.

Example:
Product 1 Image: [IMAGE]
Target Image: [IMAGE]
Search Statement: Find a product with the same color as in <Product 1> |image| </Product 1> and the same brand as in <Product 2> a product with brand Nike</Product 2>. Your Return is: Find a T-shirt with the same color as in <Product 1> |image| </Product 1> and the same brand as in <Product 2> a product with brand: Nike</Product 2> with a small logo.

Product 1 Image: <image> Target Image: <image>

Search Statement: Find a product that has the same material as <Product 1> |image| </Product 1> and the same {diff_attr} as <Product 2> a product with {diff_attr}: {diff_value}</Product 2>.

Your Return is:

Upon obtaining the differentiating attributes and their corresponding values, we can retrieve additional products through attribute-based lookup and assemble them together with the initial conditions to

form a complete search query. It should be noted that the example presented above illustrates cases with a single differentiating factor, resulting in query formations with two conditions retrieving one product. We can further prompt the model to identify additional existing differentiating factors (utilizing the aforementioned prompts), thereby expanding the number of conditions and enhancing the precision of both search statements and retrieved samples. Once we have acquired the conditions and target positive sample products, we can employ the following prompt to generate the retrieval instruction through the LLM. As shown in the black block above the text.

Finally, this augmented data, like the data generated by the original pipeline, is fed into the filtering pipeline described in Appendix A.4.

## A.4 Auto and Human Filter

### A.4.1 Auto Filter

To enhance efficiency, we implement an initial LLM filtering stage prior to manual annotation. Specifically, when employing machine-based annotation, we require adherence to the following three fundamental principles:

- **Non-omission requirement**: Essential information must be present in the text. For instance, material attributes must be explicitly mentioned within the textual content when required for retrieval.
- **Vision-centric approach**: Attributes that can be effectively communicated through visual representation should rely on visual information rather than textual description. For example, product color attributes should be primarily identified through visual features rather than written descriptions.
- **Accuracy criterion**: The characteristics of positive samples must correspond precisely with the specified retrieval features. For instance, when color consistency is required as a search criterion, the positive samples must exhibit color attributes that exactly match those of the corresponding conditional products.

This multi-modal evaluation framework ensures that the automated filtering process maintains semantic consistency across both textual and visual dimensions while maximizing retrieval relevance. The LLM prompt employed in our implementation follows structured guidelines that prioritize these three principles, resulting in higher-quality candidate samples for subsequent manual verification. For the LLM implementation, the specific prompt utilized is:

### A.4.2 Human Filter

Following the automated filtering process, all data undergoes rigorous manual refinement to ensure quality and comprehensiveness. This human verification stage consists of the following key components:

1. **Data Curation:** Annotators critically review and eliminate data entries deemed inappropriate for the dataset. Furthermore, they refine imprecise search queries to enhance their accuracy and relevance to the target products. This process ensures that the search statements precisely capture the intended product characteristics and filtering criteria.

2. **False Negative Correction and Positive Sample Expansion:** Recognizing that search queries may retrieve multiple valid products beyond those initially annotated, annotators systematically identify and include all products satisfying the established search criteria as positive samples. This step addresses potential false negatives from the automated process and creates a more comprehensive and representative set of positive examples, thereby improving the overall quality and completeness of the dataset.

The detailed annotation protocol, platform specifications, and comprehensive documentation for this manual verification process are provided in Appendix B. These resources offer complete guidance on annotation standards, quality control measures, and platform-specific instructions for the human curation phase.

# B  Data Annotation Protocol

## B.1  General Guidelines

As previously discussed, there is a significant gap in existing benchmarks [86, 63], which primarily focus on homogeneous retrieval scenarios with predefined formats, limiting themselves to single domains, single languages, or employing only singular conditions. To bridge this gap, our benchmark, MERIT, is designed to provide a comprehensive dataset for interleaved multi-condition semantic retrieval, integrating multilingual understanding with the assessment of composite multi-condition queries across diverse product categories. Our dataset follows the guidelines outlined below for data collection:

- **General Principles:**
  - Annotations must be accurate, consistent, and adhere to a high standard of academic rigor.
  - It covers multiple product categories and languages to mirror real-world applications across Southeast Asia.
  - It incorporates diverse visual contexts and attribute combinations to foster a well-rounded evaluation.
  - It provides robust evaluation settings for deterministic assessments.
- **Specific Instructions:**
  - All retrieval pairs must contain one or more images expressing specific properties.
  - All queries should be available in five languages spanning six Southeast Asian countries.
  - All queries should meet real-world e-commerce search complexity.
  - Queries should not be ambiguous and must be answerable with the products in the dataset.
  - Clearly categorize each query across seven distinct product retrieval scenarios.
  - Annotate all fields, including attribute tags, visual descriptors, and other elements that follow the format requirement.
- **Review Process:** Ensure that every annotation undergoes a peer review to maintain high standards and minimize errors or inaccuracy.

Annotations such as product attributes, language, country, category are also collected, providing detailed examples that demonstrate the semantic retrieval capabilities of the models for further analysis and usage.

## B.2  Data Format and Structure

Detailed examples of annotated retrieval pairs are provided in the guidance to serve as a reference for the annotators.

- **JSON File Format:** The structured JSON format will include fields for product identifiers, attribute types, attribute values, query conditions, target products, language identifiers, and region information.
- **Naming Conventions:**
  - Each collected sample will be stored in a separate JSON file following a standard naming rule: **product_category_{Number}**.json
  - Image Files: **image_{ProductID}_{ImageNum}**.png

## B.3  Quality Control and Validation Protocol

The integrity and reliability of the MERIT dataset are maintained through a comprehensive quality assurance framework that encompasses multilingual expertise and systematic validation procedures. The following protocol has been established and rigorously implemented throughout the data collection process:

- **(1) Product Selection and Authentication Protocol**. All candidate products must (i) look authentic. (ii) demonstrate substantial commercial viability as evidenced by popularity, and (iii) meet or exceed the predetermined aesthetic quality threshold as quantified by established metrics [71, 95]. Each product undergoes verification by authorized annotators before inclusion in the product inventory.

- **(2) Attribute Standardization and Verification Protocol**. Given the inherent insufficiency of attribute richness in commercial e-commerce data, all products must undergo attribute enrichment via our prescribed open annotation methodology. Annotators shall (i) document all observable attributes using a standardized lexicon, (ii) participate in statistical analysis for Attribute Delineation, and (iii) verify attribute consistency across similar products. All attribute assignments must receive secondary validation before entering the final database.

- **(3) Query Formulation Compliance Protocol**. To ensure statistical robustness and ecological validity, every search query must be generated through our tri-modal sampling methodology: (i) Conventional Uniform Sampling, (ii) Attribute Uniform Sampling, and (iii) High-similarity Product Prioritization Sampling, as detailed in Appendices A.3.1-A.3.3. Cold-start expansion procedures (Appendix A.3.4) shall be employed only after primary sampling methods have been exhausted. Each query composition undergoes verification against pre-established diversity metrics.

- **(4) Multi-stage Validation Protocol**. All dataset entries are subject to mandatory dual-phase validation: (i) Automated Validation Phase—employing deterministic rule-based systems and statistical outlier detection methods to identify and eliminate non-conforming samples; followed by (ii) Expert Validation Phase—requiring assessment by annotators with demonstrated proficiency in the relevant languages and product domains. Samples must achieve unanimous approval through both validation phases to MERIT inclusion in the final dataset. Non-compliant samples shall be documented for methodological refinement purposes.

## B.4 Handling Ambiguities

Instances of ambiguity or unclear data are flagged for detailed review. These instances are collaboratively examined during team meetings to establish a standardized approach for annotation. Particular attention is paid to visual attribute interpretation across different cultural and linguistic contexts.

## B.5 Ethical Considerations

- **Copyright and Licensing:** Adherence to copyright and licensing regulations is strictly enforced. Data from sources that prohibit copying or redistribution is explicitly avoided.

- **Data Privacy:** Compliance with privacy laws and ethical standards in data handling is paramount. Annotators must avoid collecting product information that contains any private information.

- **Ethical Data Usage:** All data collection and usage must respect ethical guidelines. This includes avoiding biased or harmful content and ensuring that the datasets promote fairness and inclusivity across diverse cultural contexts.

## B.6 Data Contamination Considerations

The risk of data contamination is mitigated by assigning annotators to carefully select products and attributes that extend beyond straightforward queries with easily accessible answers. It is essential that retrieval tasks rely on provided images for attribute identification rather than the common knowledge of large language models. This approach is beneficial for creating benchmarks that genuinely test the model's ability to comprehend and synthesize information from diverse visual sources across multiple languages and cultural contexts.

## B.7 Annotation Platform

We developed a GUI-based annotation platform [3, 14, 93, 94], as illustrated in Fig. 22, specifically engineered to facilitate the data annotation process for human experts. This system enables specialists

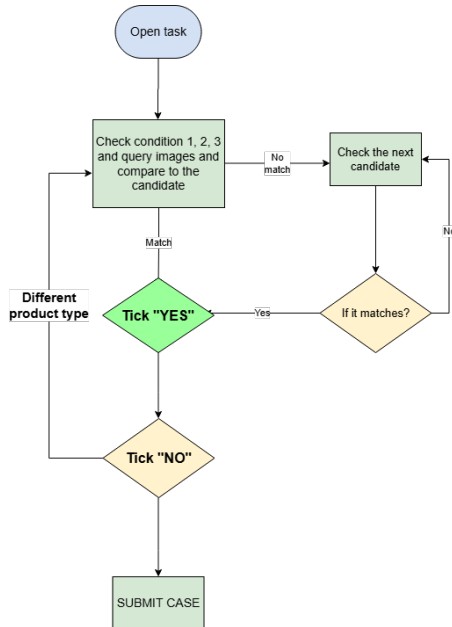

Figure 21: Flowchart of manual review of labeled data.

to efficiently visualize images, while performing annotations and modifications directly within an intuitive interface. The platform's streamlined layout significantly enhances user experience, enabling experts to execute annotation tasks with increased precision and efficiency, thus elevating both the quality and productivity of the annotation process. The primary objective of this tool is to optimize the complex annotation workflow, minimize manual intervention, and substantially improve the overall efficiency of data annotation procedures.

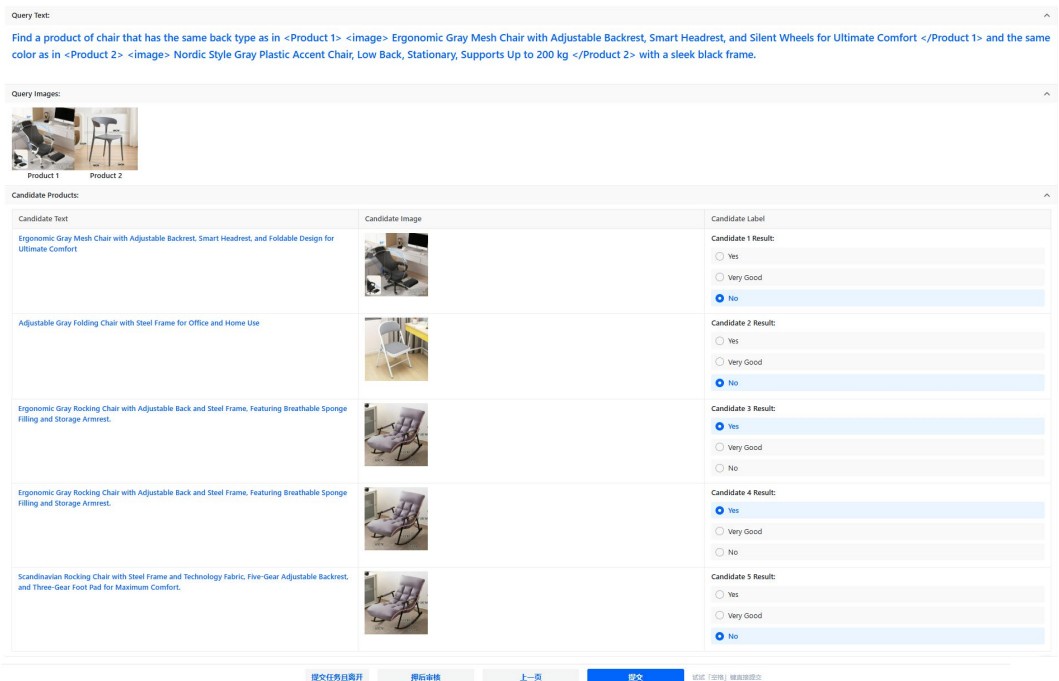

Figure 22: Annotation Platform.

## B.8 Data Preparation and Release

For evaluation purposes, we selected $10,000$ queries from a total of $320,000$ queries in MERIT as the test set. To ensure equitable representation of each source dataset within the test split and maintain distributional consistency of language and product categories between the test set and the complete dataset, we implemented a stratified sampling methodology:

1. Random sampling of queries to achieve proportional representation of languages and product categories in alignment with the full dataset distribution.

2. Supplementary random sampling of remaining queries from individual source datasets according to their respective volumetric contributions to the complete corpus.

After dividing the dataset into training and test sets, the test set underwent two additional rounds of manual annotation to further enhance quality. Our initial pre-annotation test set consisted of approximately $20,000$ queries, which was subsequently refined to $10,000$ queries through careful curation.

## C More Dataset Analysis

### C.1 Global Statics

Tab. 4 delineates the taxonomic structure and distribution of product categories within the MERIT dataset. The hierarchical organization comprises seven primary product types (Food, Clothing, Electronics, Bags, Jewelry, Furniture, and Others), further subdivided into specific secondary types to enable fine-grained analysis. This comprehensive categorization encompasses essential consumer products ranging from perishables such as fruits and beverages to durable items, including electronics and furniture. The diverse product spectrum reflects real-world product retrieval inventory diversity, providing a robust foundation for evaluating retrieval models across various product domains. With approximately 160,000 total products, MERIT represents one of the most extensive multimodal semantic retrieval benchmarks in the literature, facilitating meaningful performance assessments across different product categories and their associated attributes.

Tab. 23 and Tab. 24 present the geographical distribution of samples within the MERIT dataset across six Southeast Asian countries. Indonesia contributes the largest portion with X samples, reflecting its significant market presence in the region. The Philippines follows with 569 samples, while Thailand, Malaysia, and Vietnam contribute X, and X samples respectively. Singapore, with its smaller market size but strategic importance, accounts for X samples. This diverse geographical representation ensures that the benchmark captures the linguistic and cultural nuances essential for evaluating retrieval systems in a multilingual Southeast Asian product retrieval context.

| Language | Number |
|----------|--------|
| ID | 2281 |
| EN | 597 |
| TH | 366 |
| MS | 284 |
| VN | 268 |

Figure 23: Language distribution.

| Country | Number |
|---------|--------|
| Indonesia | 2281 |
| Philippines | 569 |
| Thailand | 366 |
| Malaysia | 284 |
| Vietnam | 268 |
| Singapore | 28 |

Figure 24: Country distribution.

Fig. 25 illustrates the cross-linguistic patterns in the retrieval process within the MERIT dataset. This Sankey diagram visualizes the flow between source query languages and target product languages, demonstrating the prevalence of both intra-linguistic searches (where source and target languages match) and cross-linguistic retrieval scenarios. The diagram reveals that Indonesian (id) serves as the predominant source language, with significant flows to other Southeast Asian languages, including Thai (th), Vietnamese (vi), and Malay (ms), as well as English (en). The thickness of each connection represents the frequency of language transitions, highlighting the multilingual nature of product retrieval interactions in the region. This visualization underscores the importance of robust cross-lingual retrieval capabilities in real-world applications, particularly in linguistically diverse markets.

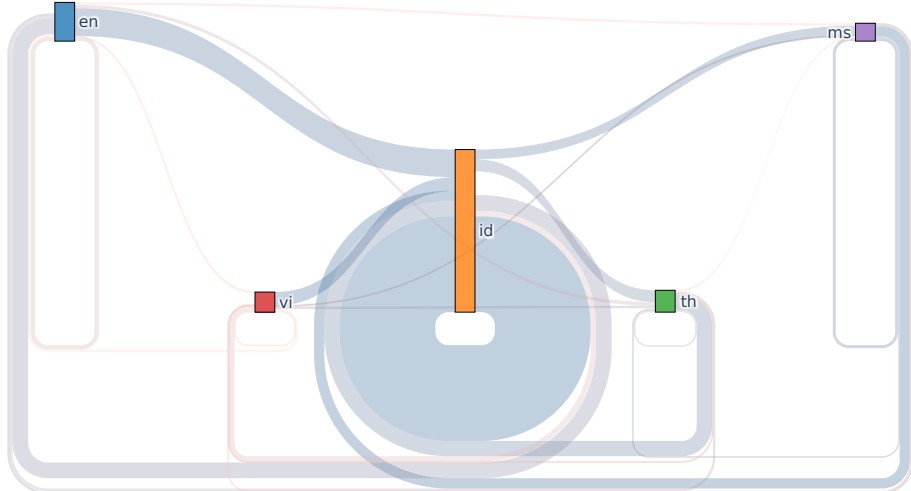

Figure 25: **Language Flow Diagram.** The flow chart shows the flow of conditional products to target products.

Fig. 26 presents the geographic distribution of retrieval patterns across six Southeast Asian countries within the MERIT benchmark. The Sankey visualization maps the flow of queries originating from one country to products associated with potentially different countries. Indonesia (ID) emerges as the primary source of queries, with substantial connections to other regional markets including the Philippines (PH), Thailand (TH), Malaysia (MY), and Vietnam (VN). The diagram illustrates both intra-national retrieval (where source and target countries match) and cross-border retrieval scenarios that reflect the interconnected nature of product retrieval in Southeast Asia. The varying widths of the flow connections quantify the frequency of these cross-market interactions, providing valuable insights into regional commerce patterns and highlighting the necessity for retrieval systems to effectively operate across national boundaries.

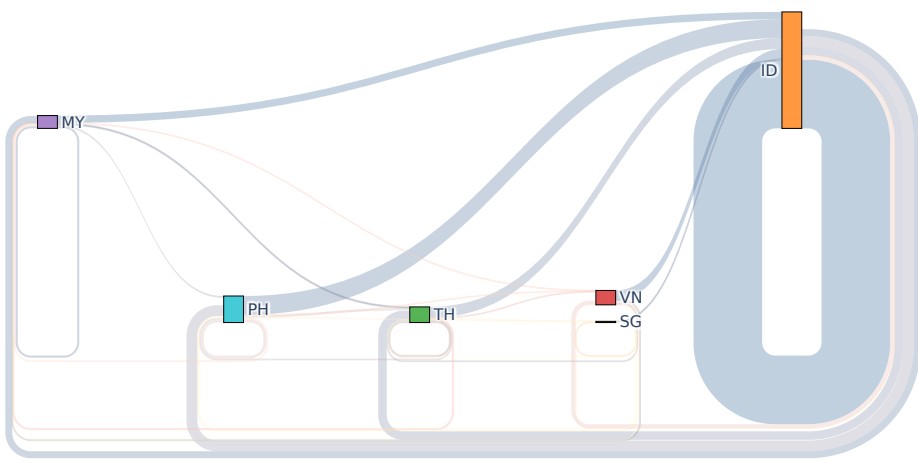

Figure 26: **Country Flow Diagram.** The flow chart shows the flow of conditional products to target products.

Fig. 27 delineates the inter-category retrieval dynamics within the MERIT dataset through a comprehensive Sankey diagram. This visualization maps the relationships between query source product classes and retrieved target product classes, revealing both intra-category retrieval (where queries and results belong to the same product category) and cross-category retrieval scenarios. The diagram

demonstrates how users searching within one product domain (*e.g.*, clothing) may retrieve items not only within that same category but also from related categories (*e.g.*, accessories or footwear). The varying thickness of connecting flows represents the frequency of these category transitions, providing quantitative insights into product category relationships. This visualization is particularly valuable for understanding the complex cross-categorical nature of product retrieval search behavior and for developing retrieval systems capable of accommodating diverse user intentions that span multiple product domains.

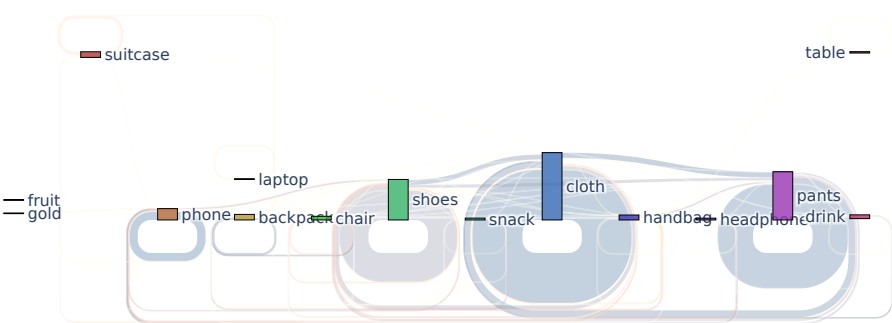

Figure 27: **Product Class Type Flow Diagram.** The flow chart shows the flow of conditional products to target products.

Fig. 28 presents a comprehensive visualization of product distribution across linguistic and geographical dimensions in the MERIT dataset. The left sunburst chart illustrates the distribution by language, with the inner ring representing five languages (id, en, th, ms, vi) and the outer ring depicting the corresponding product categories. Indonesian (id) dominates the linguistic landscape, constituting the largest portion, followed by English (en), Thai (th), Malay (ms), and Vietnamese (vi). The right sunburst chart demonstrates the geographical distribution, with the inner ring representing six Southeast Asian countries (ID, PH, TH, MY, VN) and the outer ring showing product categories within each country. Indonesia (ID) accounts for the most substantial proportion, followed by the Philippines (PH), Thailand (TH), Malaysia (MY), and Vietnam (VN). Across both dimensions, clothing items (particularly pants and clothes) and electronics (predominantly phones) emerge as the most prevalent product categories. This dual visualization effectively captures the multilingual and multicultural nature of the dataset, highlighting the proportional distribution of product types across different languages and markets in Southeast Asia.

## C.2    Attribute and Product

**Attribute Distribution**. Fig. 29 presents a bar chart of the top 30 most frequent attributes within the MERIT dataset. The chart, ordered by descending frequency, highlights the predominant attribute categories such as 'color' and 'material', providing a quantitative overview of attribute distribution. This visualization aids in understanding the key characteristics emphasized in the dataset.

**Attribute Word Cloud**. Fig. 30 displays a word cloud of the top 30 attributes from the MERIT dataset. Larger font sizes indicate higher frequency, with 'color' and 'material' standing out, offering a visual summary of the most common attributes. This representation enhances the perception of attribute prominence within the benchmark.

**Value Distribution**. Fig. 31 illustrates a bar chart of the top 30 most frequent values associated with attributes in the MERIT dataset. Ordered by frequency, it showcases dominant values like 'gray' and 'pink', providing a detailed view of value distribution. This chart supports the analysis of specific attribute values prevalent in the dataset.

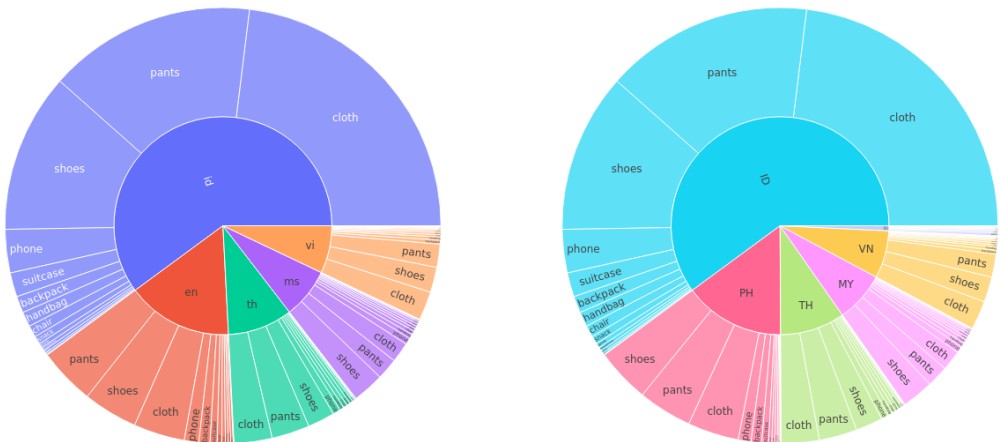

Figure 28: **Sunburst visualization of product distribution in MERIT.** The left chart shows the hierarchical relationship between languages (inner ring) and product categories (outer ring), while the right chart illustrates the distribution between countries (inner ring) and product categories (outer ring).

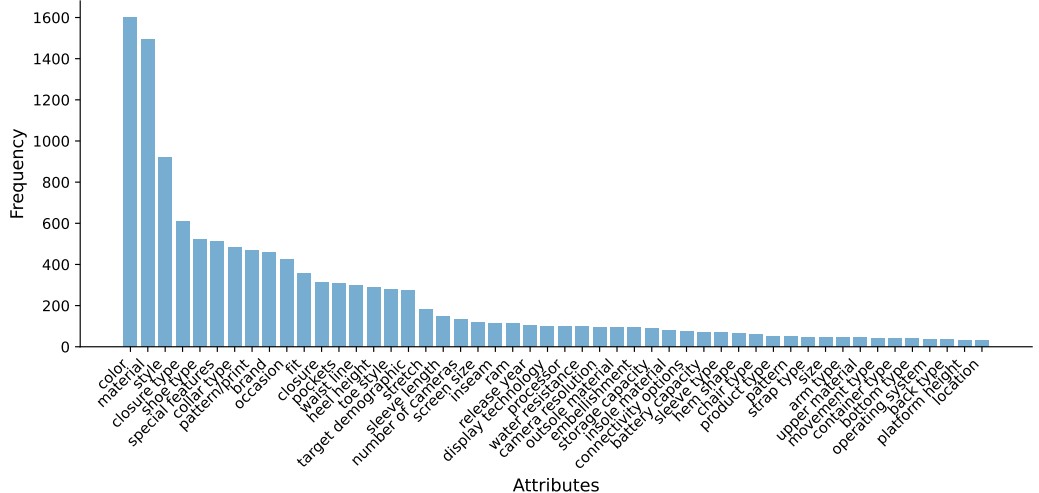

Figure 29: A bar chart of the top 30 most frequent attributes in MERIT, ordered by frequency, highlighting key attribute categories.

**Value Word Cloud**. Fig. 32 presents a word cloud of the top 30 values from the MERIT dataset. Larger font sizes reflect higher frequency, with 'gray' and 'pink' being prominent, offering a visual insight into the most common attribute values. This visualization facilitates the exploration of value diversity within the benchmark.

**Product Word Cloud** Fig. 33 presents a word cloud visualization of the 'query instruction' strings from the MERIT dataset. The word cloud highlights the most frequent terms, with larger font sizes indicating higher frequency, providing a clear overview of the key themes and vocabulary used in user queries. This visualization underscores the diversity and focus of query instructions within the benchmark, offering insights into user search behavior.

**Title** Fig. 34 displays a word cloud representation of the product titles within the MERIT dataset. The visualization emphasizes frequently occurring words with larger font sizes, revealing common

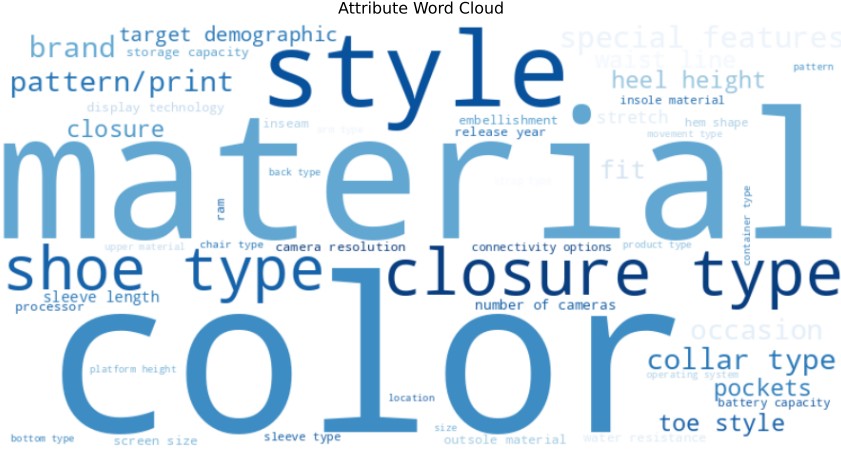

Figure 30: A word cloud of the top 30 attributes in MERIT, with larger fonts indicating higher frequency.

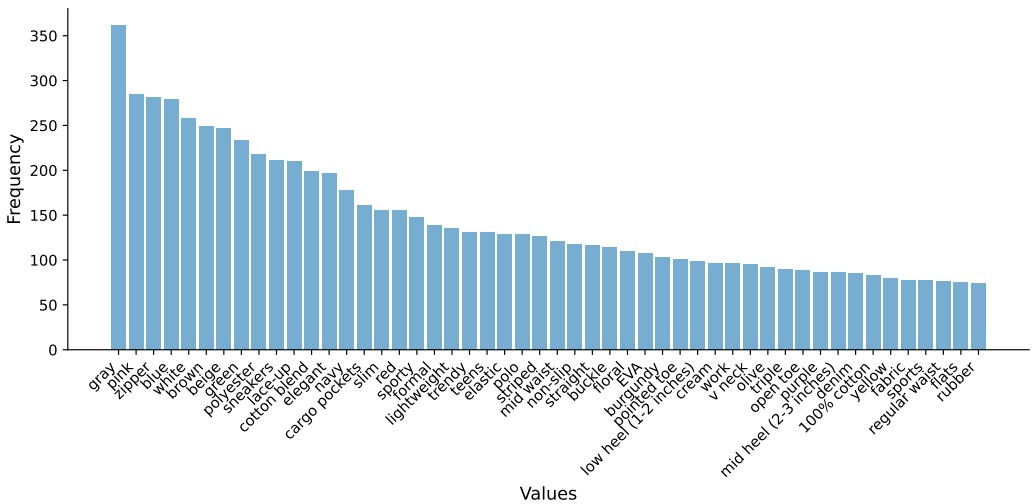

Figure 31: A bar chart of the top 30 most frequent values in MERIT, ordered by frequency, highlighting dominant attribute values.

descriptors and product characteristics in product retrieval titles. This word cloud provides a snapshot of the linguistic patterns and descriptive focus of product titles, shedding light on the nature of product representations in the benchmark.

**Product Length** Fig. 35 illustrates the distribution of character counts in product titles within MERIT. The visualization reveals significant variation in title lengths across the dataset, reflecting the diverse nature of e-commerce product descriptions. For visualization simplicity, titles exceeding 190 characters in length have been consolidated into a single category. This distribution provides insight into the complexity and descriptive depth of product representations within the benchmark.

**Query Length** Fig. 36 presents the distribution of character counts in query instructions across MERIT. The distribution demonstrates considerable variation in query complexity, reflecting the multi-condition and multilingual nature of the benchmark. For visualization clarity, query instructions exceeding 500 characters have been aggregated into a single category. This analysis provides valuable context regarding the linguistic complexity of retrieval instructions that models must process to accurately identify relevant products.

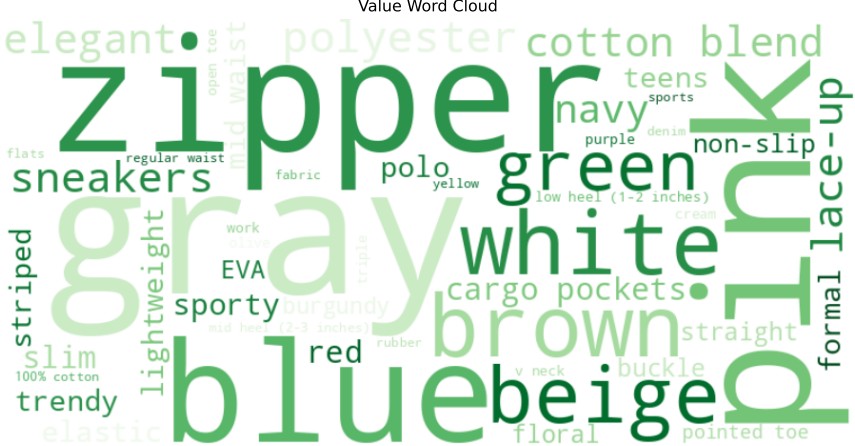

Figure 32: A word cloud of the top 30 values in MERIT, with larger fonts indicating higher frequency.

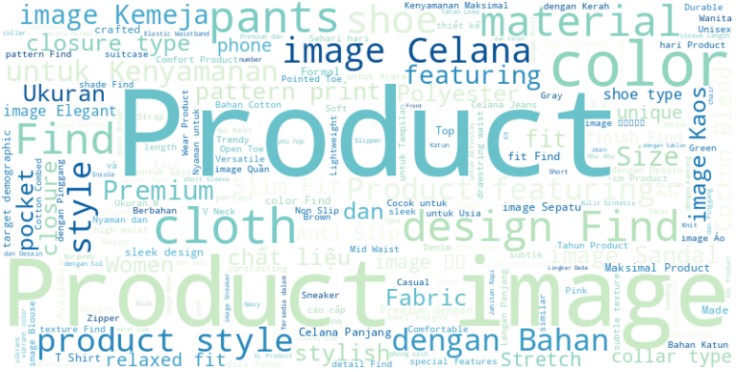

Figure 33: A word cloud of the 'query instruction' strings in MERIT. Larger font sizes represent higher frequency of terms, illustrating the key themes in user queries.

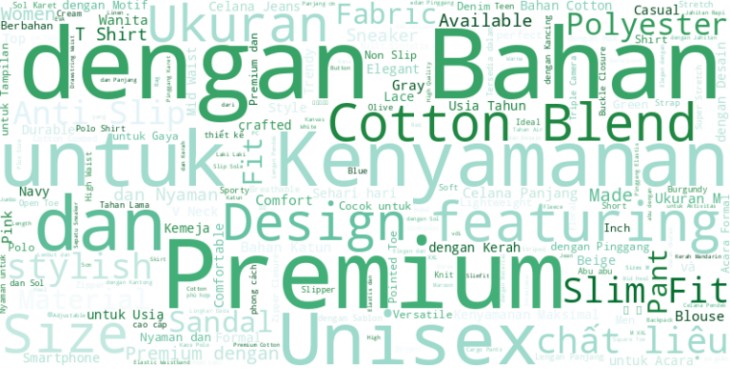

Figure 34: A word cloud of the product titles in MERIT. Larger font sizes indicate a higher frequency of terms, highlighting common descriptors in product titles.

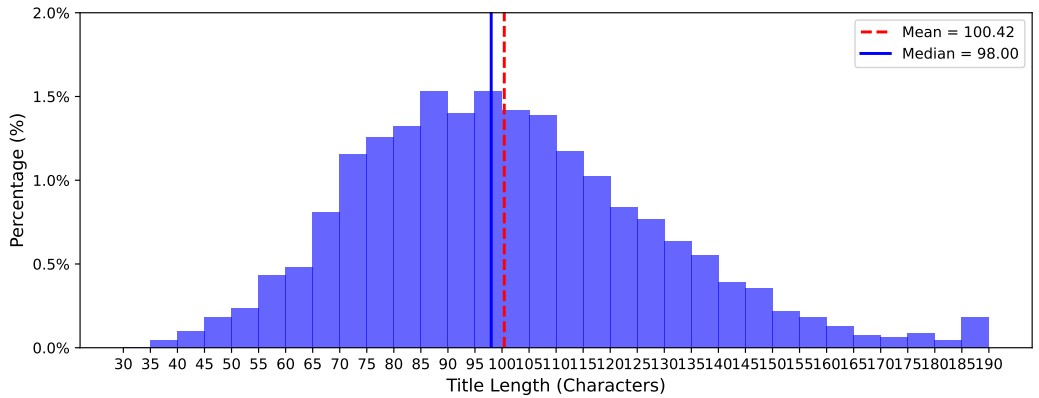

Figure 35: The distribution of the string length of product title in MERIT. Titles with a length greater than 190 are categorized as 190 for visualization simplicity.

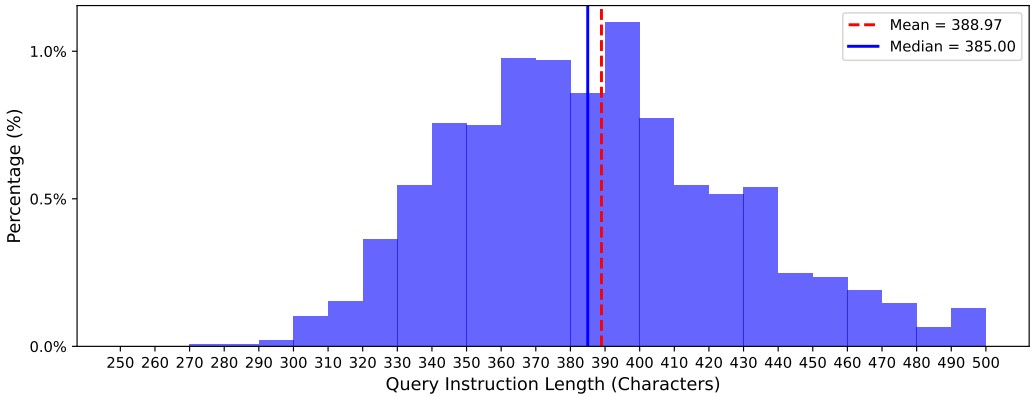

Figure 36: The distribution of the string length of query instruction in MERIT. Query instructions with a length greater than 500 are categorized as 500 for visualization simplicity.

# D Dataset Examples

Table 5: Table index of case study figures by meta-task with associated error categories. If there are more than 2 conditions, we only show the first 2.

| Case Figure | Attribute I | Attribute II | Target Product Type | Target Product Language |
|---|---|---|---|---|
| Figure 37 | Style | Color | Pants | Vietnamese |
| Figure 38 | Toe Style | Color | Shoes | Thai |
| Figure 39 | Heel Height | Color | Shoes | Thai |
| Figure 40 | Design Style | Size | Handbag | Thai |
| Figure 41 | Brand | Fashion Style | Backpack | English |
| Figure 42 | Insole Material | Color | Shoes | Indonesian |
| Figure 43 | Material | Color | Pants | Vietnamese |
| Figure 44 | Tightness | Pattern | Pants | Thai |
| Figure 45 | Movement Type | Color | Chair | English |
| Figure 46 | Fit | Color | Pants | Thai |
| Figure 47 | Closure | Color | Shoes | Thai |
| Figure 48 | Color | Material | Cloth | English |
| Figure 49 | Color | Material | Pants | English |
| Figure 50 | Brand | Color | Headphone | English |
| Figure 51 | Wheel Type | Color | Suitcase | Thai |
| Figure 52 | Camera | Color | Phone | English |
| Figure 53 | Neck shape | Color | Cloth | Indonesian |
| Figure 54 | Material | Pattern | Pants | Thai |
| Figure 55 | Collar type | Color | Cloth | Thai |
| Figure 56 | Embellishment | Sleeve type | Cloth | Thai |
| Figure 57 | Brand | Storage capacity | Phone | Thai |
| Figure 58 | Material | Closure type | Shoes | English |
| Figure 59 | Occasion | Material | Pants | English |
| Figure 60 | Lace style | Color | Shoes | Thai |
| Figure 61 | Brand | Color | Headphones | Thai |
| Figure 62 | Pockets | Color | Pants | English |
| Figure 63 | Capacity | Brand | Suitcase | English |
| Figure 64 | Color | Shoe type | Shoes | English |
| Figure 65 | Shoe type | Closure type | Shoes | English |

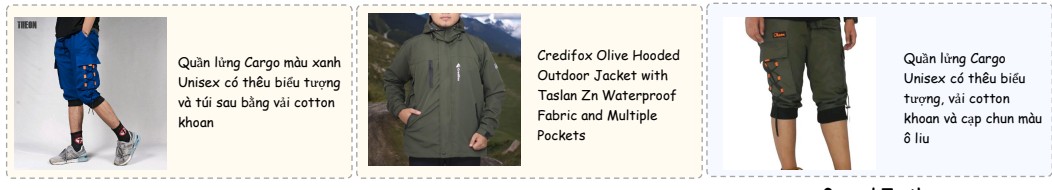

**Query Text:**

Find a product of pants that have the same product style with Product 1 and the same color with Product 2 with a visible brand logo.

Quần lửng Cargo màu xanh Unisex có thêu biểu tượng và túi sau bằng vải cotton khoan

Credifox Olive Hooded Outdoor Jacket with Taslan Zn Waterproof Fabric and Multiple Pockets

Quần lửng Cargo Unisex có thêu biểu tượng, vải cotton khoan và cạp chun màu ô liu

Condition 1  Condition 2  Ground Truth

Figure 37: A sample case. Back to List of Figures.

**Query Text:**

Find a product of shoes that have the same toe style with Product 1 and the same color with Product 2with a glossy finish.

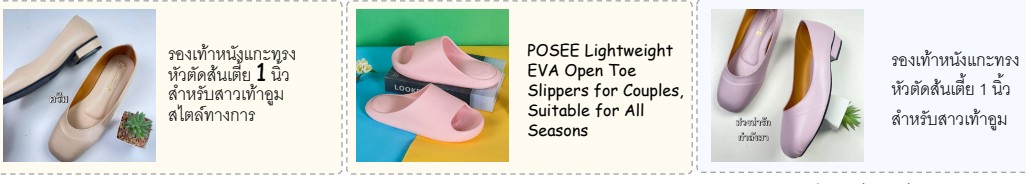

Figure 38: A sample case. Back to List of Figures.

**Query Text:**

Find a product of shoes that have the same heel height with Product 1 and the same color with Product 2 with a sleek metallic finish.

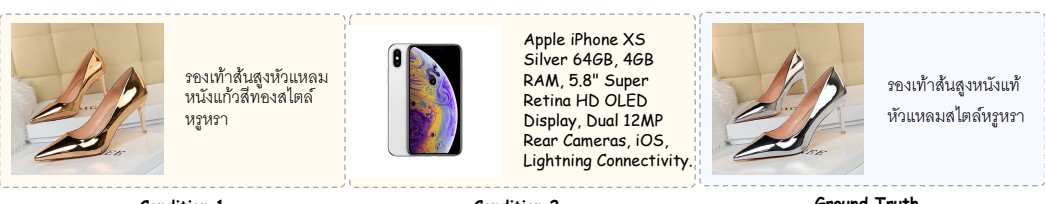

Figure 39: A sample case. Back to List of Figures.

**Query Text:**

Find a product of handbag that has the same design style with Product 1 and the same color with Product 2 with a denim texture.

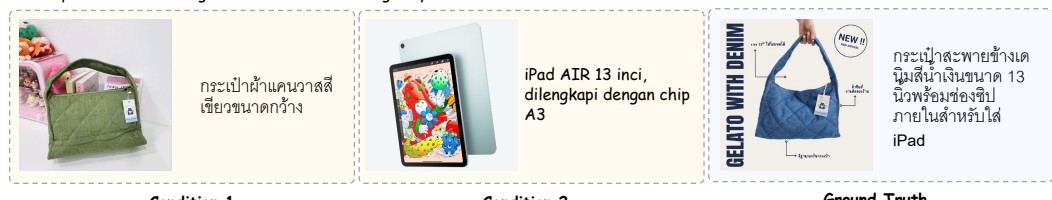

Figure 40: A sample case. Back to List of Figures.

**Query Text:**

Find a product of backpack that have the same brand with and the same fashion style with a quilted texture and a chain strap.

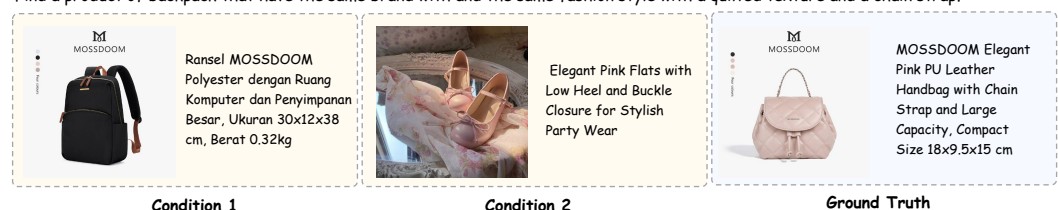

Figure 41: A sample case. Back to List of Figures.

**Query Text:**

Find a product of shoes that have the same insole material with Product 1 and the same color with Product 2 with a sleek, minimalist design.

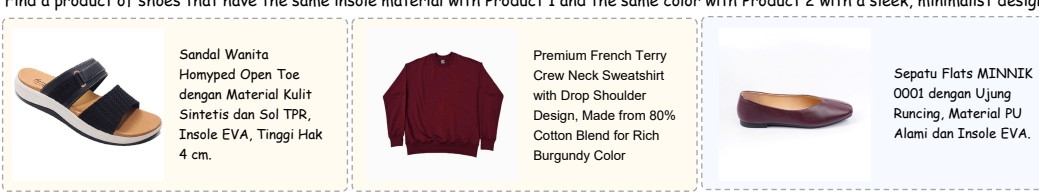

Figure 42: A sample case. Back to List of Figures.

**Query Text:**

Find a product of pants that have the same material with Product 1 and the same color with Product 2 with a subtle logo on the side.

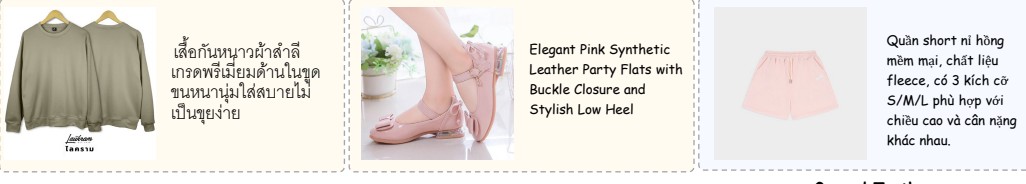

Condition 1      Condition 2      Ground Truth

Figure 43: A sample case. Back to List of Figures.

**Query Text:**

Find a product of pants that have the same tightness with Product 1 and the same pattern with Product 2.

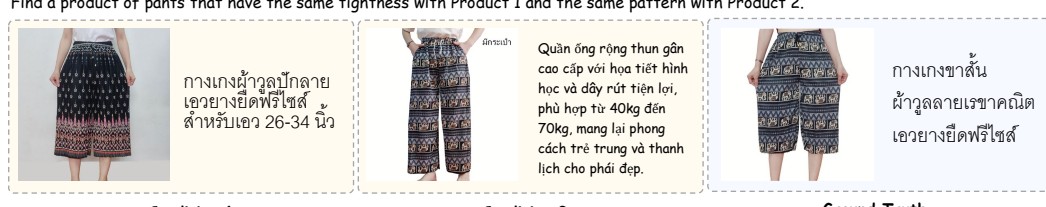

Condition 1      Condition 2      Ground Truth

Figure 44: A sample case. Back to List of Figures.

**Query Text:**

Find a product of chair that has the same movement type as in Product 1 and the same color as in WProduct 2 with a plush texture.

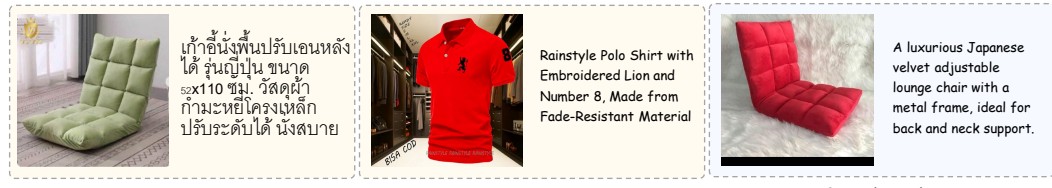

Condition 1      Condition 2      Ground Truth

Figure 45: A sample case. Back to List of Figures.

**Query Text:**

Find a product of pants that have the same fit with Product 1 and the same color with Product 2, featuring a distinct brand label.

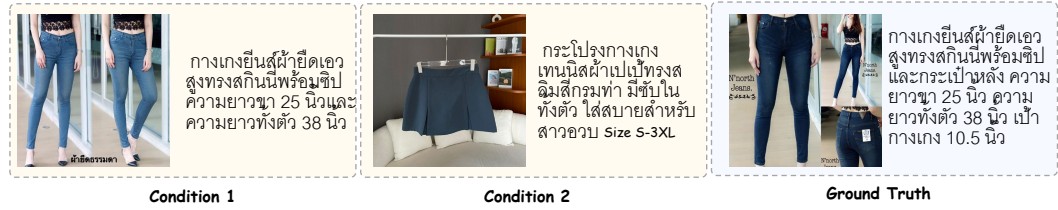

Condition 1      Condition 2      Ground Truth

Figure 46: A sample case. Back to List of Figures.

**Query Text:**

Find a product of shoes that have the same closure type with Product 1 and the same color with Product 2 with a contrasting sole.

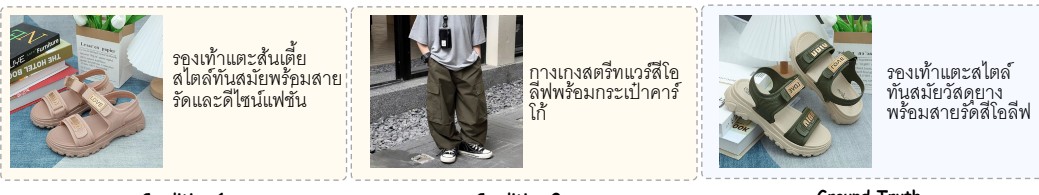

Condition 1      Condition 2      Ground Truth

Figure 47: A sample case. Back to List of Figures.

**Query Text:**

Find a product of cloth that has the same color as Product 1 and the same material as Product 2 with a button-up design.

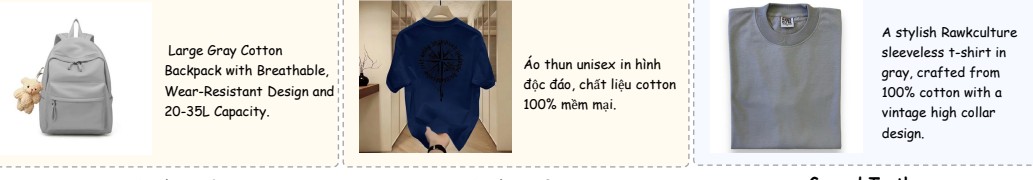

**Condition 1**  **Condition 2**  **Ground Truth**

Figure 48: A sample case. Back to List of Figures.

**Query Text:**

Find a product of pants that have the same color with Product 1 and the same material with Product 2, featuring a tailored fit and a sleek design.

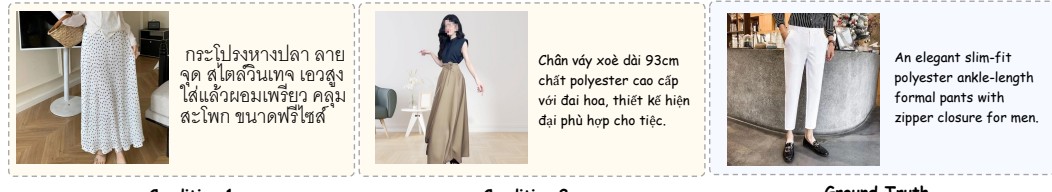

**Condition 1**  **Condition 2**  **Ground Truth**

Figure 49: A sample case. Back to List of Figures.

**Query Text:**

Find a product of headphone that have the same brand with Product 1 and the same color with Product 2 with a matching case.

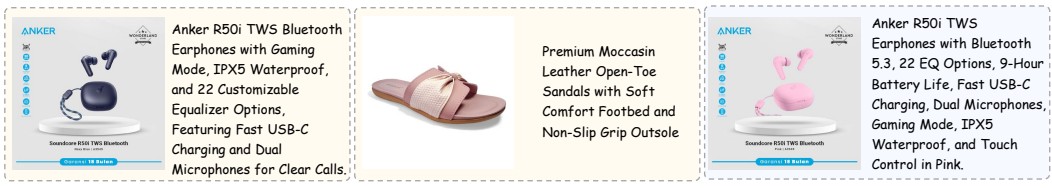

**Condition 1**  **Condition 2**  **Ground Truth**

Figure 50: A sample case. Back to List of Figures.

**Query Text:**

Find a product of suitcase that has the same wheel type as in Product 1 and the same color as in Product 2 with a larger size and a sleek handle design.

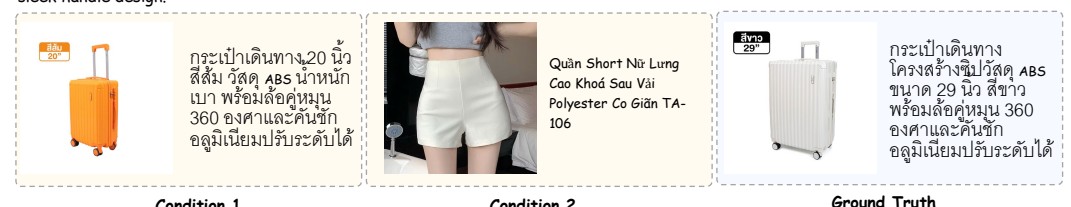

**Condition 1**  **Condition 2**  **Ground Truth**

Figure 51: A sample case. Back to List of Figures.

**Query Text:**

Find a product of phone that has the same camera as Product 1 and the same color as Product 2 with a sleeker design.

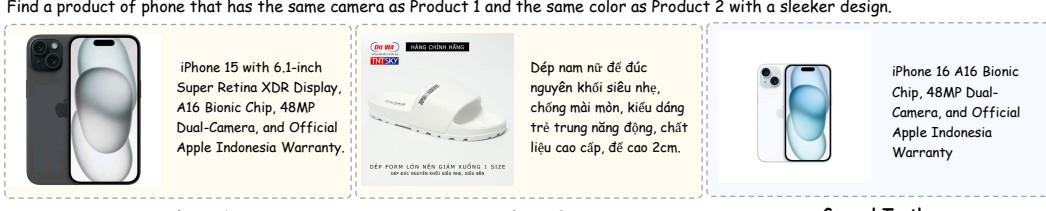

**Condition 1**  **Condition 2**  **Ground Truth**

Figure 52: A sample case. Back to List of Figures.

**Query Text:**

Find a product of cloth that has the same neck shape as Product 1 and the same color as Product 2 with a tie detail.

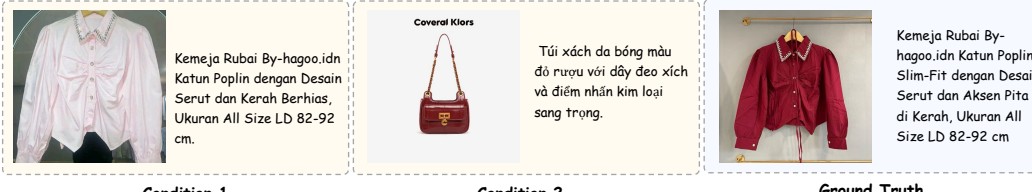

Figure 53: A sample case. Back to List of Figures.

**Query Text:**

Find a product of pants that have the same material with Product 1 and the same pattern/print with Product 2.

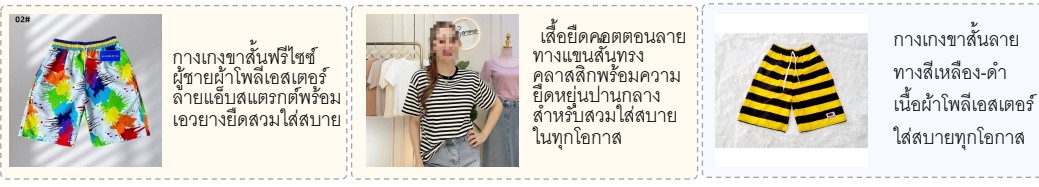

Figure 54: A sample case. Back to List of Figures.

**Query Text:**

Find a product of cloth that has the same collar type as in Product 1 and the same color as in Product 2 with intricate sleeve detailing..

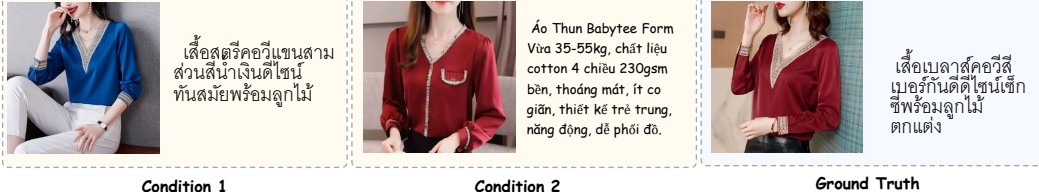

Figure 55: A sample case. Back to List of Figures.

**Query Text:**

Find a product of cloth that has the same embellishment as in Product 1 and the same sleeve type as in Product 2 with a light color and layered design.

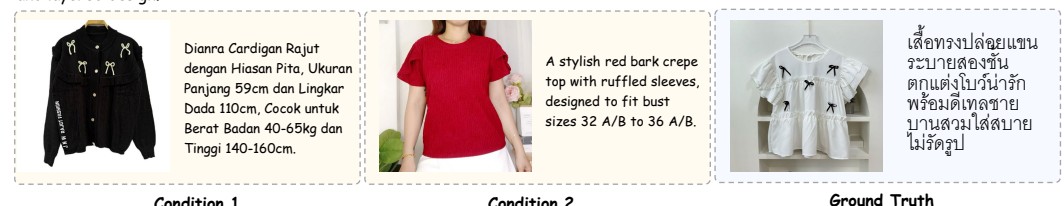

Figure 56: A sample case. Back to List of Figures.

**Query Text:**

Find a product of phone that has the same brand as in Product 1 and the same storage capacity as in Product 2 with a vibrant color finish.

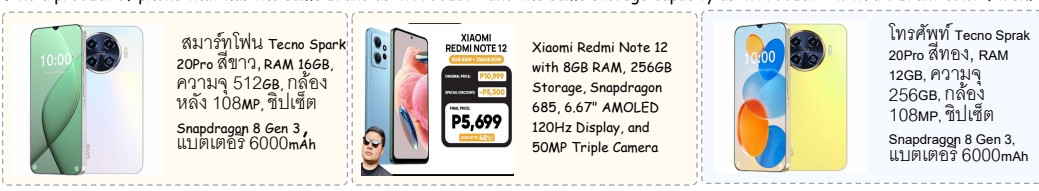

Figure 57: A sample case. Back to List of Figures.

**Query Text:**

Find a product of shoes that have the same material with Product 1 and the same closure type with Product 2 featuring a prominent logo and reinforced toe design.

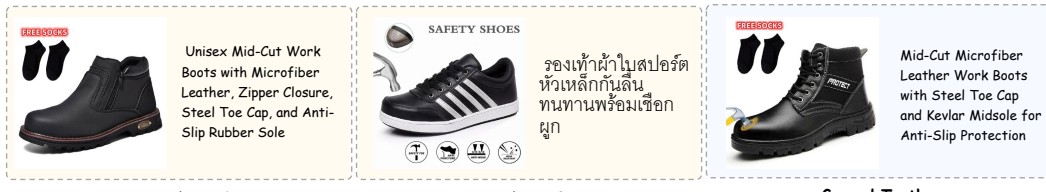

Figure 58: A sample case. Back to List of Figures.

**Query Text:**

Find a product of pants that have the same occasion with Product 1 and the same material with Product 2 in a neutral color.

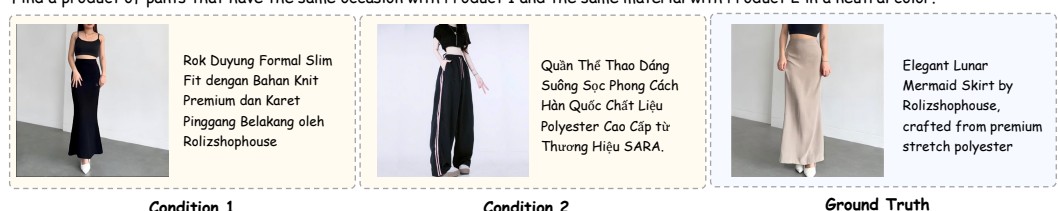

Figure 59: A sample case. Back to List of Figures.

**Query Text:**

Find a product of shoes that have the same lace style with Product 1, the same color with Product 2 and the same heel color with Product 3.

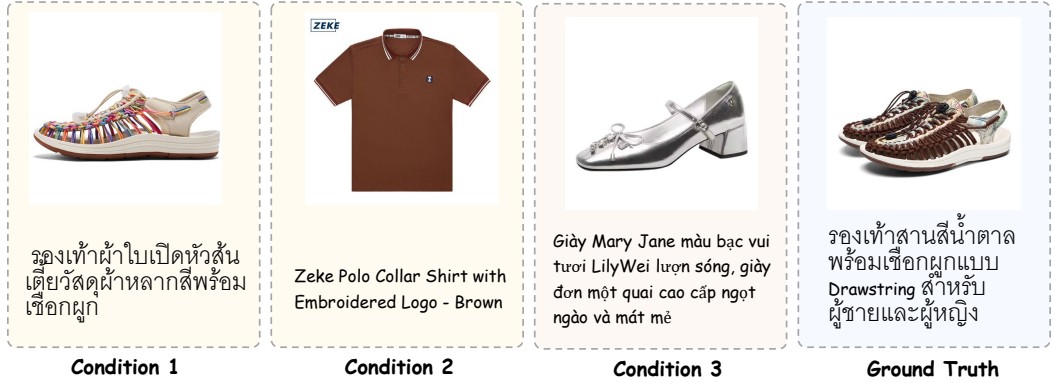

Figure 60: A sample case. Back to List of Figures.

**Query Text:**

Find a product of headphones that have the same brand as Product 1 and the same color as Product 2 with a transparent casing design.

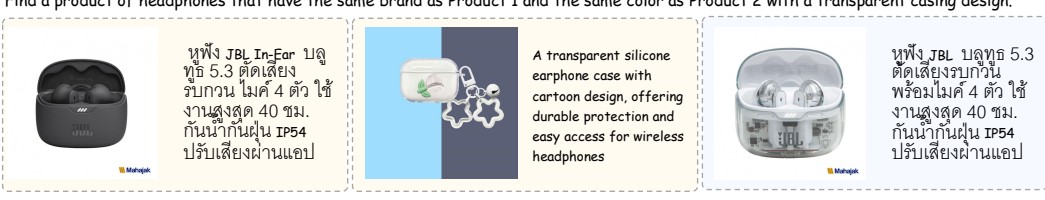

Figure 61: A sample case. Back to List of Figures.

**Query Text:**

Find a product of pants that have the same pockets with Product 1 and the same color with Product 2 with a slightly darker shade.

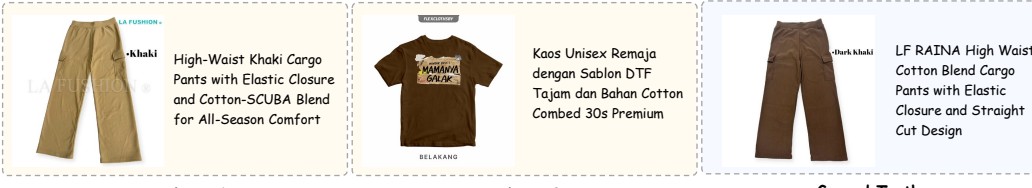

Figure 62: A sample case. 

**Query Text:**

Find a product of suitcase that has the same capacity as Product1, same brand as Product 2 and the same color as Product 3 with a sleek, minimalist design.

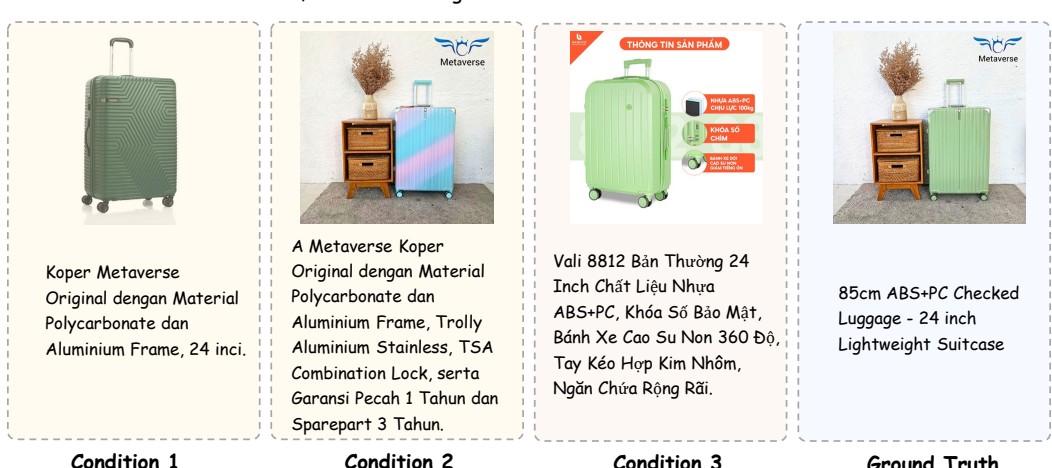

Figure 63: A sample case. 

**Query Text:**

Find a product of shoes that have the same color with Product 1 and the same shoe type with Product 2 featuring decorative elements.

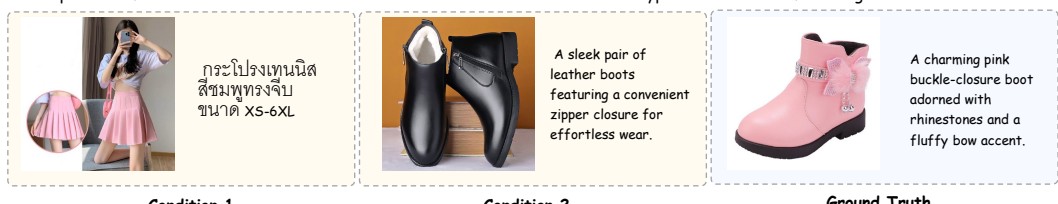

Figure 64: A sample case. 

**Query Text:**

Find a product of shoes that have the same shoe type with Product 1 and the same closure type with Product 2 with a sleek design and a low heel.

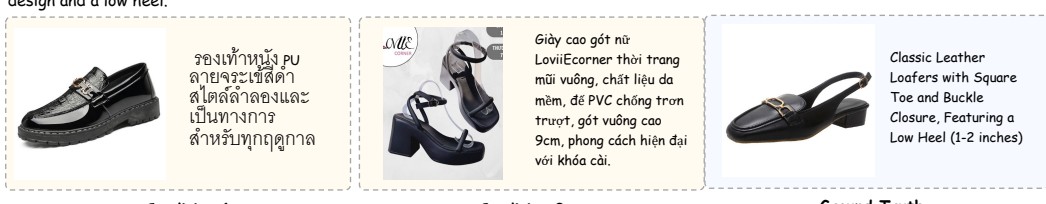

Figure 65: A sample case. 

# E   Experiments Details

## E.1   Baselines

For most MLLM-based models, we adhered to the standard evaluation protocol delineated in their respective original configurations. In instances where such protocols were not specified, we followed the established conventions outlined in VLMEvalKit [15, 69], consistently setting the temperature parameter to 0. Specifically, we categorized the evaluated models into two distinct types: **Zero-Shot MLLMs**, which have not undergone training on dedicated retrieval datasets, and **Embedding MLLMs**, which have been specifically trained on existing retrieval datasets through particular methodologies for retrieval purposes.

**Zero-Shot MLLM:**

**InternVL2.5-VL [12]** is an advanced multimodal large language model (MLLM) series that builds upon InternVL 2.0, maintaining its core model architecture while introducing significant enhancements in training and testing strategies as well as data quality. Through extensive evaluations on a wide range of benchmarks, including multi-discipline reasoning, document understanding, interleaved multi-image understanding, real-world comprehension, multimodal hallucination detection, visual grounding, multilingual capabilities, and pure language processing, InternVL 2.5 exhibits competitive performance, rivaling leading commercial models such as GPT-4o and Claude-3.5-Sonnet. Notably, InternVL 2.5 is the first open-source MLLM to achieve over 70% on the MMMU benchmark. For our experiments, we utilized the 1B variant, as well as the version enhanced through MPO training. Both models were evaluated with the maximum number of tiles constrained to one.

**Qwen2.5-VL [5].** Qwen2.5-VL represents the latest flagship model in the Qwen vision-language series, achieving significant advancements in foundational multimodal capabilities including enhanced visual recognition, precise object localization via bounding boxes, robust document parsing, and long-video comprehension with second-level event localization. The model's innovative architecture incorporates dynamic resolution processing and absolute time encoding, enabling it to natively perceive spatial scales and temporal dynamics without traditional normalization techniques, while its native dynamic-resolution Vision Transformer with Window Attention [57] maintains resolution integrity with reduced computational overhead. For our experiments, we employed the 3B parameter variant of this model, configuring the max_pixels parameter to $576 \times 28 \times 28$.

**Embedding MLLM:**

**E5-V [38].** E5-V adapts Multimodal Large Language Models (MLLMs) to generate universal multimodal embeddings, effectively bridging the modality gap between different input types. Employing an innovative single-modality training approach using exclusively text pairs, E5-V demonstrates superior performance compared to traditional multimodal training methods while reducing training costs by approximately 95% and eliminating the need for expensive multimodal data collection. Following the original paper, we obtain the global token through the prompt 'Summary above image in one word: '.

**LLaVE [41].** LLaVE represents a groundbreaking framework for universal multimodal embeddings that effectively addresses the challenge of distinguishing hard negative pairs in image-text retrieval tasks through dynamic representation learning based on discriminative difficulty. We evaluated three model sizes: 0.5B, 2B, and 7B parameter versions.

**GME-Qwen2VL [102].** General Multimodal Embedder (GME) functions as an MLLM-based dense retriever capable of processing queries and candidates across text, images, or multimodal combinations. Developed using a novel training data synthesis pipeline that addresses modality imbalance issues, GME overcomes the limitations of previous approaches that relied solely on text data for training. In our experiments, we utilized the GME implementation based on Qwen2VL-2B.

**LamRA-Qwen2.5VL [56].** LamRA is a versatile framework that repurposes MLLMs for comprehensive retrieval tasks, eliminating the need for task-specific fine-tuning. Employing a two-stage training methodology—language-only pre-training followed by multimodal instruction tuning—and joint training for both pointwise and listwise reranking, LamRA demonstrates exceptional performance across more than ten retrieval tasks in both supervised and zero-shot settings. Our evaluation utilized the LamRA implementation based on Qwen2.5VL-7B.

**BGE-VL [104].** We evaluated the BGE-VL-MLLM-S1 model, which was obtained from `https://huggingface.co/BAAI/BGE-VL-base`. BGE-VL-MLLM-S1 is trained exclusively on the Mega-Pairs dataset, achieving outstanding performance in composed image retrieval tasks.

**VLM2Vec [39].** VLM2Vec is a novel multimodal embedding framework designed to encode sequences of images and text into a unified representation space for diverse downstream applications. Unlike conventional CLIP or BLIP embeddings that operate under constraints of fixed image resolutions and text lengths, VLM2Vec accommodates inputs of arbitrary dimensions and sequence lengths, significantly enhancing its versatility across multimodal tasks. Our experiments utilized the 4B parameter version of VLM2Vec.

## E.2 Other Dataset Usage

In this section, we provide a detailed introduction to the additional datasets utilized in our paper. To validate the effectiveness of our proposed method, CORAL, we conducted comprehensive evaluations on several widely-used retrieval benchmarks.

**VisDial [16]:** This dataset presents interactive visual dialogues generated through a controlled collaboration between two annotators on Amazon Mechanical Turk. In this conversational framework, one participant assumes the role of the "questioner," with access only to a textual caption of an image, while the other serves as the "answerer," with complete visual access to the image itself. They engage in structured 10-round question-and-answer exchanges regarding image content. We repurpose this dialogically rich dataset as a cross-modal retrieval task, where the objective is to identify and retrieve the precise image that corresponds to the given conversational dialogue, thereby testing models' abilities to construct visual representations from textual discourse.

**CIRR [59]:** This dataset is meticulously designed for composed image retrieval tasks, focusing on natural language modifications of visual content. It comprises pairs of real-world reference and target images, accompanied by linguistically nuanced modification descriptions that articulate the transformative differences between the source and target images. This configuration presents a particularly challenging evaluation scenario that requires models to understand both visual foundations and linguistic modifications in a compositional manner.

**VisualNews [53]:** This corpus encompasses a substantial collection of publicly available news images paired with professionally written captions. The dataset features content from major news organizations and represents a diverse range of visual journalism across various domains, providing a real-world test of multimodal understanding in the context of current events and factual reporting. The caption-image pairs exhibit complex relationships that often require world knowledge and contextual understanding of news content.

**MSCOCO [52]:** This comprehensive benchmark dataset features over 330,000 images, each meticulously annotated with multiple human-generated captions. Originally designed for object detection, segmentation, and captioning tasks, MSCOCO has become a fundamental standard for evaluating multimodal capabilities. Its diversity spans 91 object categories captured in everyday contexts with multiple objects per image. The dataset's rich annotations facilitate cross-modal retrieval in both text-to-image and image-to-text directions, providing a robust assessment of bidirectional understanding between visual and linguistic modalities.

**NIGHTS [22]:** This dataset introduces perceptually calibrated human similarity judgments on image pairs that exhibit diverse forms of visual correspondences. The corpus consists of carefully constructed triplets: a reference image and two systematically perturbed variations, accompanied by human perceptual judgments indicating which variation maintains greater similarity to the reference. Following the methodology established in M-BEIR [84], we reconfigure this dataset into a retrieval framework for image-to-image matching, where the reference image functions as the query, and the perturbed version that aligns with human perceptual judgment serves as the target. This transformation provides a rigorous test of a model's ability to replicate human perceptual similarity assessments.

**WebQA [7]:** This dataset presents a multihop, multimodal question-answering framework that necessitates the retrieval and integration of information from Wikipedia pages to formulate responses to given queries. The dataset's complexity emerges from its requirement for models to navigate both textual and visual information across multiple reasoning steps. In our experimental context, we utilize the Wikipedia page's images and accompanying textual descriptions as candidate elements

Table 6: Comparision with other method across 8 retrieval tasks.

| | EN | ID | TH | VN | MS |
|---|---|---|---|---|---|
| Qwen2.5-VL [5] | 48.73 | 56.23 | 55.34 | 55.13 | 47.98 |
| InternVL2.5-VL [12] | 55.38 | 60.75 | 58.97 | 62.82 | 52.47 |
| BGE-VL [104] | 13.76 | 14.02 | 14.83 | 20.51 | 14.08 |

for retrieval, thereby evaluating models' capacities to identify relevant multimodal content based on query specifications.

### E.3 Main Experiments Settings

All experiments were conducted on a computing node equipped with $8\times$H100 GPUs.

Experiments were conducted for a single epoch with the following training configuration: A per-device batch size of 4 was employed with gradient accumulation steps set to 2, resulting in an effective global batch size of 64. The InfoNCE contrastive loss temperature parameter ($\tau$) was fixed at $0.02$. For negative sampling, we implemented in-batch negatives combined with cross-device negative sample gathering, achieving a final positive-to-negative ratio of $1:63$.

For full-parameter fine-tuning, we adopted a learning rate of $1e-5$ with weight decay of $0.0005$ and linear warmup ratio of $0.01$. The LoRA [28] configuration employed the following parameters: learning rate of $1e-4$ (10 times higher than full fine-tuning), identical weight decay ($0.0005$) and warmup ratio ($0.01$), with LoRA-specific hyperparameters set to $r=8, \alpha=16$, no bias terms, and a dropout rate of $0.05$ between LoRA layers.

The CORAL framework was configured with the following hyperparameters: the loss weighting coefficients $\lambda_1$ and$\lambda_2$ were both set to 0.1 to balance the objective components, while maintaining uniform masking probabilities of $0.5$ for both visual and linguistic modalities. This symmetric configuration ensures equal contribution from both vision and language streams during the masked reconstruction tasks.

Across all training regimes, we kept the vision tower completely frozen to preserve its pretrained representations. For LoRA-based adaptation, we specifically applied low-rank adaptation only to the LLM backbone components, while maintaining standard full-parameter training for all BERT decoder layers. This hybrid approach allowed us to efficiently adapt the language model while preserving the decoder's complete expressive capacity.

### E.4 More Experiments Results

**Different Languages' Performance** In Section 3.4, we present the accuracy distribution across different languages, with specific values shown in Table 6. We define a query as belonging to a particular language only when all products in both the query statement and the positive samples are in that language.

**Out of Distribution Scenarios**. As mentioned in Sec. 3.3, we tested several out-of-distribution (OOD) scenarios. Tables 8, 7, and 9 correspond to the Zero-shot, OOD, and Mixed results depicted in Figure 6(b), respectively. Zero-shot refers to direct inference using Qwen2.5-VL's `[EOS]` token. OOD indicates that for each row in Table 7, we excluded the specified OOD data (e.g., for the first row concerning Language ID OOD, we removed all queries containing the ID language from the training set, while for the test set, we only selected data where all languages were ID). Mixed refers to training conducted on the complete training dataset.

**Results on 8 Retrieval Tasks**. To further validate the efficacy of CORAL, we conducted evaluations on several retrieval datasets. The experimental results are presented in Tab. 10, with experimental configurations following the methodology described in [39].

### E.5 Error Analysis Details

In this section, we present a case study analysis of the error types made by Qwen2.5-VL-3B [5] and GME-Qwen2VL-2B [102] across various tasks. The errors are classified into the following five categories. Other less frequent error types are not included in this analysis. For the analysis, as

Table 7: **Out-of-distribution test results for Qwen2.5-VL trained on in-domain data**. AVG represents the average value, and # indicates the total number of entries in the test set.

| | | # | MRR | R@1 | R@5 | R@10 |
|---|---|---|---|---|---|---|
| Language | ID | 513 | 44.26 | 15.98 | 79.34 | 87.91 |
| | MS | 434 | 42.82 | 17.97 | 74.65 | 84.33 |
| | TH | 1020 | 45.00 | 19.80 | 75.29 | 84.22 |
| | AVG | 1967 | 44.03 | 17.92 | 76.43 | 85.49 |
| Attribute | Brand | 889 | 60.72 | 48.03 | 78.07 | 85.60 |
| | Pattern | 1002 | 54.93 | 37.23 | 77.74 | 84.53 |
| | Region | 89 | 44.29 | 19.10 | 78.65 | 86.52 |
| | AVG | 1980 | 53.31 | 34.79 | 78.15 | 85.55 |
| Class | Drink | 475 | 51.14 | 33.47 | 73.47 | 81.05 |
| | Phone | 1958 | 43.47 | 31.77 | 58.63 | 67.72 |
| | Table | 422 | 48.57 | 35.07 | 64.69 | 74.64 |
| | AVG | 2855 | 47.73 | 33.44 | 65.60 | 74.47 |

Table 8: **Out-of-distribution test results for zero-shot Qwen2.5-VL**. AVG represents the average value, and # indicates the total number of entries in the test set.

| | | # | MRR | R@1 | R@5 | R@10 |
|---|---|---|---|---|---|---|
| Language | ID | 513 | 4.28 | 2.34 | 7.02 | 9.55 |
| | MS | 434 | 6.47 | 3.92 | 9.22 | 12.44 |
| | TH | 1020 | 2.67 | 0.69 | 5.20 | 7.94 |
| | AVG | 1967 | 4.47 | 2.32 | 7.15 | 9.98 |
| Attribute | Brand | 889 | 5.46 | 2.81 | 8.44 | 13.05 |
| | Pattern | 1002 | 0.95 | 0.50 | 1.50 | 2.40 |
| | Region | 89 | 7.46 | 4.49 | 12.36 | 14.61 |
| | AVG | 1980 | 4.62 | 2.60 | 7.43 | 10.02 |
| Class | Drink | 475 | 6.68 | 3.58 | 10.95 | 13.89 |
| | Phone | 1958 | 2.18 | 1.02 | 3.47 | 5.72 |
| | Table | 422 | 2.82 | 1.18 | 4.98 | 7.35 |
| | AVG | 2855 | 3.89 | 1.93 | 6.47 | 8.99 |

mentioned in Sec. 3.4, we selected 500 samples for each model, but due to space limitations, we present only some of them here, as shown in the following Figures.

**Attribute Error**: Retrieval models often misidentify or incorrectly select attributes in product retrieval tasks, resulting in recommendations that fail to meet the specified attribute criteria. As illustrated in Fig. 66, while the recalled product matches the pattern of Product 1, its color does not align with the requirement of Product 2 (which shares the same condition). This discrepancy highlights the model's misinterpretation of attribute-based constraints.

**Visual Understanding Error**: Retrieval models accurately identify the requested attributes but fail to generate corresponding and consistent visual outputs. In such cases, while the model correctly interprets the textual instructions, it struggles to align them with appropriate visual representations. As illustrated in Fig. 67, although the language command is comprehended correctly, the retrieved product image is incorrect [82].

**Category Error**: Retrieval models retrieve products from incorrect categories, indicating a failure to accurately classify items within the prescribed product taxonomy. As illustrated in Fig. 68, the model is tasked with identifying a mobile phone bag but instead retrieves a generic bag, demonstrating a misclassification.

**Detail Error**: Retrieval models accurately identify the primary product conditions but often overlook specific secondary requirements or finer details. Consequently, their recommendations satisfy the main criteria but fail to capture critical nuances specified in the query. As illustrated in Fig. 69,

Table 9: **Out-of-distribution test results for Qwen2.5-VL trained on the whole train set of queries**. AVG represents the average value, and # indicates the total number of entries in the test set.

|  |  | # | MRR | R@1 | R@5 | R@10 |
|---|---|---|---|---|---|---|
| Language | ID | 513 | 71.34 | 57.31 | 88.89 | 94.93 |
|  | MS | 434 | 67.51 | 53.23 | 85.48 | 91.94 |
|  | TH | 1020 | 69.79 | 55.29 | 87.65 | 91.57 |
|  | AVG | 1967 | 69.55 | 55.28 | 87.34 | 92.81 |
| Attribute | Brand | 889 | 71.87 | 59.62 | 88.98 | 93.59 |
|  | Pattern | 1002 | 63.50 | 50.30 | 80.44 | 87.03 |
|  | Region | 89 | 57.81 | 38.20 | 83.15 | 93.26 |
|  | AVG | 1980 | 64.39 | 49.37 | 84.19 | 91.29 |
| Class | Drink | 475 | 53.54 | 33.26 | 78.53 | 86.74 |
|  | Phone | 1958 | 63.64 | 51.58 | 79.78 | 86.57 |
|  | Table | 422 | 52.40 | 38.63 | 69.19 | 77.96 |
|  | AVG | 2855 | 56.53 | 41.16 | 75.83 | 83.76 |

Table 10: Comparision with other method across 8 retrieval tasks.

| Model | VisDial | CIRR | VisualNews$_{T2I}$ | VisualNews$_{I2T}$ | COCO$_{T2I}$ | COCO$_{I2T}$ | NIGHTS | WebQA |
|---|---|---|---|---|---|---|---|---|
| CLIP | 30.70 | 12.60 | **78.90** | **79.60** | 59.50 | 57.70 | 60.40 | 67.50 |
| OpenCLIP | 25.40 | 15.40 | 74.00 | 78.00 | 63.60 | 62.10 | **66.10** | 62.10 |
| SigLIP | 21.50 | 15.10 | 51.00 | 52.40 | 58.30 | 55.00 | 62.90 | 58.10 |
| BLIP2 | 18.00 | 9.80 | 48.10 | 13.50 | 53.70 | 20.30 | 56.50 | 55.40 |
| MagicLens | 24.80 | 39.10 | 50.70 | 21.10 | 54.10 | 40.00 | 58.10 | 43.00 |
| E5-V | 9.20 | 6.10 | 13.50 | 8.10 | 20.70 | 14.00 | 4.20 | 17.70 |
| GME-Qwen2-VL-2B | 26.00 | 38.00 | 66.00 | 71.00 | 62.00 | 56.00 | 64.00 | 83.00 |
| Qwen2-VL-2B | 13.00 | 20.00 | 40.00 | 43.00 | 49.00 | 39.00 | 59.00 | 20.00 |
| Qwen2-VL-2B-CL | 51.00 | 39.00 | 56.00 | 52.00 | 56.00 | 45.00 | 58.00 | 67.00 |
| Qwen2-VL-2B+CORAL | **73.00** | **50.00** | 67.00 | 72.00 | **68.00** | **64.00** | 65.00 | **84.00** |

**Query Text:**

Find a T-shirt that has the same text pattern as Product 1 and the same color as Product 2.

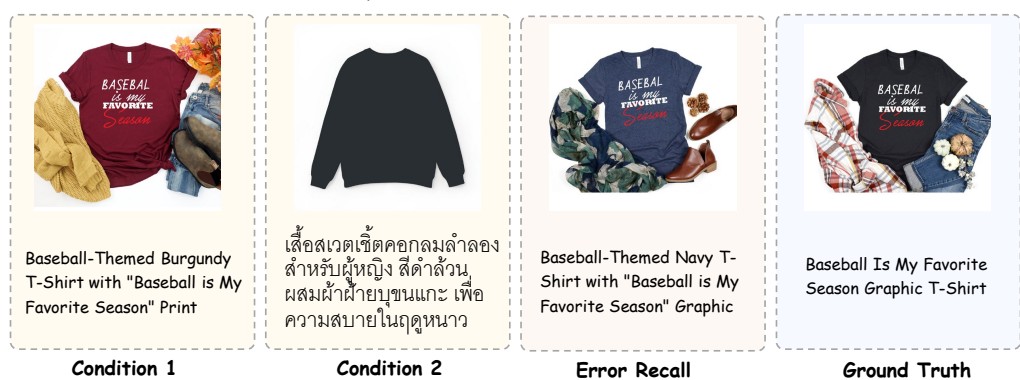

| Baseball-Themed Burgundy T-Shirt with "Baseball is My Favorite Season" Print | เสื้อสเวตเชิ้ตคอกลมลำลอง สำหรับผู้หญิง สีดำล้วน ผสมผ้าฝ้ายบุขนแกะ เพื่อ ความสบายในฤดูหนาว | Baseball-Themed Navy T-Shirt with "Baseball is My Favorite Season" Graphic | Baseball Is My Favorite Season Graphic T-Shirt |
|---|---|---|---|
| **Condition 1** | **Condition 2** | **Error Recall** | **Ground Truth** |

Figure 66: A case for Attribute Error. In this example, the same condition as product 2 is color. The recalled product meets the pattern of product 1, but the color does not meet the requirements of product 2. This is a misunderstanding of the attribute content.

although the retrieved product fulfills the two primary conditions, it does not meet the detailed requirement of "having a visibly displayed brand logo."

**Annotation Error**: Inaccuracies in dataset annotations may result in cases where the model's response appears incorrect when evaluated against imprecise ground truth. Such errors are rare, as our data undergoes multiple rounds of manual review. However, given the vast scale of the candidate set, we cannot entirely rule out the possibility of a small number of positive samples remaining unlabeled.

**Query Text:**

Find a T-shirt that has the same material as Product 1 and Product 2 .

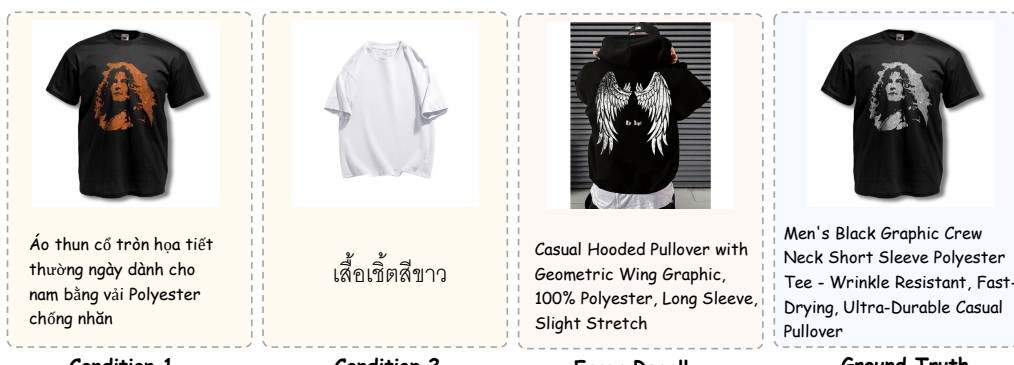

Figure 67: A case for Visual Understanding Error. In this example, the language instruction was understood correctly, but the visual image of the recalled product was wrong.

**Query Text:**

Find a phone bag that has the same material as Product 1 and the same color as Product 2

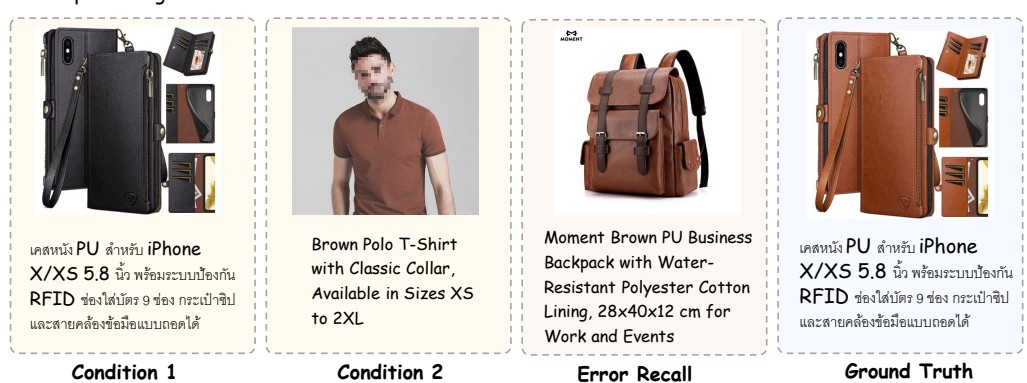

Figure 68: A case for Category Error. In this example, the goal is to find a mobile phone bag, but a bag is recalled, so it is wrong.

**Query Text:**

Find a product of pants that have the same product style with Product 1 and the same color with Product 2 with a visible brand logo.

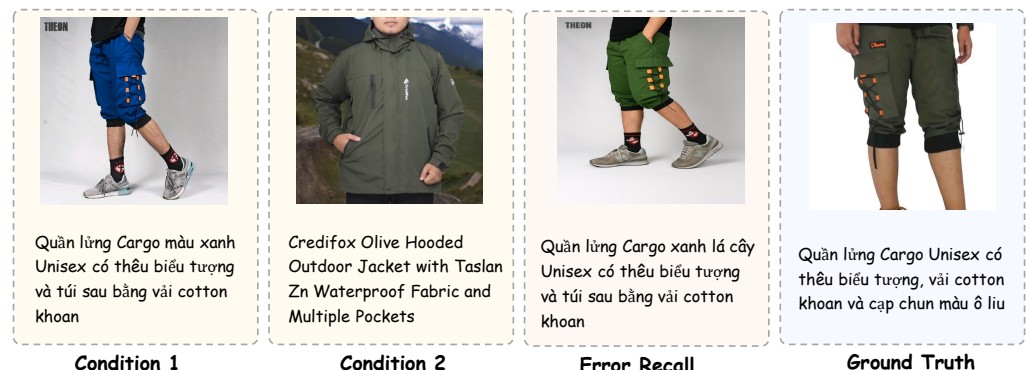

Figure 69: A case for Detail Error. In this example, although the recalled product meets both conditions, the detail condition "with a visible brand logo" is not met.

## E.6 Human preference

We have supplemented our evaluation with human preference experiments. Our human preference study is conducted in a multiple-choice comparison format. Specifically, we randomly sampled 100 items from MERIT, with each model providing their top-1 (Choice@1) or top-3 (Choice@3) most similar items as options. Five annotators were asked to select the best option (when the best item was retrieved by multiple models, we credited the model with the highest similarity score during recall). We report the average selection rate. As shown in Table 11, CORAL demonstrates superior performance in the human preference study, particularly achieving 58.9% selection rate in the Choice@3 setting.

Table 11: Human preference study on MERIT

|  | GME-Qwen2VL-2B | LamRA-Qwen2.5VL-7B | BGE-VL-7B | CORAL-Qwen2.5VL-3B |
|---|---|---|---|---|
| Choice@1 | 15.0 | 18.0 | 13.4 | 53.6 |
| Choice@3 | 11.7 | 14.4 | 14.9 | 58.9 |

## E.7 Condition Count on MERIT

As demonstrated in Table 12, we observe that the performance of nearly all models degrades with increasing condition count. Notably, under the most complex scenario with 4 conditions, even the best-performing model achieves only 36.84 R@5, highlighting the inherent difficulty posed by the combinatorial complexity of multi-condition queries.

Table 12: Model Performance Comparison

| Model | #Condition | R@1↑ | R@5↑ | R@10↑ | MRR↑ |
|---|---|---|---|---|---|
| InternVL2.5-MPO-1B-Seq | 2 | 0.42 | 1.39 | 2.30 | 0.88 |
|  | 3 | 0.00 | 0.00 | 0.78 | 0.10 |
|  | 4 | 0.00 | 0.00 | 0.00 | 0.00 |
| Qwen2.5-VL-3B-Seq | 2 | 0.09 | 0.39 | 0.55 | 0.20 |
|  | 3 | 0.00 | 0.78 | 1.56 | 0.37 |
|  | 4 | 0.00 | 0.00 | 0.00 | 0.00 |
| GME-Qwen2VL-2B-Cat | 2 | 8.45 | 53.12 | 61.24 | 27.73 |
|  | 3 | 10.24 | 55.91 | 59.06 | 28.51 |
|  | 4 | 5.26 | 36.84 | 47.37 | 15.89 |
| LamRA-Qwen2.5VL-7B-Cat | 2 | 12.05 | 39.20 | 48.13 | 23.83 |
|  | 3 | 13.39 | 35.43 | 42.52 | 22.58 |
|  | 4 | 5.56 | 27.78 | 33.33 | 15.28 |
| VLM2Vec-4B-Seq | 2 | 0.44 | 1.87 | 2.98 | 1.04 |
|  | 3 | 0.00 | 1.57 | 2.36 | 0.70 |
|  | 4 | 0.00 | 0.00 | 0.00 | 0.00 |

# F  Broader Impact

## F.1  Impact

The broader impact of MERIT carries both potential benefits and risks upon deployment and release. Some considerations are unique due to the multimodal nature of our dataset, while others reflect challenges common to retrieval systems in product retrieval environments. Built upon multilingual semantic understanding across Southeast Asian markets, MERIT inherits issues associated with cross-cultural product retrieval and multi-condition query interpretation. Below, we outline risks and mitigation strategies for its release.

**Hallucination.** On one hand, since the product titles in our dataset were generated by GPT-4o [10], there exists potential for hallucination issues [36, 96]. On the other hand, similar to other retrieval datasets [49, 103, 75], models trained on MERIT may generate outputs that are disconnected from user intentions or input conditions. This raises concerns, particularly in product retrieval applications where purchase decisions depend on accurate results, as user requirements and their modes of expression are inherently variable.

**Biases.** Bias in training data can propagate to models employing MERIT, arising from both visual feature extraction and linguistic interpretation. This may result in biased retrieval outcomes or unfair representations across diverse cultural contexts. Additionally, multilingual processing can introduce further biases in language alignment, as noted by [60].

**Ethical Impacts.** This research does not present substantial ethical concerns. Furthermore, we affirm that our open-source data and model distribution comply with all corporate guidelines and industry regulations governing intellectual property and data sharing practices.

**Expected Societal Implications.** A significant societal benefit lies in enhancing cross-cultural product retrieval experiences through improved interleaved multi-condition semantic retrieval. However, challenges remain in ensuring fairness across linguistic and cultural boundaries. Strong ethical standards and ongoing evaluation are essential for maximizing positive impact. These issues aren't unique to our method but are prevalent across different techniques for multi-condition retrieval. Despite the challenges, we believe the benefits significantly outweigh the potential limitations, allowing ongoing investigation and improvement of retrieval models while engaging the community in developing better approaches [102, 84, 62, 99]. Moreover, the release of MERIT can foster new applications and research directions, contributing to the progress and responsible deployment of retrieval systems in multilingual product retrieval environments.

## F.2  Limitations

*(i)* Our dataset, while comprehensive, may inherit limitations from real-world product retrieval data, such as imbalances in product categories and potential biases in attribute distributions across different Southeast Asian markets. *(ii)* Despite rigorous filtering, the dataset might inevitably contain some inconsistencies between visual attributes and textual descriptions, which could adversely affect model training and evaluation.

