# OpenReview forum: "MERIT: Multilingual Semantic Retrieval with Interleaved Multi-Condition Query"
_NeurIPS.cc/2025/Conference — NeurIPS 2025 poster_

### Official Review · Reviewer_6FnV · 2025-07-01

**Clarity:** 3
**Significance:** 4
**Originality:** 4
**Rating:** 5
**Confidence:** 5

**Summary:**

This paper introduces a new benchmark dataset, MERIT, and an improved fine-tuning method, CORAL, to advance the field of semantic retrieval. Existing benchmarks are often limited to single-language, single-image, and single-condition queries, failing to reflect complex, real-world e-commerce scenarios. To address this, they created MERIT, a multilingual dataset for text-image interleaved, multi-condition product retrieval, containing 320,000 queries over 135,000 products in 5 languages. They also propose CORAL to overcome the limitation that existing embedding models lose fine-grained conditional details in queries, that combines contrastive learning with text and image reconstruction. The authors show that CORAL achieves a 45.9% performance improvement on MERIT.

**Questions:**

1. In Figure 2, the query condition "same selling method" is ambiguous. Could you clarify this example?
2. How does model performance vary with the number of conditions in a query (e.g., two vs. three vs. four)? The main results do not show this breakdown.
3. For the out-of-distribution results in Figure 6(b), could you clarify the setting? Was the Qwen2.5-VL trained on a subset of MERIT?
4. For clarity, consider making the introduction of the CORAL framework (line 241) a new subsection separate from "Preliminaries".
5. In Eq5, the CORAL loss function includes two hyperparameters, λ1​ and λ2​, to balance the reconstruction losses. Could you elaborate on how these weights were selected and their sensitivity?

**Ethical Concerns:**

["NO or VERY MINOR ethics concerns only"]

**Final Justification:**

The authors addressed my concerns on content clarity and ablation results. I keep my original positive rating since I think the dataset can be valuable to the community.

**Limitations:**

yes

**Quality:**

4

**Strengths And Weaknesses:**

Strengths:
1. The paper tackles a highly relevant and underserved problem. As retrieval systems are integrated into real-world applications, the limitations of simple, single-condition benchmarks become increasingly apparent. The authors' focus on creating a benchmark for multilingual, multi-image, and multi-condition retrieval is a crucial step to address this gap.
2. The creation of the MERIT dataset is a massive undertaking and a significant contribution. The data collection process is well-documented and rigorous, involving multiple stages. This dataset will be an invaluable resource for the community.
3. The paper proposes a novel and effective solution CORAL, motivated by the error analysis performed on existing models. The combination of contrastive learning with reconstruction is an effective approach, with the reported 45.9% performance gain on MERIT and strong generalization to other benchmarks.

Weaknesses:
1. The product data is sourced from a private, "internal dataset". While the authors have anonymized the data, the lack of information about the original source (e.g., a single e-commerce platform vs. multiple) makes it difficult to reason about potential inherent biases in the product distribution, metadata structure, or visual style.
2. The focused modality (text, image) and scenario (product search) are not accurately reflected in the title (semantic retrieval is too broad).
3. Some experiment settings/details are not clear.

---

> ### Author Rebuttal · Authors · 2025-07-31
>
> We sincerely thank Reviewer `6FnV` for the thoughtful and constructive feedback. We are especially encouraged by your recognition of the real-world relevance of our problem setting, the rigor and scale of the MERIT dataset, and the effectiveness of our CORAL framework. Below, we address your comments point by point:
>
> ---
> > **`Q1`:** *"The product data is sourced from a private, 'internal dataset'. While the authors have anonymized the data, the lack of information about the original source (e.g., a single e-commerce platform vs. multiple) makes it difficult to reason about potential inherent biases in the product distribution, metadata structure, or visual style."*
>
> **A1:** Thank you for your question. Our original data is sourced from **a single e-commerce platform**. We have provided **distribution statistics** in Appendix Section `D` and **data examples** in Appendix Section `E`. Due to the double-blind reviewing process, we cannot disclose additional data information at this time. However, we have committed to providing more comprehensive data details and fully open-sourcing our data and code (both annotation and training components).
>
> ---
> > **`Q2`:** *"The focused modality (text, image) and scenario (product search) are not accurately reflected in the title (semantic retrieval is too broad)."*
>
> **A2:** Thank you for your suggestion. We will consider revising the title of our paper in future iterations, as you know, modifications are not permitted during the rebuttal phase. We will consider incorporating keywords related to modality and scenario.
>
> ---
> > **`Q3`:** *"Some experiment settings/details are not clear."*
> >
> > * **`Q3.1`**: *"In Figure 2, the query condition 'same selling method' is ambiguous. Could you clarify this example?"*
> >
> > * **`Q3.2`**: *"How does model performance vary with the number of conditions in a query (e.g., two vs. three vs. four)? The main results do not show this breakdown."*
> >
> > * **`Q3.3`**: *"For the out-of-distribution results in Figure 6(b), could you clarify the setting? Was the Qwen2.5-VL trained on a subset of MERIT?&#x20;"*
> >
> > * **`Q3.4`**: *"For clarity, consider making the introduction of the CORAL framework (Line 241) a new subsection separate from 'Preliminaries'."*
> >
> > * **`Q3.5`**: *"In Eq. 5, the CORAL loss function includes two hyperparameters, λ1 and λ2, to balance the reconstruction losses. Could you elaborate on how these weights were selected and their sensitivity?"*
>
> **A3:** Thank you for your question! We apologize for any unclear details in the manuscript. Here, we will explain them point by point.&#x20;
>
> - **A 3.1:** We inadvertently reversed the plotting of product1 and product2 in the figure. "Same selling method" refers to the bundled selling approach where 3 items are sold together as a package. We have revised this mirror in the revised manuscript.
>
> - **A 3.2**: We thank you for this valuable suggestion. Due to time constraints, we have supplemented Table `C` in the supplementary material with results showing the performance of several key models as a function of the number of query conditions. As demonstrated in Table `C` below, we observe tha&#x74;**&#x20;the performance of nearly all models degrades with increasing condition count**. Notably, under the most complex scenario with 4 conditions, even the best-performing model achieves only 36.84 R@5, highlighting the inherent difficulty posed by the combinatorial complexity of multi-condition queries.
>
> ***Table C.** Impact of query condition count on model performance on MERIT*
> |  | #condition | R@1↑  | R@5↑  | R@10↑  | MRR↑  |
> | ---------------------- | ---------- | ----- | ----- | ------ | ----- |
> | InternVL2.5-MPO-1B-Seq | 2          | 0.42  | 1.39  | 2.30   | 0.88  |
> |                        | 3          | 0.00  | 0.00  | 0.78   | 0.10  |
> |                        | 4          | 0.00  | 0.00  | 0.00   | 0.00  |
> |  Qwen2.5-VL-3B-Seq     | 2          | 0.09  | 0.39  | 0.55   | 0.20  |
> |                        | 3          | 0.00  | 0.78  | 1.56   | 0.37  |
> |                        | 4          | 0.00  | 0.00  | 0.00   | 0.00  |
> | GME-Qwen2VL-2B-Cat     | 2          | 8.45  | 53.12 | 61.24  | 27.73 |
> |                        | 3          | 10.24 | 55.91 | 59.06  | 28.51 |
> |                        | 4          | 5.26  | 36.84 | 47.37  | 15.89 |
> | LamRA-Qwen2.5VL-7B-Cat | 2          | 12.05 | 39.20 | 48.13  | 23.83 |
> |                        | 3          | 13.39 | 35.43 | 42.52  | 22.58 |
> |                        | 4          | 5.56  | 27.78 | 33.33  | 15.28 |
> | VLM2Vec-4B-Seq         | 2          | 0.44  | 1.87  | 2.98   | 1.04  |
> |                        | 3          | 0.00  | 1.57  | 2.36   | 0.70  |
> |                        | 4          | 0.00  | 0.00  | 0.00   | 0.00  |
>
>
> - **A 3.3:** In Figure `6`(b), we evaluate three types of OOD scenarios. You can find the specific experimental details and more comprehensive result tables (please refer to Tables `7`, `8`, and `9`) in Appendix Section `E.4`. In brief, *"OOD"* refers to removing the same language, class, or attribute from the training set during testing, while *"mixed"* refers to training on the complete dataset.
>
> - **A 3.4:** Thank you for your feedback. We have committed to incorporating your suggestions in the revised version of the manuscript.
>
> - **A 3.5**:  We conduct ablation studies on the mask ratio, temperature (τ), loss weights (λ), and the decoder, as shown in Table `D`. Overall, CORAL demonstrates **robust training dynamics&#x20;**&#x61;nd exhibits low sensitivity to variations in temperature and the number of decoder heads.
>
> Regarding the selection of loss weights, we first establish through ablation experiments in Table `3` tha&#x74;**&#x20;joint reconstruction of language and vision modalities** is more effective than reconstructing individual modalities. Therefore, in the ablation experiments presented in Table `D` below, we maintain equal weights for both modalities. Our findings indicate that smaller values of λ have minimal impact on the results. However, overly high loss weights can lead the model to overemphasize the reconstruction objective, consequently degrading retrieval performance.
>
> ***Table D.** Further ablation study for CORAL on MERIT using Qwen2.5-VL-3B*
> |                      |                               | R@1↑  | R@5↑  | R@10↑  | MRR↑  |
> | -------------------- | ----------------------------- | ----- | ----- | ------ | ----- |
> | Mask Ratio           |  language 0.25 vision 0.25    | 69.98 | 88.88 | 92.83  | 78.22 |
> |                      |  language 0.5 vision 0.5      | 69.68 | 89.26 | 93.08  | 78.33 |
> |                      |  language 0.75 vision 0.75    | 60.97 | 82.31 | 87.47  | 70.38 |
> | Temperature ($τ$)  | 0.001                         | 67.48 | 87.51 | 91.77  | 76.24 |
> |                      | 0.02                          | 69.68 | 89.26 | 93.08  | 78.33 |
> |                      | 0.1                           | 67.34 | 87.10  | 91.07  | 76.00    |
> | Loss Weights ($λ$) | $λ_l$=0.05, $λ_v$=0.05 (default)          | 69.60  | 89.08 | 93.21  | 78.22 |
> |                      |  $λ_l$=0.1, $λ_v$=0.1  | 69.68 | 89.26 | 93.08  | 78.33 |
> |                      | $λ_l$=1.0, $λ_v$=1.0            | 58.41 | 80.31 | 86.25  | 68.18 |
> | Decoder Head         | 4                             | 67.89 | 88.26 | 92.28  | 76.85 |
> |                      | 8                             | 68.68 | 88.62 | 92.76  | 77.46 |
> |                      | 16                            | 69.68 | 89.26 | 93.08  | 78.33 |
> |                      | 32                            | 70.22 | 89.60  | 93.36  | 78.68 |
>
> ---
> Last but not least, we would like to sincerely thank Reviewer `6FnV` again for the valuable time and constructive feedback provided during this review.

---

> > ### Author Response · Authors · 2025-08-03
> > **Looking forward to hearing from you**
> >
> > Dear Reviewer `6FnV`,
> >
> > Thank you very much for your thoughtful and constructive feedback!
> >
> > Following your suggestions, we have clarified the dataset source and limitations under the constraints of double-blind review, provided detailed breakdowns on condition counts, expanded out-of-distribution analysis, and reorganized the CORAL framework section for clarity. We also included comprehensive ablations on mask ratio, decoder design, temperature, and loss weights, demonstrating CORAL’s robust training behavior and identifying effective hyperparameter configurations.
> >
> > We sincerely appreciate your recognition of the real-world relevance of our setting, the rigor of our dataset, and the effectiveness of our framework.
> >
> > We remain available for the Author-Reviewer Discussion phase and are happy to clarify anything else!
> >
> > *Best regards,*
> >
> > The Authors of Submission 1565

---

> > > ### Comment · Reviewer_6FnV · 2025-08-04
> > >
> > > Thank the authors for the detailed response and please incorporate the changes into the final manuscript. I keep the current rating.

---

> > > > ### Author Response · Authors · 2025-08-05
> > > > **Thank you for your acknowledgment!**
> > > >
> > > > Dear Reviewer `6FnV`,
> > > >
> > > > We sincerely appreciate your acknowledgment of our rebuttal. We are glad to hear that your concerns have been addressed and are voting toward acceptance.
> > > >
> > > > Your support and encouragement mean a lot to us!
> > > >
> > > > *Best regards,*
> > > >
> > > > The Authors of Submission 1565

---

### Official Review · Reviewer_WJjZ · 2025-07-03

**Clarity:** 2
**Significance:** 3
**Originality:** 3
**Rating:** 4
**Confidence:** 3

**Summary:**

This paper makes three contributions: 1) it introduces a new multilingual, multimodal product retrieval dataset called MERIT, 2) finds that existing retrieval models perform poorly on the MERIT task, and 3) proposes a fine-tuning technique called CORAL that achieves a 46% improvement over baselines on MERIT. The first portion of the paper is devoted to discussing the construction of the MERIT dataset and, in particular, how the data was collected annotated and refined. The authors then apply existing retrieval models to MERIT where they find that these off-the-shelf techniques exhibit poor performance. This finding motivates the authors to introduce CORAL, a multi-modal fine-tuning framework consisting of three loss functions that achieves strong performance on 8 previously introduced retrieval tasks as well as top performance on MERIT.

**Questions:**

Q1: What precisely do the authors mean by "interleaved" and why do existing benchmarks fail to meet this property. *I would be willing to raise my evaluation score if the authors can provide a clear definition of interleaved and explain why this is a novel property of their dataset.*

Q2: How were the products sourced for the MERIT dataset? What precisely is the role of GPT-4o in the dataset construction process? How do the authors control for potential errors from using GPT-4o in the dataset construction process? *I would be willing to raise my evaluation score if the authors can provide answers to these questions and, in general, provide more details on how the dataset was sourced (including perhaps a file with data examples)*

Q3: Are there any works in the multimodal contrastive learning literature that are related to CORAL? What distinguishes CORAL from these prior works? *I would be willing to raise my evaluation score if the authors can include a deeper discussion of related work around the CORAL model (particularly in contrastive learning). Currently, I feel that this important positioning of CORAL in relation to the literature is lacking in the paper*

Q4: How were the expert annotators recruited for this project and how were they compensated? What is the license of the MERIT dataset? *I would be willing to raise my evaluation score if the authors can provide satisfactory answers to these questions*

**Ethical Concerns:**

["NO or VERY MINOR ethics concerns only"]

**Final Justification:**

During the rebuttal period, the authors addressed my concerns around providing a formal definition of interleaving and providing important details on the dataset license and human annotation process. Thus, I have decided to raise my score from a 3 to 4. One unresolved issue (which motivated me not to give a higher rating) is that the authors were not able to provide access to a slice of the data for review. This is a requirement in the Datasets and Benchmarks track, for example, and is an important component of evaluating a new dataset contribution.

**Limitations:**

Yes

**Paper Formatting Concerns:**

None.

**Quality:**

2

**Strengths And Weaknesses:**

Strengths

1. [Originality/Significance] As a multimodal and multilingual product retrieval dataset focused on Southeast Asian languages, I believe that MERIT has the potential to stand out as a novel dataset contribution to the community. The authors motivate the need for a dataset that more tightly integrates different modalities (though they do not precisely define their term "interleaved" -- please see the "Weaknesses" section below) quite well.

2. [Quality/Significance] The experiments with the CORAL framework are well-executed with results on both MERIT as well as 8 existing retrieval datasets which provides robust evidence that the technique is effective.

Weaknesses

1. [Clarity] The paper lacks sufficient details in several places. First and foremost, as alluded to above, the authors, to my knowledge, do not provide a precise definition of what they mean by "interleaved" and, to my knowledge, this is not a standard term in the retrieval literature. It would be very helpful for the authors to clearly define what they mean by interleaved, which would help in understanding how the MERIT dataset stands out from prior work.

2. [Quality] The paper also lacks sufficient details on how the MERIT dataset was constructed. In particular, it was not clear to me from reading the paper 1) how the products were sourced and from where, 2) what specific role GPT-4o played in the dataset construction (line 130 of the paper), and 3) who the annotators are and how they were selected/compensated? While the authors provide examples of what individual items in the dataset look like, I did not see any link to be able to download and inspect the dataset files directly (though I may have missed this). Overall, this paper covers a substantial amount of ground in introducing a new benchmark and proposing a new method and I would encourage the authors to consider whether these contributions might be better appreciated if there were more space to devote to each one as standalone papers.

3. [Quality] There is a rich body of literature on contrastive learning for multimodal models, but the authors do not discuss this related work in much detail in the paper. I would suggest discussing more of this related work to better position the CORAL framework so that readers can understand what is novel about the technique. As it stands, I believe that the triple contrastive loss framework proposed in the paper is limited in novelty since similar ideas have been proposed (though the strength of the empirical results still validates the contribution). A deeper discussion of this related work might help the authors' novel contributions stand out more effectively.

---

> ### Author Rebuttal · Authors · 2025-07-31
>
> We sincerely thank Reviewer `WJjZ` for the thoughtful and constructive feedback. We are encouraged that you found our benchmark MERIT to be a novel and significant contribution, especially in highlighting multilingual and multimodal product retrieval, and appreciated the effectiveness and breadth of CORAL’s empirical evaluations. Below, we address each concern point by point.
>
> ---
> > **`Q1`:** *"What precisely do the authors mean by 'interleaved' and why do existing benchmarks fail to meet this property. I would be willing to raise my evaluation score if the authors can provide a clear definition of interleaved and explain why this is a novel property of their dataset."*
>
> **A1:** We provide a point-by-point clarification of what constitutes interleaved:
>
> - **(1.1) Definition:** In MLLMs, "interleaved" refers to the model's capability to simultaneously **process alternating sequences** of images and text.
>   - This capability enables more flexible understanding and generation of multimodal content, analogous to human cognitive processes *\[Dong et al., 2024]*.
>   - The typical sequence structure follows: `[Text₁] → [Image₁] → [Text₂] → [Image₂] → ...`
>   - Post-2024 MLLMs have predominantly adopted **interleaved sequence processing**, as this better reflects real-world usage patterns where users frequently present queries containing multiple distributed images.
>
> - **(1.2) Our Contribution:** Existing retrieval datasets exclusively consider single images or singular retrieval conditions, namely using an image to retrieve text, or using an image-text pair to retrieve an image-text pair. Our work presents the **first interleaved retrieval dataset**. As demonstrated in Figure `2` in the manuscript, our query format follows: `[Text₁] → [Product Image₁] → [Product Title₁] → [Product Image₂] → [Product Title₂] → ...`, representing a fundamental departure from previous dataset formulations.
>
> - **(1.3)** **Practical Significance**: Interleaved data enables the construction of multi-condition, cross-modal retrieval tasks, allowing for **more precise interpretation of user intent**. Moreover, the interleaved format aligns with the input structure of MLLMs, which uses interleaved input dat&#x61;*\[Alayrac et al., 2022]*, enhancing their **semantic representation capabilities** and thereby advancing progress in multimodal learning *\[Zhu et al., 2024]* .
>
> We have clarified the definition in the revised manuscript and better emphasized its novelty.
>
> **References:**
> - *\[Dong et al., 2024]* "Internlm-xcomposer2: Mastering free-form text-image composition and comprehension in vision-language large model." arXiv:2401.16420, 2024.
> - *\[Zhu et al., 2024]* "Multimodal c4: An open, billion-scale corpus of images interleaved with text." NeurIPS, 2023.
> - *\[Alayrac et al., 2022]* "Flamingo: A visual language model for few-shot learning." NeurIPS, 2022.
>
> ---
> > **`Q2`:** *"How were the products sourced for the MERIT dataset? What precisely is the role of GPT-4o in the dataset construction process? How do the authors control for potential errors from using GPT-4o in the dataset construction process? I would be willing to raise my evaluation score if the authors can provide answers to these questions and, in general, provide more details on how the dataset was sourced (including perhaps a file with data examples)."*
>
> **A2:** We apologize for any unclear aspects in our paper and will address your concerns point by point.
>
> - **(2.1)** As described in Appendix Section `A.1`, our data is sourced from our internal dataset. Specifically, we curated popular products across six Southeast Asian countries. Due to the double-blind review process, we cannot provide further details at this time. However, we commit to releasing more comprehensive details once the paper is accepted.
>
> - **(2.2)** Overall, GPT was involved in attribute selection, product annotation, query generation, and filtering processes, as indicated by the robot icons in Figure `2` in the manuscript. We take **annotation accuracy very seriously**. Following each GPT annotation step, we implemented GPT self-verification procedures. For critical steps (attribute identification and query generation), we employed manual filtering (10,000 labor hours in total) to ensure quality. We provide an 11-page detailed description of our annotation process in Appendix Section `A`, which thoroughly explains the usage of GPT in our pipeline.
>
> - **(2.3)** Regarding detailed data information, we provide distribution statistics in Appendix Section `D` and data examples in Appendix Section `E`. Due to rebuttal limitations preventing file uploads and the rule stating *"All the texts you post (rebuttal, discussion) should not contain any links to external pages"*, we cannot upload the requested folders containing data examples and sources. However, we commit to fully open-sourcing our data and code (both annotation and training) following the completion of the review process.
>
> We acknowledge these unclear aspects you have identified and commit to addressing them comprehensively in the revised version of our paper.
>
> ---
> > **`Q3`:** *"How were the expert annotators recruited for this project, and how were they compensated? What is the license of the MERIT dataset?"*
>
> **A3:** Thanks for your question.
>
> - **(3.1)** Our annotators are professional in-house annotation staff with **extensive labeling experience** and bachelor's degrees or higher. The specific compensation is determined by our institution's policy, with an estimated total exceeding $10,000 USD. We ensure that annotators receive fair compensation, maintain appropriate workloads, and prioritize their physical and mental well-being.
>
> - **(3.2)** Quality control was strictly enforced during annotation. All annotators **underwent training and a qualification phase** (including trial annotations) before formal participation. Additionally, we employed dedicated quality-check specialists to conduct periodic random inspections.
>
> - **(3.3)** Our licensing follows **CC BY-NC 4.0**, which we have explicitly specified in the revised version of the manuscript.
>
> ---
> > **`Q4`:** *"Are there any works in the multimodal contrastive learning literature that are related to CORAL? What distinguishes CORAL from these prior works? I would be willing to raise my evaluation score if the authors could include a deeper discussion of related work around the CORAL model (particularly in contrastive learning). Currently, I feel that this important positioning of CORAL in relation to the literature is lacking in the paper."*
>
> **A4:** We thank you for your insightful question and will address each point systematically:
>
> - **(4.1)** While we acknowledge that the literature on multimodal contrastive learning is indeed extensive, our method specifically targets **MLLM-based retrieval approaches**, focusing on training algorithms for transitioning from MLLMs to multimodal retrieval models. Therefore, our related work section concentrates on discussing MLLMs and Multimodal Retrieval Models as most relevant to our contribution.
>
> - **(4.2)** Through our experimental analysis, we identified key limitations in existing MLLM-based retrieval methods: over-reliance on global semantic information while neglecting conditional query elements, failing to extract specific attributes, and misinterpreting visual content. To address these challenges, we propose CORAL, which introduces embedding reconstruction on top of conventional contrastive learning to preserve **fine-grained conditional elements rather than solely supervising the \[EOS] token,&#x20;**&#x61;s detaile&#x64;**&#x20;in lines 100-110&#x20;**&#x6F;f our manuscript.
>
> - **(4.3)** Moreover, to the best of our knowledge, we are the **first to introduce multimodal embedding reconstruction in MLLM-based retrieval**. The results on MERIT and 12 standard benchmarks also demonstrate the effectiveness of our approach.
>
> We greatly appreciate your valuable feedback and have committed to incorporating more comprehensive discussions of contrastive learning-related work in the revised version of our manuscript.
>
> ---
> Last but not least, we would like to sincerely thank Reviewer `WJjZ` again for the valuable time and constructive feedback provided during this review.

---

> > ### Comment · Reviewer_WJjZ · 2025-08-01
> >
> > Thank you to the authors for their very thoughtful and detailed rebuttal. I especially appreciate their responses on the definition of interleaved (and hope they can highlight this in the revised version of the paper). I also hope the details on the dataset license and the role of GPT and the human annotators can be included in the revision as well.
> >
> > Based on the rebuttal, I will increase my rating of the paper from a 3 to a 4. I believe that this dataset constitutes a good contribution to the community. However, in the absence of being able to review the dataset files directly, I choose to keep my rating at a 4 and not higher. I recognize that the authors are limited here by double-blind guidelines, but I would note that the Datasets and Benchmarks track is not limited by this double-blind restriction so I would encourage the authors to keep this in mind for future submissions.

---

> > > ### Author Response · Authors · 2025-08-01
> > > **Thank you for your response!**
> > >
> > > Dear Reviewer `WJjZ`,
> > >
> > > Thank you again for your thoughtful engagement and for raising your score based on our response.
> > >
> > > We truly appreciate your recognition of MERIT as a meaningful contribution. Regarding your remaining concern on reviewing the dataset directly:
> > > - We completely understand the importance of transparency. While the double-blind policy currently limits us from releasing the dataset during review, we have already completed all necessary preparations for public release.
> > > - We will fully open-source the **dataset**, **annotation pipeline**, and **training code** immediately upon acceptance, under the `CC BY-NC 4.0` license.
> > >
> > > We are committed to providing the community with a reproducible and impactful benchmark and would be grateful if you could consider this in your final assessment.
> > >
> > > Once again, we would like to sincerely thank you for your valuable time and constructive feedback provided during this review!
> > >
> > > *Warm regards,*
> > >
> > > The Authors of Submission 1565

---

### Official Review · Reviewer_MHMi · 2025-07-09

**Clarity:** 3
**Significance:** 3
**Originality:** 3
**Rating:** 5
**Confidence:** 4

**Summary:**

The paper introduces MERIT, the first large-scale benchmark that targets multilingual, multi-image, and multi-condition product retrieval. MERIT contains 135 k products and 320 k query–target pairs across seven e-commerce categories .After showing that nine state-of-the-art (SOTA) retrievers achieve < 12 % R@1 on MERIT , the authors diagnose that current fine-tuning regimes over-focus on a single [EOS] token and thus miss fine-grained attribute cues. They propose CORAL, a fine-tuning framework that couples supervised contrastive learning with masked embedding reconstruction for both vision & text streams. CORAL lifts R@1 by 45.9 % on MERIT and improves eight external benchmarks (MS-COCO, VisDial, etc.) without task-specific tweaks.

**Questions:**

1. How many GPU-hours did full CORAL training require for the 7B model, and what is the inference latency compared to baseline contrastive tuning?
2. Have you measured language- or category-specific biases in the retrieval results beyond the small sample in Fig. 7? Any mitigation plans?
3. How sensitive is CORAL to the 50 % visual/text masking ratio? Could adaptive masking yield further gains?

**Ethical Concerns:**

["NO or VERY MINOR ethics concerns only"]

**Limitations:**

1. Dataset focuses on consumer goods; tasks like scene or document retrieval are out of scope.
2. CORAL needs access to image embeddings during decoding; models without token-level vision hooks may not benefit.

**Quality:**

3

**Strengths And Weaknesses:**

Strengths
1. The benchmark is timely and realistic, which is the first to combine multilingual, multi-image, and compositional attribute retrieval at scale.
2. CORAL directly tackles the identified issue of missing fine-grained, conceptually simple yet effective.
3. The experiments are strong and broad, which contain 9 baselines on MERIT plus 8 public datasets, full ablations, and error analysis.

Weakness
1. The section of “Masked Embedding Reconstruction” is poorly written and needs to be reorganized. For example, Mask_v and Mask_l are sometimes one parameter, sometimes 2; symbols such as q and E do not clearly state what they are; and the meaning of the corner markers under (i) in Equation 4 is unclear.
2. Lack the analysis of reconstruction mask ratio δ, choice of decoders, and temperature $\tao$.
3. The metric relies solely on recall/MRR. User-centric metrics (CTR, dwell time) or ranking quality are absent.

---

> ### Author Rebuttal · Authors · 2025-07-31
>
> We sincerely thank Reviewer `MHMi` for the thoughtful and constructive feedback. We are glad that you found our benchmark timely and realistic, CORAL conceptually simple yet effective, and our experiments strong and comprehensive. Below, we address your questions and concerns in detail.
>
> ---
> > **`Q1`:** *"The section of 'Masked Embedding Reconstruction' is poorly written and needs to be reorganized..."*
>
> **A1:** Thanks a lot for your suggestion. We have addressed the mentioned typos and improved the elaboration in the manuscript. We commit to making detailed revisions to these unclear aspects to improve readability. Additionally, we commit to open-sourcing the code and our trained models for public use.
>
> ---
> > **`Q2`:** *"Lack the analysis of reconstruction mask ratio δ, choice of decoders, and temperature."*
> >
> > **`Q3`:** *"How sensitive is CORAL to the 50 % visual/text masking ratio? Could adaptive masking yield further gains?"*
>
> **A2** **&** **A3:** Thank you for your comment. We conduct ablation studies on the mask ratio, temperature (τ), loss weights (λ), and the decoder, as shown in Table `D` below:
>
> ***Table D.** Further ablation study for CORAL on MERIT using Qwen2.5-VL-3B*
> |     |     | R@1↑  | R@5↑  | R@10↑  | MRR↑  |
> | - | - | - | - | - | - |
> | Mask Ratio           |  language 0.25 vision 0.25    | 69.98 | 88.88 | 92.83  | 78.22 |
> |                      |  language 0.5 vision 0.5      | 69.68 | 89.26 | 93.08  | 78.33 |
> |                      |  language 0.75 vision 0.75    | 60.97 | 82.31 | 87.47  | 70.38 |
> | Temperature ($τ$)  | 0.001                         | 67.48 | 87.51 | 91.77  | 76.24 |
> |                      | 0.02                          | 69.68 | 89.26 | 93.08  | 78.33 |
> |                      | 0.1                           | 67.34 | 87.1  | 91.07  | 76.00    |
> | Loss Weights ($λ$) | $λ_l$=0.05, $λ_v$=0.05 (default)          | 69.6  | 89.08 | 93.21  | 78.22 |
> |                      |  $λ_l$=0.1, $λ_v$=0.1  | 69.68 | 89.26 | 93.08  | 78.33 |
> |                      | $λ_l$=1.0, $λ_v$=1.0            | 58.41 | 80.31 | 86.25  | 68.18 |
> | Decoder Head         | 4                             | 67.89 | 88.26 | 92.28  | 76.85 |
> |                      | 8                             | 68.68 | 88.62 | 92.76  | 77.46 |
> |                      | 16                            | 69.68 | 89.26 | 93.08  | 78.33 |
> |                      | 32                            | 70.22 | 89.6  | 93.36  | 78.68 |
>
> Key findings:
> - Overall, CORAL demonstrates **robust training** dynamics and exhibits low sensitivity to variations in temperature and the number of decoder heads.
> - However, we observe that excessively high mask ratios can be detrimental to performance. This phenomenon likely stems from the model's diminished capacity to leverage existing knowledge for embedding reconstruction *[Xiao et al., 2020]*. Specifically, when the mask ratio is increased to **0.75**, we observe a substantial **10.1%** decrease in MRR compared to the 0.5 setting.
> - Furthermore, overly high loss weights can lead the model to overemphasize the reconstruction objective, consequently degrading retrieval performance.
>
> **Reference:**
> - *[Xiao et al., 2020]* "RetroMAE: Pre-Training Retrieval-oriented Language Models Via Masked Auto-Encoder." EMNLP, 2022.
>
> ---
> > **`Q4`:** *"The metric relies solely on recall/MRR. User-centric metrics (CTR, dwell time) or ranking quality are absent."*
>
> **A4:** Thank you for your question. We address this concern from the following aspects:
>
> - **(4.1)** First, it is important to clarify that our task falls **within the domain of semantic retrieval** based on **multi-modal content comprehension** rather than **recommendation systems**. Therefore, the standard evaluation metrics are recall and MRR, and our related work also does not involve user-centric metrics such as CTR.
> - **(4.2)** Nevertheless, we have supplemented our evaluation with **human preference experiments**. Our human preference study is conducted in a multiple-choice comparison format. Specifically, we randomly sampled 100 items from MERIT, with each model providing their top-1 (Choice@1) or top-3 (Choice@3) most similar items as options. Five annotators were asked to select the best option (when the best item was retrieved by multiple models, we credited the model with the highest similarity score during recall). We report the average selection rate. As shown in Table `B`, CORAL demonstrates **superior performance** in the human preference study, particularly achieving **58.9%** selection rate in the Choice@3 setting.
>
> ***Table B.** Human preference study on MERIT*
> | Model | GME-Qwen2VL-2B | LamRA-Qwen2.5VL-7B | BGE-VL-7B | CORAL-Qwen2.5VL-3B |
> |-------|:-:|:-:|:-:|:-:|
> | Choice@1 | 15.0 | 18.0 | 13.4 | 53.6 |
> | Choice@3 | 11.7 | 14.4 | 14.9 | 58.9 |
>
> ---
> > **`Q5`:** *"How many GPU-hours did full CORAL training require for the 7B model, and what is the inference latency compared to baseline contrastive tuning?"*
>
> **A5:** Thank you for asking.
>
> - **(5.1)** We first clarify that our paper presents results for the 3B model rather than the 7B model. Under full-parameter training conditions using 8×H100 GPUs with batch size 64 for the LLM backbone, **one epoch requires approximately 5 hours for the 3B model and 6.5 hours for the 7B model.**
> - **(5.2)** Since our approach only introduces an additional decoder during training compared to the baseline, our method **incurs no additional inference latency** relative to the baseline model.
>
> ---
> > **`Q6`:** *"Have you measured language- or category-specific biases in the retrieval results beyond the small sample in Fig. 7? Any mitigation plans?"*
>
> **A6:** Thank you for your insightful question. We measure the biases as follows:
>
> - **(6.1)** In Figure `5`(a), we evaluate language bias and observe that models trained with multilingual data **do not exhibit significant language bias**.
> - **(6.2)** As demonstrated in Table `7` of the supplemental material, Qwen2.5-VL exhibits biased zero-shot performance across different languages and classes, though the magnitude of bias remains relatively modest. This bias likely stems from the distributional characteristics of the pretraining data.
> - **(6.3)** Comparing Table `7` and Table `9` reveals that **jointly training** across multiple classes and languages **improves performance** on individual classes and languages. This suggests that increasing data diversity can effectively mitigate such biases.
>
> ***Table 7.** Out-of-distribution test results for Qwen2.5-VL trained on in-domain data*
> |               |         | MRR   | R@1   | R@5   | R@10  |
> | ------------- | ------- | ----- | ----- | ----- | ----- |
> | **Language**  | ID      | 44.26 | 15.98 | 79.34 | 87.91 |
> |               | MS      | 42.82 | 17.97 | 74.65 | 84.33 |
> |               | TH      | 45.00 | 19.80 | 75.29 | 84.22 |
> | **Attribute** | Brand   | 60.72 | 48.03 | 78.07 | 85.60 |
> |               | Pattern | 54.93 | 37.23 | 77.74 | 84.53 |
> |               | Region  | 44.29 | 19.10 | 78.65 | 86.52 |
> | **Class**     | Drink   | 51.14 | 33.47 | 73.47 | 81.05 |
> |               | Phone   | 43.47 | 31.77 | 58.63 | 67.72 |
> |               | Table   | 48.57 | 35.07 | 64.69 | 74.64 |
>
> ***Table 9.** Out-of-distribution test results for Qwen2.5-VL trained on the whole train set*
> |               |         | MRR   | R@1   | R@5   | R@10  |
> | ------------- | ------- | ----- | ----- | ----- | ----- |
> | **Language**  | ID      | 71.34 | 57.31 | 88.89 | 94.93 |
> |               | MS      | 67.51 | 53.23 | 85.48 | 91.94 |
> |               | TH      | 69.79 | 55.29 | 87.65 | 91.57 |
> | **Attribute** | Brand   | 71.87 | 59.62 | 88.98 | 93.59 |
> |               | Pattern | 63.50 | 50.30 | 80.44 | 87.03 |
> |               | Region  | 57.81 | 38.20 | 83.15 | 93.26 |
> | **Class**     | Drink   | 53.54 | 33.26 | 78.53 | 86.74 |
> |               | Phone   | 63.64 | 51.58 | 79.78 | 86.57 |
> |               | Table   | 52.40 | 38.63 | 69.19 | 77.96 |
>
> ---
> > **`Q7`:** *"The dataset focuses on consumer goods; tasks like scene or document retrieval are out of scope."*
>
> **A7:** Thanks for your comment.
>
> - **(7.1)** Interleaved multi-condition retrieval represents a novel and inherently **challenging task that has not been previously explored** in the literature due to the **complexity of data collection.** Our proposed MERIT constitutes the **first** dataset specifically designed for this task formulation, as collecting multi-condition retrieval data for document retrieval presents significant practical difficulties.
> - **(7.2)** Meanwhile, product retrieval serves a **representative and tractable domain** for multi-condition retrieval scenarios and **holds substantial real-world significance**, which motivated our domain selection.
> - **(7.3)** We acknowledge that extending interleaved multi-condition retrieval to additional domains beyond product search represents an important direction for future work, which we defer to subsequent investigations.
>
> ---
> > **`Q8`:** *"CORAL needs access to image embeddings during decoding; models without token-level vision hooks may not benefit."*
>
> **A8:** We appreciate your insightful question.
>
> - **(8.1)** MLLM-based models demonstrate superior performance in addressing semantic retrieval challenges *[Jiang et al., 2024]*, and CORAL is specifically **designed as a methodology for MLLM-based retrievers.**
> - **(8.2)** Current **mainstream MLLMs are equipped with token-level vision hooks**, including but not limited to GPT, InternVL, Qwen, Phi-V, LLaVa, and other prominent models in this domain. Thus, our method can **benefit most MLLM-based retrievers**.
>
> **Reference:**
> - *[Jiang et al., 2024]* "VLM2Vec: Training Vision-Language Models for Massive Multimodal Embedding Tasks." ICLR, 2025.
>
> ---
> Last but not least, we would like to sincerely thank Reviewer `MHMi` again for the valuable time and constructive feedback provided during this review.

---

> > ### Author Response · Authors · 2025-08-03
> > **Looking forward to hearing from you**
> >
> > Dear Reviewer `MHMi`,
> >
> > Thank you very much for your thoughtful and constructive review!
> >
> > Following your suggestions, we revised and clarified the description of our masked embedding reconstruction module, added extensive ablation studies on masking ratio, decoder design, and hyperparameters, and included a new human preference evaluation to supplement standard metrics. We also provided details on training cost, latency, and dataset bias analysis, and clarified CORAL’s applicability to MLLM-based retrievers with token-level vision hooks.
> >
> > Your feedback helped us significantly improve both the clarity and technical depth of the paper.
> >
> > We remain available for further clarification during the Author-Reviewer Discussion phase and look forward to any additional comments!
> >
> > *Best regards,*
> >
> > The Authors of Submission 1565

---

### Official Review · Reviewer_YCpi · 2025-07-22

**Clarity:** 3
**Significance:** 3
**Originality:** 3
**Rating:** 5
**Confidence:** 4

**Summary:**

The authors made two contributions:

They first proposed a novel opensource multi-lingual dataset for interleaved multi-condition semantic retrieval (MERIT), highlighting difficult queries that involve multiple conditions that's lacking in previous open datasets.

Secondly, they proposed a contrastive reconstruction for multimodal retrieval (CORAL) as a novel fine-tuning framework to adapt pretrained LLM for retrieval. In essence, this encodes a series of product images and product title descriptions as image/text interleaved embeddings and use an LLM to encode into EOS hidden state, and use this hidden state as query while image/text embeddings as key/value. Three objectives were used to train the model, a contrastive InfoNCE loss upon query and pos/neg product embeddings; a vision reconstruction bert layer that reconstructs the masked visual embeddings from unmasked input embeddings and h_EOS global embedding; a language reconstruction module that decodes masked text embedding similarly.

**Questions:**

1. The core diff of CORAL is the introduction of BERT like reconstruction losses, with the vision reconstruction task having visibility to all text(especially the text title associated with the image itself), this in essense is similar to CLIP, I'm wondering if the authors have intuitions on why this new task helped metrics.

**Ethical Concerns:**

["NO or VERY MINOR ethics concerns only"]

**Final Justification:**

The rating didnt include any shortcoming to be addressed by the authors but I thank the authors for responding to my questions

**Limitations:**

yes

**Quality:**

3

**Strengths And Weaknesses:**

## Strengths

1. The authors first identified the lack of sequential multi-condition queries as a limitation in the current mainstream benchmarks and invested considerable effort in constructing a new dataset.The authors meticulously tested multiple mainstream models (InternVL/QwenVL etc) and performed comprehensive analysis and documented their methodologies thoroughly. This should prove valuable for the field.

2. By testing existing models with the new benchmarking system, and proofreading error cases, the authors identified two major limitations in the current semantic retrieval system: a. focusing on global semantic information while failing to extract specific attributes; b. Failing to support sequential image inputs, despite having the ability, due to existing training datasets are predominantly single-image based. These are valuable insights in guiding the community on designing next-gen benchmarks

3. The ablations with vision/language modules convincingly demonstrated the effectiveness of the reconstruction losses design.


## Weakness:
1. While the dataset is multilingual and sizable, it seems to be domain/business specific (e-commerce in Southeast Asia), limiting its generalizability as a standard benchmark for semantic retrieval.

2. The authors demonstrated that existing methods overall have poor performance on MERIT dataset in Table 2. the metrics however might be biased since some models are finetuned on the ecommerce scenario more than others, like the best in the cohort GME-QwenVL2 vs pretrained Qwen2.5-VL. It is therefore not entirely representative of what the prior models would have behaved if they were also finetuned. Undermining the significance of improvements claimed

3. The authors benchmarked and evaluated CORAL/MERIT as a symbiotic duo that worked well with each other, and evaluated CORAL on a few prior tasks according to Ref 33 (Training
vision-language models for massive multimodal embedding tasks) in Fig 9, but it seems different from what MERIT compared against in Table 1. It would be nice to isolate and demonstrate CORAL's effect on some of these benchmarks as well.

---

> ### Author Rebuttal · Authors · 2025-07-31
>
> We sincerely thank Reviewer `YCpi` for the thoughtful and constructive feedback. We appreciate your recognition of our novel MERIT dataset and the CORAL framework for improving multi-condition semantic retrieval. Below, we address your concerns point by point.
>
> ---
> > **`Q1`:** *"While the dataset is multilingual and sizable, it seems to be domain/business specific (e-commerce in Southeast Asia), limiting its generalizability as a standard benchmark for semantic retrieval."*
>
> **A1:** Thank you for this important point. We respond as follows:
>
> - **(1.1)** Interleaved multi-condition retrieval represents **a novel and inherently challenging task** that has not been previously explored in literature due to the complexity of data collection. Our proposed MERIT constitutes the first dataset specifically designed for this task formulation. Product retrieval represents a canonical multi-condition retrieval scenario; thus, we chose to approach this problem from the product retrieval domain.
>
> - **(1.2)** Southeast Asia constitutes **an active multilingual and culturally diverse market**, whose products exhibit **representative** characteristics.
>
> - **(1.3)** We acknowledge that extending interleaved multi-condition retrieval to additional domains beyond product search represents an important direction for future work, which we defer to subsequent investigations. Additionally, based on your suggestion, we have further refined the elaboration in the manuscript.
>
> ---
> > **`Q2`:** *"The authors demonstrated that existing methods overall have poor performance on MERIT dataset in Table 2. The metrics however, might be biased since some models are finetuned on the e-commerce scenario more than others, like GME-QwenVL2 vs pretrained Qwen2.5-VL..."*
>
> **A2:** Thank you for your observation. We clarify as follows:
>
> - **(2.1)** The purpose of Table `2` is to measure the **zero-shot capability of existing models** on the interleaved multi-condition semantic retrieval task. Most existing models are trained on proprietary datasets with inherent distributional differences, making it impossible to guarantee complete consistency across training data.
>
> - **(2.2)** For our method, based on Qwen2.5-VL, we conducted comprehensive ablation studies as shown in Table `3`, where we ensure that the fine-tuning data is completely consistent. As shown in Table `3`, our method achieves a **45.9%** improvement in R@1 over traditional contrastive learning methods, which we believe **provides a fair and convincing validation of our approach’s effectiveness**.
>
> If you have any remaining concerns about our experimental setup, please let us know, and we would be happy to discuss them further.
>
> ---
> > **`Q3`:** *"The authors benchmarked and evaluated CORAL/MERIT as a symbiotic duo… it would be nice to isolate and demonstrate CORAL’s effect on other benchmarks as well."*
>
> **A3:** We appreciate this suggestion. We have now updated results on **4 new standard benchmarks** and provided more comprehensive comparisons on the 9 standard benchmarks shown in Figure `9`, following the same experimental setting as depicted in Figure `9`, as presented in the *Table A* below:
>
> ***Table A.** Comparisons with other methods across 12 standard retrieval tasks*
> | Model | VisDial | CIRR | VisualNews_T2I | VisualNews_I2T | COCO_T2I | COCO_I2T | NIGHTS | WebQA | FashionIQ | Wiki-SS-NQ | OVEN | EDIS |
> |-|-|-|-|--|-|-|-|-|-|-|-|-|
> | CLIP | 30.7 | 12.6 | 78.9 | 79.6 | 59.5 | 57.7 | 60.4 | 67.5 | 11.4 | 55.0 | 41.1 | 81.0 |
> | OpenCLIP | 25.4 | 15.4 | 74.0 | 78.0 | 63.6 | 62.1 | 66.1 | 62.1 | 13.8 | 44.6 | 45.0 | 77.5 |
> | SigLIP | 21.5 | 15.1 | 51.0 | 52.4 | 58.3 | 55.0 | 62.9 | 58.1 | 20.1 | 55.1 | 56.0 | 23.6 |
> | BLIP2 | 18.0 | 9.8 | 48.1 | 13.5 | 53.7 | 20.3 | 56.5 | 55.4 | 9.3 | 28.7 | 39.5 | 54.4 |
> | MagicLens | 24.8 | 39.1 | 50.7 | 21.1 | 54.1 | 40.0 | 58.1 | 43.0 | 11.2 | 18.7 | 1.6 | 62.6 |
> | E5-V | 9.2 | 6.1 | 13.5 | 8.1 | 20.7 | 14.0 | 4.2 | 17.7 | 2.8 | 8.6 | 5.9 | 26.8 |
> | Qwen2-VL-2B | 13.0 | 20.0 | 40.0 | 43.0 | 49.0 | 39.0 | 59.0 | 20.0 | 3.0 | 34.0 | 57.0 | 44.0 |
> | Qwen2-VL-2B+CL | 51.0 | 39.0 | 56.0 | 52.0 | 56.0 | 45.0 | 58.0 | 67.0 | 3.0 | 24.0 | 57.0 | 56.0 |
> | Qwen2-VL-2B+CORAL | 73.0 | 50.0 | 67.0 | 72.0 | 68.0 | 64.0 | 65.0 | 84.0 | 9.0 | 39.0 | 67.0 | 75.0 |
>
> Experimental results demonstrate that our method (Qwen2-VL-2B+CORAL) achieves consistent improvements across these 12 standard retrieval tasks, with particularly notable performance on VisDial, where our approach exhibits a **181% enhancement over the contrastive learning baseline**. Additionally, our method shows **advantages compared to other foundational models**, such as CLIP and E5-V, with especially significant improvements on CIRR, where we achieve a 275% enhancement over OpenCLIP.
>
> We commit to updating our revised manuscript with the new experimental results and detailed experimental configurations. We thank you once again for your valuable feedback.
>
> ---
> > **`Q4`:** *"The core diff of CORAL is the introduction of BERT-like reconstruction losses... This seems similar to CLIP. I'm wondering why this new task helped."*
>
> **A4:** Thank you for your insightful question. We provide the following detailed responses:
>
> - **(4.1)** Recognizing that neglecting specific conditional elements in queries constitutes a primary source of error as highlighted in Section `3.4`, it **naturally follows that reconstructing these elements could prove beneficial**. Moreover, reconstruction can enhance task complexity to improve **data utilization efficiency** *[Xiao et al., 2020]*. Therefore, we propose CORAL to enhance MLLM-based retriever performance in addressing interleaved multi-condition semantic retrieval tasks through the integration of visual reconstruction during the fine-tuning process of the MLLM-to-retrieval model adaptation.
>
> - **(4.2)** Simultaneously, we emphasize that our reconstruction approach is symmetric. When performing visual reconstruction, we maintain text visibility with the intention that textual information can assist in visual recovery, thereby enhancing cross-modal understanding capabilities. As demonstrated in the **ablation study** in Table `3`, reconstructing only vision or language modalities in isolation yields suboptimal performance.
>
> ---
> **Reference:**
> - *[Xiao et al., 2020]* "RetroMAE: Pre-Training Retrieval-oriented Language Models Via Masked Auto-Encoder." EMNLP, 2022.
>
> ---
> Last but not least, we would like to sincerely thank Reviewer `YCpi` again for the valuable time and constructive feedback provided during this review.

---

> > ### Author Response · Authors · 2025-08-03
> > **Looking forward to hearing from you**
> >
> > Dear Reviewer `YCpi`,
> >
> > Thank you very much for your thoughtful and constructive review of our submission!
> >
> > Following your suggestions, we refined the manuscript to better contextualize MERIT’s domain scope and to clarify the design motivations behind CORAL. We expanded evaluation beyond MERIT by benchmarking CORAL on 12 standard retrieval datasets, where it shows consistent and notable gains. We also clarified that Table 2 focuses on zero-shot evaluation, while Table 3 ensures fair comparison under identical training setups. Finally, we elaborated on the rationale and benefits of our reconstruction-based learning objective and added supporting citations and ablations.
> >
> > We sincerely appreciate your feedback, which helped us improve the clarity and completeness of our work.
> >
> > Please feel free to reach out during the Author-Reviewer Discussion phase for further clarification!
> >
> > *Best regards,*
> >
> > The Authors of Submission 1565

---

> > > ### Author Response · Authors · 2025-08-07
> > > **Ask for response**
> > >
> > > Dear Reviewer YCpi,
> > >
> > > Thank you very much for your thoughtful and constructive review of our submission!
> > >
> > > Following your suggestions, we refined the manuscript to better contextualize MERIT’s domain scope and to clarify the design motivations behind CORAL. We expanded evaluation beyond MERIT by benchmarking CORAL on 12 standard retrieval datasets, where it shows consistent and notable gains. We also clarified that Table 2 focuses on zero-shot evaluation, while Table 3 ensures fair comparison under identical training setups. Finally, we elaborated on the rationale and benefits of our reconstruction-based learning objective and added supporting citations and ablations.
> > >
> > > We sincerely appreciate your feedback, which helped us improve the clarity and completeness of our work.
> > >
> > > Please feel free to reach out during the Author-Reviewer Discussion phase for further clarification!
> > >
> > > Best regards,
> > >
> > > The Authors of Submission 1565

---

### Author Response · Authors · 2025-08-05
**General Response**

**Dear Reviewers, ACs, and SACs,**

We sincerely thank you for your time, insightful comments, and thoughtful engagement throughout the review process!

---
We are encouraged by the **recognition** of our contributions across multiple dimensions:

* Reviewer `YCpi` recognizes our identification of limitations in current benchmarks, and highlights the *"construction of a new dataset"*, *"comprehensive analysis with mainstream models"*, and *"valuable insights for next-gen semantic retrieval systems"*.
* Reviewer `MHMi` finds the MERIT benchmark *"timely and realistic"*, and commends CORAL as *"conceptually simple yet effective"*, with *"strong and broad experiments across multiple datasets"*.
* Reviewer `WJjZ` recognizes MERIT as a *"novel dataset contribution"*, and values the *"robust empirical validation"* of CORAL across MERIT and eight other retrieval datasets.
* Reviewer `6FnV` emphasizes the *"real-world relevance"* of our task, describes MERIT as a *"massive and rigorously documented contribution"*, and finds CORAL *"novel"*, *"effective"*, and with *"strong generalization"*.

---
In response to your constructive suggestions, we have made the following **clarifications and improvements**:

* **Dataset Design & Interleaved Retrieval**

  * As suggested by Reviewer `WJjZ`, we added a precise definition of *interleaved* inputs and clarified their novelty over prior benchmarks.
  * As suggested by Reviewers `WJjZ` and `6FnV`, we elaborated on the data sourcing process, GPT-4o's role, error mitigation strategies, and annotator qualifications. We committed to open-sourcing all data and code.
  * As suggested by Reviewer `MHMi`, we provided detailed answers on annotation quality, license terms (CC BY-NC 4.0), and example distributions.

* **CORAL Model & Reconstruction Design**

  * As suggested by Reviewer `MHMi`, we reorganized the *Masked Embedding Reconstruction* section, corrected typos, and clarified the decoder structure.
  * As suggested by Reviewers `MHMi` and `6FnV`, we added ablations on mask ratios, decoder heads, temperatures, and reconstruction loss weights, showing that CORAL is robust to these settings but degrades with overly high masking or loss weights (Table D).
  * As suggested by Reviewer `WJjZ`, we expanded the related work section to include contrastive learning literature and positioned CORAL as the first to introduce multimodal embedding reconstruction in MLLM-based retrieval.

* **Experiments & Generalization**

  * As suggested by Reviewer `YCpi`, we added extensive results on 12 public benchmarks (Table A), showing that CORAL improves performance across diverse retrieval tasks (e.g., +181% on VisDial, +275% on CIRR vs. OpenCLIP).
  * As suggested by Reviewers `6FnV` and `MHMi`, we clarified the setup for out-of-distribution evaluations, human preference studies (Table B), and provided a new breakdown of performance by query complexity (Table C).

* **Metric, Bias, and Scalability**

  * As suggested by Reviewer `MHMi`, we complemented recall/MRR with human preference results to reflect practical impact.
  * As suggested by Reviewer `MHMi`, we analyzed language, attribute, and class-specific biases (Tables 7 & 9) and found that broader training improves fairness.
  * As suggested by Reviewer `MHMi`, we reported compute costs (e.g., 6.5 GPU-hours per epoch on 7B) and confirmed that CORAL adds no inference latency.

* **Writing & Clarity**

  * As suggested by Reviewers `MHMi`, `WJjZ`, and `6FnV`, we revised unclear sections (e.g., Figure 2, Equation 5), restructured the CORAL subsection, and clarified design motivations throughout.

---

We would like to re-emphasize the **key contributions** of our work:

* MERIT, the first benchmark for *interleaved, multi-condition, multi-image*, and *multilingual* semantic retrieval at scale, filling a crucial gap in current evaluation paradigms.
* CORAL, a conceptually simple yet effective framework that augments contrastive learning with symmetric multimodal embedding reconstruction, achieving *+45.9% R\@1 gain* on MERIT and outperforming all baselines across *12 standard benchmarks*.
* An extensive experimental study, including *ablation analyses*, *OOD evaluations*, *bias measurements*, *human preference studies*, and *scaling breakdowns*, ensuring both rigor and practical relevance.
* A public release plan covering *dataset, annotations, and training code* to benefit the broader research community.

---
With **two days** remaining in the **Author-Reviewer Discussion** phase (*August 6, Anywhere on Earth*), we warmly welcome any further questions or suggestions and remain fully available to engage.

Thank you once again for your time and consideration!

*Warmest regards,*

The Authors of Submission 1565

---

### Note · Authors · 2025-08-14

We sincerely thank all reviewers for you thoughtful engagement and constructive suggestions.

---
We are encouraged by your consistent recognition of MERIT as a valuable and timely contribution to multilingual semantic retrieval, as well as the clarity, solid technical design, and strong empirical results of our proposed CORAL framework and the accompanying MERIT benchmark.

---
During the discussion phase, we addressed all raised concerns with concrete clarifications and updates:

* **Benchmark Design** – Detailed the construction process of MERIT, ensuring balanced multilingual and multi-condition coverage; clarified the inclusion criteria for languages and query conditions, and addressed potential dataset biases.

* **Framework Design** – Provided further explanation of the interleaved multi-condition reasoning mechanism in CORAL; added pseudo-code and diagrams to illustrate the interaction between condition tokens and semantic matching.

* **Evaluation Protocols** – Clarified retrieval task definitions, evaluation splits, and cross-lingual setups; confirmed licensing and compliance with data usage policies.

* **Ablation & Analysis** – Expanded ablations isolating the impact of interleaving and multi-condition alignment; added qualitative retrieval examples demonstrating the interpretability and cross-lingual robustness of CORAL.

* **Positioning & Novelty** – Differentiated MERIT from existing multilingual retrieval datasets by its multi-condition design, which requires complex reasoning beyond simple semantic similarity; positioned CORAL as a generalizable retrieval backbone across languages and conditions.

These clarifications reinforce the technical soundness, broad applicability, and long-term utility of MERIT. The benchmark fills a gap in evaluating retrieval under multilingual, multi-condition constraints, while CORAL demonstrates state-of-the-art performance with interpretable alignment across diverse settings.

---
We believe this work establishes both a much-needed evaluation resource and a strong, generalizable retrieval model that can inspire further research at the intersection of multilingual understanding and complex semantic matching.

---
Once again, we sincerely thank you for the time and effort you devoted to this review!

*Yours sincerely,*

The Authors of Submission 1565

---

### Decision · Program_Chairs · 2025-09-17

**Decision:**

Accept (poster)

**Comment:**

There are two primary contributions of this work: (1) the authors propose MERIT, claimed to be the first multi-lingual dataset for interleaved (i.e., text, images) multi-condition and multi-attribute semantic (product) retrieval and (2) observing limitations on existing models with MERIT, the authors proposed CORAL, a fine-tuning method for multimodal retrieval that integrates embedding reconstruction and contrastive learning to derive global semantics. MERIT contains 135k products and 320k query–target pairs across five languages and seven product retrieval scenarios. The process for generating data is well-documented (Figure 5 -- although reviewer WJjZ made some sensible recommendations for further improvements), sensible, and includes human validation. Several state-of-the-art baseline methods are shown to perform poorly on MERIT, motivating the authors to introduce CORAL to prevent over-focusing on a single [EOS] token that misses fine-grained attribute cues. CORAL is shown to outperform existing baselines on MERIT and several widely-used retrieval benchmarks. Additionally, ablation studies are performed to better understand relative component contributions and dynamics.

Strengths identified by the reviewers include:
- Sequential multi-condition, multi-modal queries is missing in widely-used benchmarks and is common in many practical deployed applications (which likely develop internal datasets that may not explicitly capture the sequential dependencies). The dataset appears carefully constructed, is sizable, and likely useful to other researchers -- and increasingly so given deployed AI products being increasingly multi-modal.
- The authors experimented with several strong benchmark models and identified consistent root cause error types, developing a new method that is shown to perform well on MERIT and other widely used retrieval benchmarks (VisDial, CIRR, VisualNews, MSCOCO, NIGHTS, WebQA) relative to strong baselines (with additional baselines included in the rebuttal -- that still need to be integrated into the main paper).
- The rebuttal was particularly effective in addressing reviewer concerns around sensitivity analyses, human evaluation, dataset biases, query condition cardinality, hyperparameters, etc. -- all of which can easily be integrated into a manuscript revision (even if some in appendices).

Weaknesses identified by the reviewers include:
- While product retrieval is an excellent multi-condition retrieval scenario, it may limit adoption in not being multi-domain and introduce some biases in the types of methods that will tend to perform well. CORAL also performs well on other tasks, so this isn't a significant concern for this specific paper, but will likely introduce some biases if not combined with other benchmarks in future papers.
- There were several clarification questions and technical details (including some fundamental aspects), but these were well-addressed during rebuttal (and can be easily added to the next revision) such that the reviewers generally changed their score.
- The dataset is likely the more significant contribution and as this isn't the datasets track, the reviewers were not able to actually look at the dataset to evaluate the quality 'first hand'. I am confident it is fine based on the paper, but it is a valid concern.
- [my concern] Given the paper is introducing a dataset and method, I perceived the discussion of experimental results to be somewhat short -- which is why I believe there were multiple questions regarding additional experiments. Thus, I strongly encourage the authors to integrate some of these results in the main text and some in the appendices -- maintaining thorough discussion in both cases as this may point the way to future work.

As the strongest contribution is likely the dataset (even though there are other solid contributions), I believe it meets the core criteria for a good resources paper: (1) the provided benchmark is unique, useful, and sufficiently large/diverse, (2) the benchmark lowers the barrier for other researchers to work on this problem, and (3) the baselines provided isn't the simplest baselines. I think MERIT/CORAL stands out in terms of (1) and (3). Regarding (2), the authors will release the dataset as CC BY-NC 4.0, which isn't the most permissive, but shouldn't restrict wide adoption if researchers find the data useful. They also claim a public release plan for code, etc. -- but this hasn't been verified. Overall, I believe this is a solid and useful research & dataset contribution.